# Instructions and experiential learning have similar impacts on pain and pain-related brain responses but produce dissociations in value-based reversal learning

**Lauren Y Atlas[1,2,3]\*, Troy C Dildine[1,4], Esther E Palacios-Barrios[5], Qingbao Yu[1], Richard C Reynolds[3], Lauren A Banker[1], Shara S Grant[1], Daniel S Pine[3]**

[1]National Center for Complementary and Integrative Health, National Institutes of Health, Bethesda, United States; [2]National Institute on Drug Abuse, National Institutes of Health, Baltimore, United States; [3]National Institute of Mental Health, National Institutes of Health, Bethesda, United States; [4]Department of Clinical Neuroscience, Karolinska Institutet, Solna, Sweden; [5]Department of Psychology, University of Pittsburgh, Pittsburgh, United States

**\*For correspondence:**
lauren.atlas@nih.gov

**Competing interest:** The authors declare that no competing interests exist.

**Abstract** Recent data suggest that interactions between systems involved in higher order knowledge and associative learning drive responses during value-based learning. However, it is unknown how these systems impact subjective responses, such as pain. We tested how instructions and reversal learning influence pain and pain-evoked brain activation. Healthy volunteers (n=40) were either instructed about contingencies between cues and aversive outcomes or learned through experience in a paradigm where contingencies reversed three times. We measured predictive cue effects on pain and heat-evoked brain responses using functional magnetic resonance imaging. Predictive cues dynamically modulated pain perception as contingencies changed, regardless of whether participants received contingency instructions. Heat-evoked responses in the insula, anterior cingulate, and other regions updated as contingencies changed, and responses in the prefrontal cortex mediated dynamic cue effects on pain, whereas responses in the brainstem's rostroventral medulla (RVM) were shaped by initial contingencies throughout the task. Quantitative modeling revealed that expected value was shaped purely by instructions in the Instructed Group, whereas expected value updated dynamically in the Uninstructed Group as a function of error-based learning. These differences were accompanied by dissociations in the neural correlates of value-based learning in the rostral anterior cingulate, thalamus, and posterior insula, among other regions. These results show how predictions dynamically impact subjective pain. Moreover, imaging data delineate three types of networks involved in pain generation and value-based learning: those that respond to initial contingencies, those that update dynamically during feedback-driven learning as contingencies change, and those that are sensitive to instruction. Together, these findings provide multiple points of entry for therapies designs to impact pain.

## Editor's evaluation

This fundamental paper advances our understanding of the commonalities and differences in the neural basis of directly experienced and instructed aversive learning in humans. The study uses compelling experimental design and analyses along with functional magnetic resonance imaging

(fMRI) to inform neuro-computational models of how explicitly informed vs experientially acquired information influences learning about cues predicting painful stimuli. This work will be of broad interest to neuroscientists interested in pain and aversive learning and memory.

## Introduction

Predictions and expectations shape perception across many domains, through processes such as predictive coding. This is particularly apparent in the context of pain as evidenced by data on placebo analgesia and expectancy-based pain modulation (*Büchel et al., 2014*; *Ongaro and Kaptchuk, 2018*; *Kaptchuk et al., 2020*). While most studies of predictive coding examine probabilistic error-driven learning, humans also use verbal instructions to shape predictions, with instructions acting either alone or through effects on learning (for reviews, see *Koban et al., 2017*; *Mertens et al., 2018*; *Atlas, 2019*). Placebo analgesia depends on expectations formed through conditioning or associative learning (e.g. prior treatment experiences) as well as verbal instruction and explicit knowledge (e.g. the doctor's instruction), yet it is unknown how these factors combine dynamically to shape pain and pain-related brain responses. We introduced a novel pain reversal learning task to measure the dynamic effects of predictive cues on subjective pain and brain responses to noxious heat and isolate whether instructions and learning shape pain through independent mechanisms.

Most studies of placebo analgesia combine suggestion and conditioning to maximize expectations and measure downstream responses. These experiments indicate that placebos reliably reduce acute pain (*Forsberg et al., 2017*; *Zunhammer et al., 2018*) and alter stimulus-evoked responses in multiple brain regions, including the insula, dorsal anterior cingulate, and thalamus, as well as pain modulatory regions including the opioid-rich periaqueductal gray (PAG), the dorsolateral prefrontal cortex (DLPFC), and the rostral anterior cingulate cortex (rACC) (*Atlas and Wager, 2014b*). To what extent do these mechanisms depend on instructed knowledge or associative learning? Behavioral experiments indicate the potential for dissociations (*Montgomery and Kirsch, 1996*; *Benedetti et al., 2003*; *Colloca et al., 2008a*; *Colloca et al., 2008b*). In one study (*Benedetti et al., 2003*), participants underwent several days of conditioning with active treatments for pain, motor performance in Parkinson's disease, or drugs that affect hormonal responses (cortisol or growth hormone). Participants subsequently received verbal instructions that they would receive a drug that leads to the opposite effect of conditioning. All participants actually received placebo. Placebo effects on outcomes that could be consciously monitored (pain and motor responding) reversed with instruction, while hormonal responses continued to mimic conditioning. Other studies indicate that instructions only reverse placebo analgesia after brief conditioning (*Schafer et al., 2015*). Thus, placebo effects on specific outcomes manifest unique sensitivities to instructed knowledge alone or through effects on learning, suggesting the two processes may act through distinct mechanisms.

These behavioral studies also highlight the use of instructed reversals to distinguish between purely associative processes and those that are sensitive to higher order knowledge. This connects placebo with an established literature on how instructions influence appetitive and aversive learning (*Grings, 1973*; *McNally, 1981*; *Costa et al., 2015*; *Mertens and De Houwer, 2016*; *Atlas, 2019*). Neuroimaging studies of reinforcement learning indicate that instructions can shape reward learning, and that this occurs through interactions between the DLPFC and striatum (*Doll et al., 2009*; *Doll et al., 2011*; *Li et al., 2011a*). We previously showed that corticostriatal interactions also support the effect of instructed reversals on aversive learning, but that the amygdala learned from aversive outcomes irrespective of instruction (*Atlas et al., 2016*; *Atlas, 2019*). This provides a potential mechanism by which some outcomes may continue to respond to associative learning in spite of instructions, while others may update with instruction, consistent with behavioral dissociations (*Benedetti et al., 2003*). Importantly, most previous work on how instructions shape learning has measured autonomic responses during classical conditioning or binary choices in instrumental learning tasks. Acute pain tasks provide a unique opportunity to measure how learning and instructions shape conscious, subjective decisions, which are likely to be distinct from autonomic responses or instrumental choice.

We asked how instructions and learning combine to dynamically shape pain and pain-related brain responses. Participants underwent a pain reversal learning task and were assigned to an Instructed Group, who was informed about contingencies and reversals, or an Uninstructed Group, who learned purely through experience (*Figure 1*). We used multilevel mediation analysis to identify brain regions

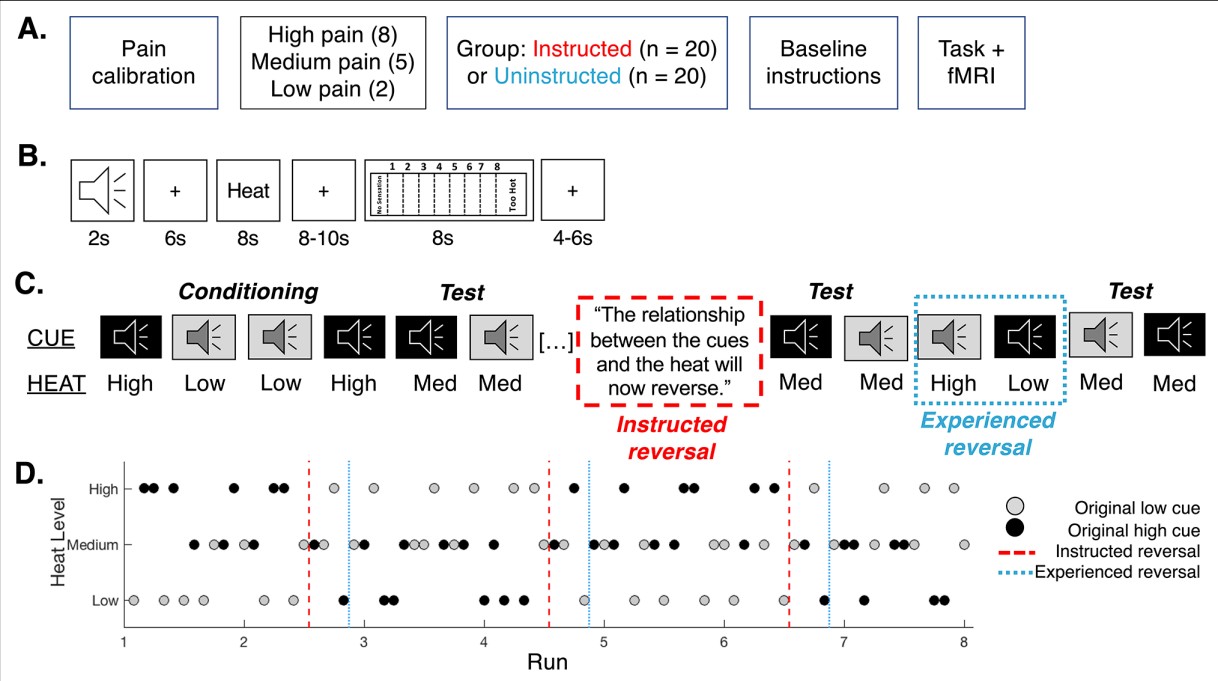

**Figure 1.** Experimental design. (**A**) Experimental design. Participants underwent a pain calibration that identified temperatures corresponding to maximum tolerable pain (high pain; 8), pain threshold (low pain; 2), or medium pain (5). They were then positioned in the fMRI scanner and randomly assigned to group. Participants in the Instructed Group were informed about contingencies, while participants in the Uninstructed Group were told to pay attention to the associations between auditory cues and heat but were not informed about the specific cue-outcome contingencies. (**B**) Trial structure. On each trial, a 2 s auditory cue preceded heat delivered to the participants left forearm. Participants rated perceived pain following offset using an 8-point continuous visual analogue scale. Trials were 48 s long. (**C**) Instructed and experience-based reversals. Participants first underwent a brief conditioning phase of 5–6 trials in which Original Low Cues (gray) were followed by heat calibrated to elicit low pain (level 2) and Original High Cues (black) were followed by heat calibrated to elicit high pain (level 8). Conditioning was immediately followed by intermittent test trials, in which we delivered medium heat following each cue to test the effects of predictive cues on perceived pain. Following the initial test phase, participants in the Instructed Group were informed about reversals and we delivered medium stimuli to test the effects of instructions. We then paired high heat with the Original Low cue and low heat with the Original High cue, which should act as an experiential reversal, and again administered medium heat to test whether pain reverses upon experience. (**D**) Example trial order. There were three reversals across the entire task. We used two trial orders that were counterbalanced across participants.

that are modulated by instructions or learning and modulate subjective pain. We also fit computational models of instructed learning (*Atlas et al., 2016*; *Atlas et al., 2019*) to pain ratings to determine how instructions and associative learning dynamically shape pain, and to isolate brain regions that track expected value during pain reversal learning. We were most interested in understanding how instructions and learning affect brain responses within brain networks involved in pain and value-based learning. We hypothesized that instructions and learning would both dynamically shape pain, and that instructed reversals would lead to immediate reversals of pain reports and heat-evoked brain responses in the DLPFC and pain processing network.

## Results

### Heat intensity effects on pain, autonomic responses, and brain responses to noxious heat are similar across groups

Prior to the fMRI experiment, all participants underwent an adaptive pain calibration procedure (*Atlas et al., 2010*; *Mischkowski et al., 2019*; *Dildine et al., 2020*; *Amir et al., 2021*) to identify each participant's pain threshold, tolerance, and the reliability of the temperature-pain association (i.e. $r^2$; see Materials and methods). Consistent with our IRB protocol, four participants were dismissed prior to the fMRI portion of the experiment due to low reliability (n=3) or pain tolerance above 50°C (n=1). For

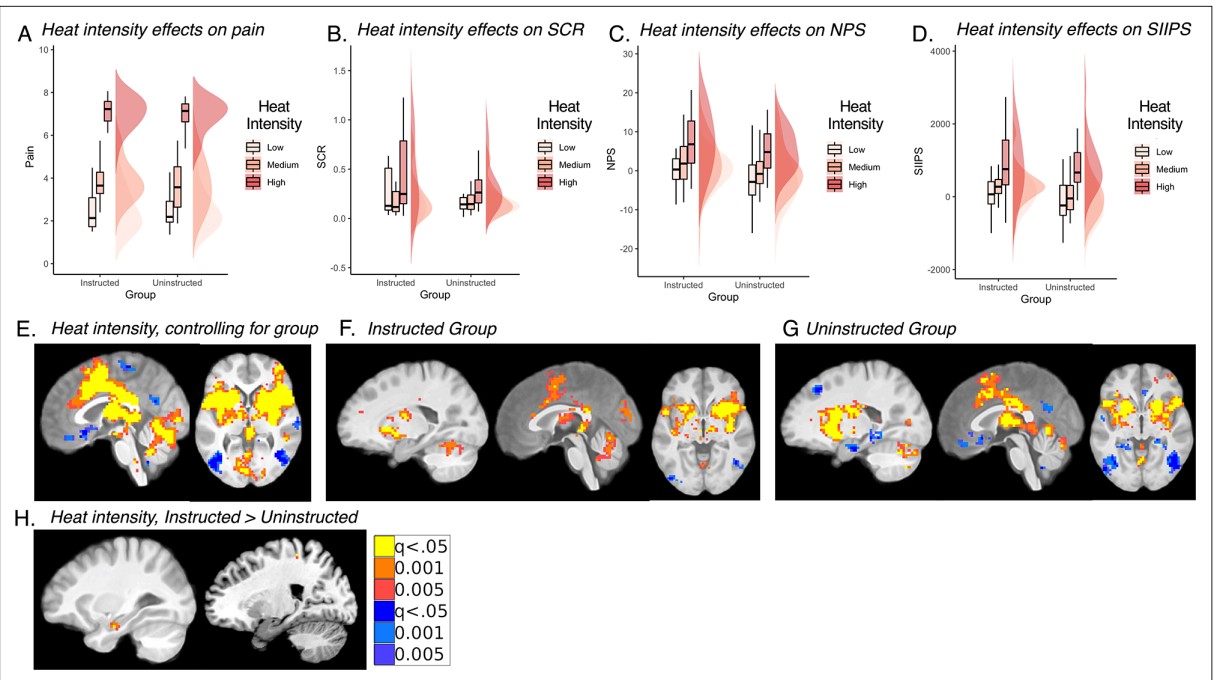

**Figure 2.** Effects of heat intensity on pain, autonomic responses, and brain responses to noxious heat. (**A-D**) There were no differences between groups in the effect of heat intensity on pain (**A**), skin conductance responses (SCR; **B**), or pattern expression in the neurologic pain signature (NPS; **C**) or stimulus-intensity independent pain signature (SIIPS; **D**). All outcomes showed robust effects of heat intensity (see *Tables 1–2* and *Figure 2—source data 1*). Data were visualized using the R toolboxes ggplot2 (*Wickham, 2016*) and Raincloud plots (*Allen et al., 2021*). (**E-H**): Whole-brain voxel-wise analyses revealed robust effects of variations in stimulus intensity on heat-evoked activation within brain regions involved in pain, controlling for group (**E**), which were confirmed with separate analyses within the Instructed Group (**F**) and the Uninstructed Group (**G**). FDR-corrected p-values for contrasts E-G exceeded *P*<.005 and therefore we used maps thresholded at *P*<.001 for inference. Only the left hippocampus and right S1 showed significant group differences at corrected levels (**H**). Differences were driven by temperature-induced deactivation in the Uninstructed Group, as depicted in G. For additional regions identified in voxelwise results, see *Figure 2—figure supplement 1* and *Figure 2—source data 2 and 3*.

The online version of this article includes the following source data and figure supplement(s) for figure 2:

**Source data 1.** Heat intensity effects on heat-evoked autonomic responses.

**Source data 2.** Heat intensity effects: Small-volumes corrected results.

**Source data 3.** Heat intensity effects: Uncorrected results.

**Figure supplement 1.** Heat intensity effects: Whole-brain uncorrected results.

each participant who continued to the fMRI phase, we used linear regression to identify temperatures associated with ratings of low pain (*M*=42.04 °C, SE = 0.43), medium pain (*M*=44.71 °C, SE = 0.37), and high pain (*M*=47.30 °C, SE = 0.30). There were no differences between groups in the reliability of the association between temperature and pain, as measured by $r^2$ (*M*=0.803, SE = 0.022; p>0.2), or in temperatures applied during the task (all p's>0.1).

We next examined pain as a function of heat intensity (i.e. temperature level: low, medium, or high) during the fMRI experiment (see *Figure 2A*). Bayesian model comparison indicated that the best model included fixed effects of Heat Intensity, Cue, Phase, and Group and all possible interactions, along with random intercepts and slopes for all factors. All models revealed significant effects of Heat Intensity, Cue, Phase, Cue x Phase, and Heat Intensity x Cue x Phase interactions across participants (see *Table 1*). We also observed a significant Group x Cue x Phase interaction and a significant Group x Heat Intensity x Cue x Phase interaction, which were likely to be driven by the critical medium heat trials, as reported below. Bayesian posterior estimates indicated that the effects of Heat Intensity, Cue x Phase interactions, and Heat Intensity x Cue x Phase interactions were practically significant with enough evidence to reject the null (<1% in ROPE), while the main effect of Phase supported the null (i.e. no effect of Phase; 99.8% in ROPE), despite being statistically significant. All other effects were of undecided significance (i.e. not enough evidence to accept or reject the null); complete results are

**Table 1.** Heat intensity effects on pain across all participants (n=40)[*].

| Predictors | Estimates | | | Confidence intervals | | | P-Value / probability of direction | | | | Bayesian estimates | |
| --- | --- | --- | --- | --- | --- | --- | --- | --- | --- | --- | --- | --- |
| | LMER[†] | NLME[‡] | BRMS[§] | LMER[†] | NLME[‡] | BRMS[§] | LMER[†] | NLME[‡] | BRMS[§] | % in ROPE | Rhat | ESS |
| (Intercept) | 3.64 | 3.646 | 3.641 | 3.40–3.88 | [3.408, 3.885] | [3.438, 3.849] | <0.001 | 0.000 | 100.00% | 0 | 1.005 | 1597.656 |
| Group | 0.11 | 0.101 | 0.112 | -0.13–0.35 | [-0.144, 0.347] | [-0.092, 0.318] | 0.356 | 0.409 | 80.83% | 82.783 | 1.001 | 1632.917 |
| **Heat Level** | 2.09 | 2.078 | 2.083 | 1.90–2.27 | [1.897, 2.258] | [1.929, 2.238] | <0.001 | 0.000 | 100.00% | 0 | 1 | 4205.817 |
| Cue | 0.32 | 0.293 | 0.323 | 0.17–0.47 | [0.148, 0.437] | [0.203, 0.447] | <0.001 | 0.000 | 100.00% | 12.692 | 1 | 9174.052 |
| Phase | 0.1 | 0.107 | 0.099 | 0.01–0.19 | [0.022, 0.193] | [0.026, 0.170] | 0.026 | 0.014 | 98.40% | 99.842 | 1.001 | 7792.54 |
| Group x Heat Level | -0.04 | -0.064 | -0.042 | -0.23–0.14 | [-0.245, 0.116] | [-0.195, 0.113] | 0.636 | 0.484 | 67.15% | 97.225 | 1.001 | 4410.973 |
| Group x Cue | 0.1 | 0.115 | 0.098 | -0.05–0.25 | [-0.029, 0.259] | [-0.025, 0.220] | 0.197 | 0.119 | 90.05% | 96.508 | 1 | 8745.241 |
| Heat Level x Cue | 0.16 | 0.128 | 0.158 | -0.03–0.35 | [-0.055, 0.311] | [0.002, 0.307] | 0.097 | 0.169 | 95.27% | 79.3 | 1 | 17260.576 |
| Group x Phase | 0 | -0.001 | -4.95E-04 | -0.09–0.09 | [-0.086, 0.085] | [-0.074, 0.071] | 0.999 | 0.985 | 50.43% | 100 | 1 | 8609.937 |
| Heat Level * Phase | -0.06 | -0.055 | -0.059 | -0.15–0.04 | [-0.139, 0.03] | [-0.133, 0.017] | 0.23 | 0.205 | 89.04% | 100 | 1 | 15179.351 |
| **Cue * Phase** | 0.58 | 0.615 | 0.575 | 0.39–0.77 | [0.424, 0.806] | [0.410, 0.731] | <0.001 | 0.000 | 100.00% | 0.025 | 1 | 9874.616 |
| (Group * Heat Level) * Cue | -0.04 | -0.033 | -0.036 | -0.22–0.15 | [-0.216, 0.15] | [-0.185, 0.112] | 0.709 | 0.724 | 64.35% | 98.242 | 1 | 17007.967 |
| (Group * Heat Level) * Phase | -0.03 | -0.037 | -0.029 | -0.12–0.07 | [-0.121, 0.047] | [-0.105, 0.049] | 0.546 | 0.390 | 73.22% | 100 | 1 | 14459.934 |
| (Group *Cue) * Phase | 0.23 | 0.249 | 0.231 | 0.04–0.42 | [0.058, 0.44] | [0.073, 0.392] | 0.017 | 0.011 | 98.79% | 52.8 | 1 | 10768.063 |
| **(Heat Level *Cue) *Phase** | 1.79 | 1.774 | 1.782 | 1.60–1.97 | [1.59, 1.958] | [1.634, 1.939] | <0.001 | 0.000 | 100.00% | 0 | 1 | 21294.817 |
| (Group *Heat Level *Cue) *Phase | -0.2 | -0.167 | -0.203 | -0.39 to -0.01 | [-0.351, 0.018] | [-0.359,-0.054] | 0.038 | 0.076 | 98.48% | 63.242 | 1 | 23048.272 |

[*]This table presents results of linear mixed models predicting subjective pain as a function of Heat Level (High vs Medium vs Low), Group (Instructed vs Uninstructed), Cue (Original High vs Original Low), and Phase (Original vs Reversed). All predictors were dummy-coded and mean centered to facilitate interpretation of coefficients and interactions. Model specification was based on Bayesian model comparison. We compared three types of linear mixed models: frequentist analysis using the "lme" function of nlme (*Pinheiro et al., 2021*) accounting for autoregression, and Bayesian estimation using mildly informative conservative priors (i.e. centered on 0 for all effects). Effects that are both statistically and practically significant are bolded, whereas effects that are statistically significant but not practically significant (i.e. >2.5% in the region of partial equivalence (ROPE)) are italicized.

[†]Estimates based on a linear mixed effects model implemented in the "lmer" function of lme4 (*Bates et al., 2015*) using the following code: lmer(Pain~Group*Templevels*Cue*Phase+(1+Templevels + Cue*Phase||Subject)). Confidence intervals were obtained using the "tab_model" function from sjPlot (*Lüdecke, 2021*) and corresponds to the 95% confidence interval.

[‡]Estimates based on a linear mixed effects model implemented in the "lme" function of nlme (*Pinheiro et al., 2021*) including autoregression using the following code: lme(Pain~Group*Templevels*Cue*Phase, random = ~1 + Templevels +Cue*Phase|Subject, correlation = corAR10, na.action=na.exclude). Confidence intervals were obtained using the 'intervals' function from nlme (*Pinheiro et al., 2021*) and corresponds to the 95% confidence interval.

[§]Estimates based on Bayesian model linear mixed models using the "brms" function (*Bürkner, 2017*) using the following code: brm Pain~Group*Templevels*Cue*Phase+(1+Templevels + Cue*Phase|Subject,prior = set_prior("normal(0,2.5)", class="b"), save_all_pars = TRUE, silent = TRUE, refresh = 0, iter = 4000, warmup = 1000). Posterior estimates, including the probable direction (which is roughly equivalent to 1- frequentist p-value), 89% confidence intervals, and the ROPE were obtained using the "describe_posterior" function from the package BayesTestR (*Makowski et al., 2019a*) and interpreted as in *Makowski et al., 2019b*. The Region of Partial Equivalence (ROPE) was defined as [−0.237, 0.237]. We report the median estimate for each parameter.

reported in *Table 1*. We observed similar results when we restricted analyses to pain ratings from the 36 participants with useable fMRI data; see *Supplementary file 1*.

Next, we analyzed heat-evoked autonomic responses during the experiment. SCR and pupil dilation were both significantly influenced by Heat Intensity and exhibited Heat Intensity x Cue X Phase interactions (see *Figure 2—source data 1*). Both factors had practically significant effects on SCR (<1% in ROPE), whereas Bayesian analyses of pupillary outcomes indicated that evidence was not sufficient to reject the null hypothesis (100% in ROPE). Because there was no meaningful effect of temperature on pupil dilation and the number of subjects with useable pupil data was substantially less than those with useable skin conductance, we focused on SCR in subsequent analyses of cue effects on physiological arousal. There was no main effect of Group on pupil dilation or SCR, nor any interactions between Group and Heat Intensity for either outcome, suggesting that temperature effects on physiological arousal were similar regardless of whether individuals were instructed about contingencies (see *Figure 2B* and *Figure 2—source data 1*). For complete results, see *Figure 2—source data 1*.

We also evaluated brain responses to noxious stimulation as a function of heat intensity. We note that FDR-corrected thresholds exceeded 0.001 for all voxelwise analyses apart from moderation by group; we therefore interpret main effects of heat intensity at p<0.001. We observed robust intensity-related changes within pain modulatory regions, including bilateral insula, striatum, dorsal anterior cingulate, thalamus, and other regions that did not differ between groups (see *Figure 2E–G* and *Figure 2—source data 2 and 3*). Consistent with this, we observed robust expression of both the Neurologic Signature Pattern (NPS; *Wager et al., 2013*) and Stimulus-Intensity Independent Pain Signature (SIIPS; *Woo et al., 2017*) as a function of temperature-related changes in both groups (all p's<0.001, see *Table 2*) and signature pattern expression did not differ by group (all p's>0.2; see *Figure 2C&D* and *Table 2*). Thus variations in heat intensity were positively associated with increases in pain-related activation in pain-related regions regardless of whether individuals were instructed about contingencies. Whole brain FDR-correction did reveal significant group differences in the left hippocampus and right primary somatosensory cortex driven by stronger intensity effects in the Instructed Group (see *Figure 2H* and *Figure 2—source data 2 and 3*). Within value-related ROIs, we observed positive effects of heat intensity on the bilateral striatum that did not differ by Group, whereas the VMPFC showed significant Group differences, driven by negative associations between temperature and VMPFC activation in the Uninstructed Group, but not the Instructed Group (see *Table 2*). There were no associations between heat intensity and amygdala activation.

## Predictive cues modulate expectations and pain whether learned through instruction or experience

Analyses across all trials indicated potential influences of predictive cues and cue-based reversals on pain, as indicated by the Cue x Phase and Heat Intensity x Cue x Phase interactions. To measure cue-based expectancy effects more directly, we measured cue effects on (1) expectancy ratings and (2) pain reports on medium heat trials, which were crossed with predictive cues. We first examined expectations as a function of Cue prior to conditioning, that is immediately after instruction. Consistent with our manipulation, there was a significant Group x Cue interaction on expectancy at baseline (F(1,38) = 8.959, p=0.005), driven by significant differences in the Instructed Group (p=0.0027) but not the Uninstructed Group (p>0.3), as shown in *Figure 3A*. There were no main effects of Group or Cue prior to conditioning (all p's>0.1). Following the first acquisition block, we collected a second set of expectancy ratings. We again observed a significant Group x Cue interaction (F(1,38) = 7.102, p=0.011) as well as a main effect of Cue (F(1,38) = 31.195, p<0.001). Post-hoc comparisons indicated that both groups reported higher expectancy with the high pain cue (see *Figure 3A*), but that differences were larger in the Instructed Group (p<0.001), relative to the Uninstructed Group (p=0.003). Thus, instructions and learning both modulated cue-based expectations about pain.

We next asked whether cue-based expectations in turn modulate subjective pain on medium heat trials. We first measured effects of cues on pain ratings during the acquisition phase, that is prior to the first reversal, and asked whether effects vary based on whether learning is paired with verbal instruction. Bayesian model comparison indicated that the best model included fixed effects of Group, Cue, and Trial, with random intercepts and random slopes for Cue and Trial. Consistent with other studies of expectancy-based pain modulation (*Atlas et al., 2010*; *Wiech et al., 2014*; *Reicherts et al., 2016*;

**Table 2.** Effects of heat, cues, and learning on responses in value-related regions of interest and pain-related signature patterns*.

| Analysis | Effect | Left striatum | Right striatum | Left amygdala | Right amygdala | VMPFC | NPS | SIIPS |
|---|---|---|---|---|---|---|---|---|
| | All participants, controlling for Group | b=0.19, p<0.001 | b=0.14, p<0.001 | - | - | b=−0.27, p<0.001 | b=3.74, p<0.001 | b=439.87, p<0.001 |
| | Instructed vs Uninstructed | - | - | - | - | b=0.15, p=0.048 | - | - |
| | Instructed Group | CI = [0.08 0.22], t(17) = 3.05, p=0.007 | CI = [0.14 0.25], t(17) = 5.37, p<0.001 | - | - | ns | CI = [2.39 5.49]; t(35) = 5.49; p<0.001 | CI = [197.59 572.36], t(35) = 4.33; p<0.001 |
| Effect of heat intensity | Uninstructed Group | CI = [0.07 0.25], t(17) = 3.60, p=0.002 | CI = [0.12 0.28], t(17) = 5.34, p<0.001 | - | - | CI = [-0.67–0.19], t(17) = −3.81, p=0.001 | CI = [2.28 4.91]; t(17) = 5.77; p<0.001 | CI = [317.40 672.13]; t(17) = 5.89; p<0.001 |
| | Path a | a=0.05, p=0.058 | a=0.05, p=0.079 | - | - | ns | ns | n.s. |
| Mediation of current cue contingencies | Path b | b=0.13, p=0.007 | b=0.16, p<0.001 | - | - | ns | b=0.01, p=0.004 | b=0.00, p<0.001 |
| | Path a*b | - | - | - | - | ns | n.s. | n.s. |
| | Path a | - | - | - | - | a=−0.09, p=0.015 | n.s. | n.s. |
| | Path b | b=0.13, p=0.006 | b=0.16, p=0.001 | - | - | ns | b=0.01, p=0.006 | b=0.00, p<0.001 |
| Mediation of original cue contingencies | Path a*b | - | - | - | - | ns | ns | a*b=0.01, p=0.065 |
| | All participants, controlling for Group | - | - | - | - | - | - | - |
| | Instructed vs Uninstructed | - | b=0.24, p=0.03 | - | - | - | - | - |
| | Instructed Group | CI = [0.078 0.51]; t(17) = 2.85; p=0.011 | CI = [0.03 0.42]; t(17) = 2.47; p=0.024 | - | - | - | - | - |
| Association with expected value based on fits to pain | Uninstructed Group | - | - | - | - | - | - | - |
| Association with unsigned prediction error | All participants, controlling for Group | b=1.02, p=0.003 | b=0.67, p=0.062 | b=1.61, p=0.004 | b=1.31, p=0.007 | - | - | - |
| | Instructed vs Uninstructed | - | - | - | - | - | - | - |
| | Instructed Group | - | CI = [0.17 2.59]; t(17) = 2.41; p=0.028 | CI = [0.57 3.40]; t(17) = 2.96; p=0.009 | CI = [0.88 3.77]; t(17) = 3.39; p=0.004 | - | - | - |
| | Uninstructed Group | - | CI = [0.06 1.26]; t(17) = 2.33; p=0.033 | - | - | - | - | - |

*Table 2 continued on next page*

*Table 2 continued*

| Analysis | Effect | Left striatum | Right striatum | Left amygdala | Right amygdala | VMPFC | NPS | SIIPS |
|---|---|---|---|---|---|---|---|---|
| | Instruction vs Feedback-driven EV | - | - | - | - | - | - | - |
| Instructed vs feedback-driven expected value within Instructed Participants | Instruction-based EV | CI = [0.07 0.53]; t(17) = 2.73; p=0.014 | CI = [0.02 0.44]; t(17) = 2.33; p=0.03 | - | - | - | - | - |
| | Feedback-driven EV | - | - | - | - | - | - | - |

*This table reports results of tests within a priori regions of interest (ROIs) involved in expected value and pain-related signature patterns, the Neurologic Pain Signature (NPS; **Wager et al., 2013**) and the Stimulus Intensity Independent Pain Signature (SIIPS; **Woo et al., 2017**). For mediation analyses, trial-level responses (i.e. area-under-the-curve estimates) were extracted and averaged across each ROI or computed as the dot-product between trial estimates and pattern expression for NPS and SIIPS, and then multilevel mediation analyses were evaluated. For regressions with heat intensity, expected value, and unsigned prediction error, we used linear models and one-sample t-tests across beta estimates and contrast maps. See Materials and Methods for additional details and **Figure 5—figure supplement 1** for ROI images.

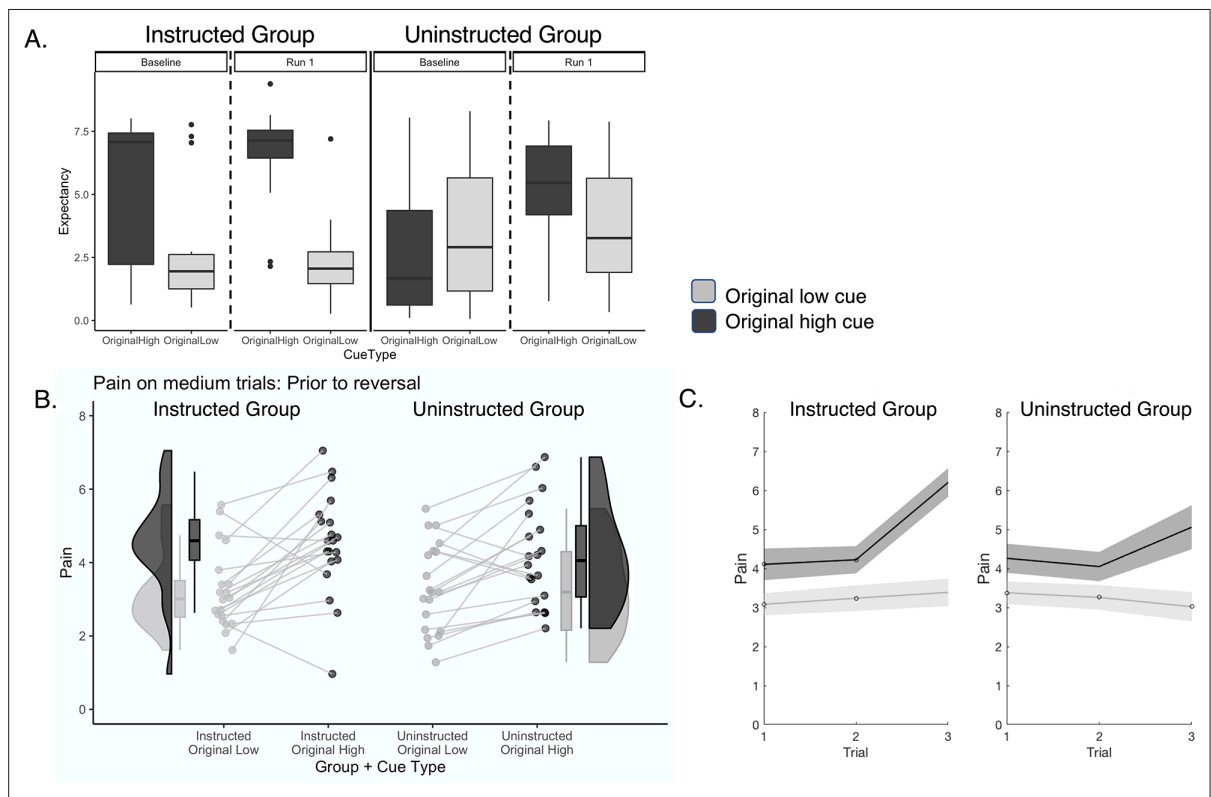

**Figure 3.** Effects of instructions and learning on expected pain and pain ratings prior to reversal. (**A**) Expectancy ratings prior to reversal. Participants in the Instructed Group (Top Left) expected higher pain in response to the Original High Cue relative to the Original Low Cue at baseline (left) and differences in expectations grew larger following conditioning and the first test phase (right). Participants in the Uninstructed Group did not report differences prior to the task (left), consistent with the fact that they were not instructed about specific cue-outcome contingencies. Following conditioning and the first test phase, Uninstructed Group participants expected higher pain in response to the Original High Cue, relative to the Original Low Cue. Cue-based differences in expectancy ratings were larger in the Instructed Group. (**B**) *Predictive cue effects on pain prior to reversal.* We measured the effects of predictive cues on perceived pain prior to the first reversal (see **Table 3**). Both groups reported higher pain when medium heat was preceded by the high pain cue (black) relative to the low pain cue (gray) and this effect was present in nearly all participants. (**C**) *Cue effects increase over time.* Both groups show larger cue-based differences in perceived pain on medium heat trials as a function of experience prior to the first reversal, but effects of time were larger in the Instructed Group. Data were visualized using the R toolboxes ggplot2 (**Wickham, 2016**) and Raincloud plots (**Allen et al., 2021**). Error bars and shaded areas denote standard error of the mean (n = 20 per group).

*Fazeli and Büchel, 2018*; *Michalska et al., 2018*; *Abend et al., 2021*), all models indicated that participants reported higher pain when medium heat was preceded by high pain cues than low pain cues (main effect of Cue: see *Figure 3B* and *Table 3*), and this effect was practically significant based on Bayesian modeling (0% in ROPE). There was a significant Group x Cue interaction (see *Table 3*) which was of undecided significance (8% in ROPE). Importantly, post-hoc analyses within groups indicated that both groups reported practically significant effects of Cue on pain prior to the first reversal (see *Figure 3B* and *Table 3*), although effects were larger in the Instructed Group. We also observed a statistically significant Group x Cue x Trial interaction, although this was of undecided significance (35.45% in ROPE; see *Table 3*). Post-hoc analyses within groups indicated that Cue effects increased over time in the Instructed Group (see *Figure 3C* and *Table 3*), as did pain reports overall, whereas there were no interactions with time in Uninstructed Group participants. Together, these results indicate that instructions and learning both shape pain prior to reversal, that effects are somewhat larger in Instructed Group participants, and that the dynamics of expectancy effects on pain may differ as a function of whether individuals learn from experience or instruction. For complete results, please see *Table 3*.

## Cue-based expectations and cue effects on pain and SCR update as contingencies reverse

We next tested whether expectations and cue effects on pain updated as contingencies reversed, and whether they did so differently as a function of instruction. We computed an expectancy rating difference score (Original High Pain expectancy – Original Low Pain expectancy; see *Figure 4B*) for each pre-block rating and measured effects across the entire task as a function of Group and Phase (i.e. Original vs. Reversed Contingencies; see vertical dashed lines in *Figures 1D and 4A*). We observed a main effect of Phase (B=–2.03, p<0.001), indicating that differential expectations varied as contingencies reversed, and significant Group x Phase interaction (B=4.12, p<0.001). Post-hoc analyses indicated that only the Instructed Group reported differences in expectation that varied significantly as a function of Phase, whereas the Uninstructed Group showed weaker variations in expectations as contingencies reversed (see *Figure 4A and B*).

We next examined pain reports in response to medium heat across all trials, including reversals (see *Figure 4*). Bayesian model comparison using a normal distribution indicated the most likely model included fixed effects of Group, Cue, Phase, and Trial, with random intercepts and slopes. All models revealed significant Cue x Phase interactions on pain, indicating that cue effects on pain varied as contingencies reversed (see *Figure 4C and D* and *Table 4*), and this effect was sufficient to reject the null hypothesis of no interaction (<1% in ROPE). All models also revealed main effects of Cue, such that individuals reported higher pain in response to the original high pain cue than the original low pain cue, and main effects of Phase, such that pain was higher on original contingencies relative to reversals, and these effects were significant in frequentist analyses but were of undecided significance based on Bayesian estimates (see *Table 4*). Finally, frequentist analysis approaches revealed significant Group x Cue x Phase interactions, driven by stronger reversals of Cue effects in the Instructed Group (see *Figure 4D*). Post hoc analyses conducted separately by Group indicated nearly 100% probability of positive Cue x Phase interactions in each group, although evidence was only sufficient to reject the null hypothesis in the Instructed Group (see *Table 4*). We observed similar results when we restricted analyses to pain ratings from the 36 participants with useable fMRI data, although the Group x Cue x Phase interaction was marginally significant in frequentist approaches; see *Supplementary file 1* for complete details. We also observed consistent findings when we tested the model with a beta distribution, which was found to provide better fits based on posterior prediction (see *Supplementary file 2*). Thus predictive cues shape pain perception even as contingencies change, whether or not participants are instructed about contingencies. In addition, reversals may be slightly larger in participants who are explicitly instructed about contingencies and reversals, however group differences were not practically meaningful based on Bayesian statistics.

We also tested whether cues and reversals impacted physiological responses to medium heat, as measured by heat-evoked SCR. Heat-evoked SCRs were influenced by predictive cues on medium trials and reversed as contingencies changed, but the magnitude of these differences did not differ by Group (see *Figure 4D* and *Figure 4—source data 1*). While effects were statistically significant based on frequentist models, they were not sufficient to reject the null hypothesis of no difference

**Table 3.** Multilevel model evaluating effects of Group, Cue, and Trial on medium heat pain prior to reversal*.

| | Predictors | Estimates | | | Confidence intervals | | | P-Value / probability of direction | | | Bayesian estimates† | | |
|---|---|---|---|---|---|---|---|---|---|---|---|---|---|
| | | LMER‡ | NLME§ | BRMS† | LMER‡ | NLME§ | BRMS† | LMER‡ | NLME§ | BRMS† | % in ROPE | Rhat | ESS |
| All participants (n=40) | (Intercept) | 3.89 | 3.875 | 3.883 | 3.55–4.23 | [3.53, 4.221] | [3.598, 4.181] | <0.001 | 0.000 | 100.00% | 0 | 1 | 4860.246 |
| | Group | 0.18 | 0.167 | 0.178 | -0.16–0.51 | [-0.183, 0.518] | [-0.106, 0.480] | 0.305 | 0.339 | 83.67% | 45.258 | 1 | 5043.495 |
| | Cue | 1.27 | 1.254 | 1.261 | 0.89–1.66 | [0.857, 1.651] | [0.939, 1.568] | <0.001 | 0.000 | 100.00% | 0 | 1 | 11882.091 |
| | Trial | 0.11 | 0.107 | 0.11 | 0.02–0.21 | [0.011, 0.202] | [0.031, 0.190] | 0.023 | 0.029 | 98.39% | 88.483 | 1 | 13555.299 |
| | Group * Cue | 0.44 | 0.425 | 0.442 | 0.06–0.83 | [0.028, 0.823] | [0.116, 0.758] | 0.024 | 0.036 | 98.45% | 8.892 | 1 | 12094.57 |
| | Group * Trial | 0.14 | 0.133 | 0.135 | 0.04–0.23 | [0.035, 0.231] | [0.049, 0.213] | 0.007 | 0.008 | 99.52% | 75.492 | 1 | 13395.889 |
| | Cue * Trial | 0.14 | 0.132 | 0.142 | -0.01–0.30 | [-0.032, 0.296] | [0.005, 0.274] | 0.071 | 0.114 | 95.07% | 62.758 | 1 | 14333.687 |
| | (Group *Cue) *Trial | 0.2 | 0.186 | 0.201 | 0.04–0.37 | [0.016, 0.357] | [0.060, 0.343] | 0.016 | 0.033 | 98.70% | 35.45 | 1 | 16585.744 |
| Instructed Group (n=20) | (Intercept) | 4.07 | 4.034 | 4.076 | 3.63–4.51 | [3.594, 4.475] | [3.691, 4.455] | <0.001 | 0 | 100.00% | 0 | 1 | 4421.861 |
| | Cue | 1.73 | 1.694 | 1.724 | 1.14–2.32 | [1.088, 2.3] | [1.204, 2.218] | <0.001 | 0 | 100.00% | 0 | 1 | 7914.074 |
| | Trial | 0.26 | 0.252 | 0.26 | 0.08–0.44 | [0.073, 0.43] | [0.106, 0.422] | 0.005 | 0.0063 | 99.28% | 17.733 | 1 | 6993.832 |
| | Cue * Trial | 0.35 | 0.294 | 0.361 | 0.04–0.66 | [-0.031, 0.618] | [0.088, 0.631] | 0.027 | 0.0755 | 97.65% | 13.558 | 1 | 7952.078 |
| Uninstructed Group (n=20) | (Intercept) | 3.74 | 3.737 | 3.741 | 3.22–4.25 | [3.215, 4.259] | [3.261, 4.185] | <0.001 | 0 | 100.00% | 0 | 1 | 3062.443 |
| | Cue | 0.89 | 0.885 | 0.874 | 0.40–1.38 | [0.378, 1.392] | [0.450, 1.262] | <0.001 | 0.0008 | 99.88% | 0.55 | 1.001 | 9645.377 |
| | Trial | 0 | -0.005 | -0.005 | -0.11–0.10 | [-0.113, 0.103] | [-0.095, 0.087] | 0.928 | 0.9333 | 53.46% | 99.433 | 1 | 8985.322 |
| | Cue * Trial | -0.03 | -0.035 | -0.035 | -0.20–0.14 | [-0.211, 0.142] | [-0.184, 0.112] | 0.704 | 0.6975 | 65.33% | 90.7 | 1 | 12054.973 |

*This table presents results of a linear mixed model predicting subjective pain on medium heat trials as a function of Group (Instructed vs Uninstructed), Cue (Original High vs Original Low), and Trial prior to the first reversal, as well as post-hoc tests in each Group. See **Table 1** for additional information about model specification and presentation.

†Estimates based on Bayesian model linear mixed models using the 'brms' function (**Bürkner, 2017**) using the following code: brmPain~Group*Cue*Trial+(1+Cue*Trial|Subject,prior = set_prior(" normal(0,2.5)", class="b"), save_all_pars = TRUE, silent = 0, iter = 4000,, warmup = 1000). Posterior estimates and the Region of Partial Equivalence were obtained using the "describe_posterior" function from the package BayesTestR (**Makowski et al., 2019a**) and interpreted as in **Makowski et al., 2019b**. The Region of Partial Equivalence (ROPE) was defined as [-0.17, 0.17] across all participants, [-172,.172] when restricted to the Instructed Group, and [-.168,.168] when restricted to the Uninstructed Group.

‡Estimates based on a linear mixed effects model implemented in the 'lmer' function of lme4 (**Bates et al., 2015**) using the following code: lmer(Pain_Medium~Group*Cue*Trial+(1+Cue*Trial|Subject)).

§Estimates based on a linear mixed effects model implemented in the 'lme' function of nlme (**Pinheiro et al., 2021**) including autoregression using the following code: lme(Pain~Group *Cue*Trial, random = ~1 + Cue*Trial|Subject, correlation = corAR1, na.action=na.exclude).

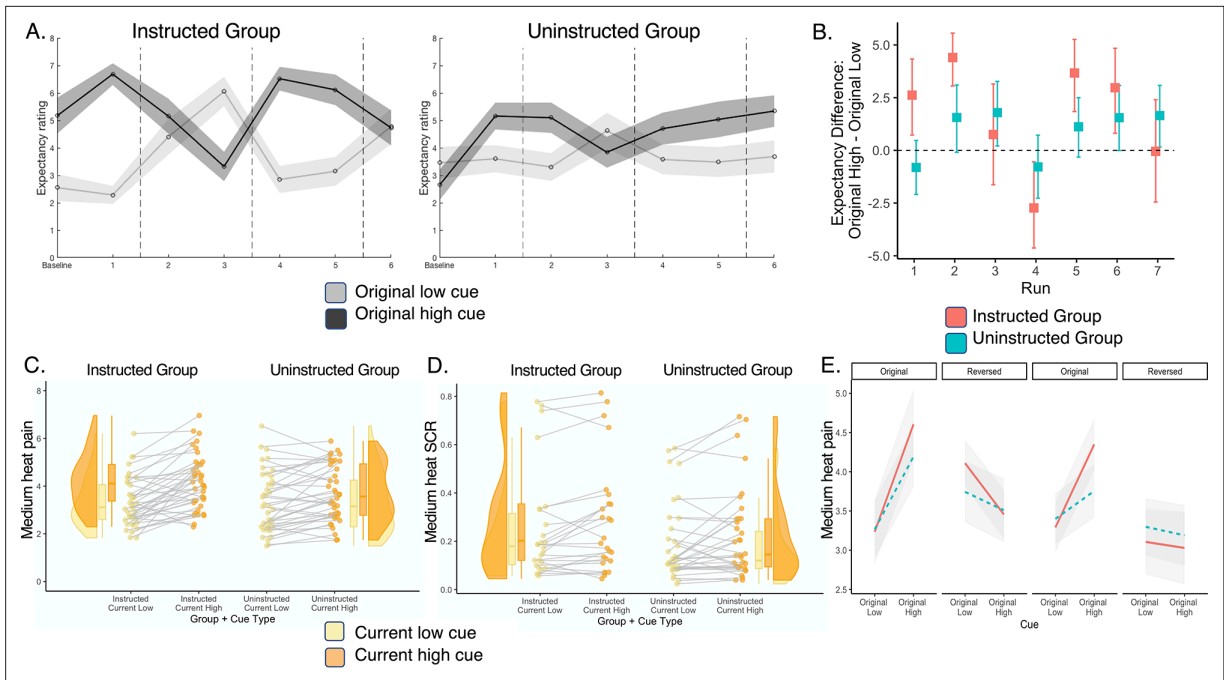

**Figure 4.** Expectations and pain ratings update as contingencies change. We analyzed cue-based expectations and the effects of cues on pain ratings in response to medium heat across the entire task, including reversals. Reversals were coded relative to instructions in the Instructed Group and relative to experience in the Uninstructed Group (see *Figure 1C*). (**A**) Expectancy ratings across the entire task. Both groups updated expectations as contingencies reversed. (**B**) Cue-based differences in expectancy. The Instructed Group (Red) shows larger differences in expectancy as a function of phase, although both groups show significant Cue x Phase interactions across the task, indicating that both instructions and experiential learning dynamically shape expectations. (**C**) Effects of current cue contingencies on subjective pain. We analyzed Cue x Phase interactions on pain to evaluate whether individuals report higher pain with the cue that is currently paired with high heat (Original High Cue on original contingency blocks, Original Low Cue on reversed blocks). Both groups reported higher pain when medium heat was paired with the current high cue relative to the current low cue. (**D**) Effects of current cue contingencies on heat-evoked SCR. Similar to pain, both groups displayed elevated heat-evoked SCR when medium heat was paired with the current high cue relative to the current low cue. (**E**) Pain reversals are larger in Instructed Group participants. As with expectancy ratings, both groups showed significant reversals of cue effects on subjective pain as contingencies changed, but reversals were larger in Instructed Group participants. Individual participants' ratings are presented in *Figure 4—figure supplement 1* and retrospective ratings are reported in *Figure 4—figure supplement 2*. Errors and shaded regions denote standard error of the mean (n = 20 per group).

The online version of this article includes the following source data and figure supplement(s) for figure 4:

**Source data 1.** Cue effects on skin conductance responses.

**Figure supplement 1.** Cue and phase effects for individual participants.

**Figure supplement 2.** Retrospective ratings.

based on Bayesian models (see *Figure 4—source data 1*). Importantly, only 13 Instructed Group participants had variations in heat-evoked SCR on medium heat trials and were included in analyses; we therefore take these results with caution and did not analyze associations between trial-by-trial SCR and brain responses. We also evaluated cue effects on anticipatory SCR, that is responses to the cue in the interval prior to heat stimulation, in exploratory analyses. In contrast to other outcomes, anticipatory arousal was associated with a main effect of Cue and a significant Group x Phase interaction (see *Figure 4—source data 1*), but we did not observe any interactions between Cue and Phase, suggesting that anticipatory responses did not vary as contingencies change. However, analyses were limited to 29 participants and Bayesian analyses indicated that the data support the null hypothesis of no effect, and thus we do not make inference based on anticipatory arousal.

**Table 4.** Multilevel model evaluating effects of Group, Cue, and Phase on medium heat pain across the entire task[*].

| Predictors | Estimates | | | Confidence intervals | | | P-Value / probability of direction | | | Bayesian estimates[†] | | |
|---|---|---|---|---|---|---|---|---|---|---|---|---|
| | LMER[‡] | NLME[§] | BRMS[†] | LMER[‡] | NLME[§] | BRMS[†] | LMER[‡] | NLME[§] | BRMS[†] | % in ROPE | Rhat | ESS |
| **All participants (n=40)** | | | | | | | | | | | | |
| (Intercept) | 3.63 | 3.624 | 3.621 | 3.31–3.95 | [3.301, 3.948] | [ 3.342, 3.889] | <0.001 | 0.000 | 100.00% | 0 | 1.001 | 2237.328 |
| Group | 0.11 | 0.100 | 0.103 | −0.22–0.43 | [−0.233, 0.433] | [−0.162, 0.379] | 0.52 | 0.546 | 73.12% | 62.4 | 1.002 | 2181.022 |
| *Cue* | 0.29 | 0.286 | 0.287 | 0.14–0.44 | [0.146, 0.426] | [ 0.160, 0.413] | <0.001 | 0.000 | 100.00% | 8.425 | 1 | 17294.743 |
| *Phase* | 0.1 | 0.100 | 0.104 | 0.00–0.20 | [0.001, 0.198] | [ 0.015, 0.183] | 0.046 | 0.047 | 97.38% | 91.908 | 1 | 10287.406 |
| Group * Cue | 0.06 | 0.095 | 0.065 | −0.09–0.21 | [−0.045, 0.235] | [−0.062,, 0.189] | 0.401 | 0.182 | 79.30% | 92.3 | 1 | 17827.085 |
| Group * Phase | −0.01 | −0.002 | −0.005 | −0.11–0.09 | [−0.1, 0.097] | [−0.087, 0.079] | 0.907 | 0.974 | 54.11% | 99.925 | 1 | 10239.068 |
| Cue * Phase | 0.58 | 0.643 | 0.582 | 0.38–0.78 | [0.443, 0.843] | [ 0.415, 0.740] | <0.001 | 0.000 | 100.00% | 0.042 | 1 | 11512.539 |
| (*Group * Cue*) * Phase | 0.24 | 0.248 | 0.241 | 0.04–0.44 | [0.048, 0.447] | [ 0.077, 0.400] | 0.018 | 0.015 | 98.97% | 25.2 | 1 | 10802.199 |
| **Instructed Group (n=20)** | | | | | | | | | | | | |
| (Intercept) | 3.74 | 3.732 | 3.731 | 3.32–4.16 | [3.313, 4.15] | [ 3.363, 4.107] | <0.001 | 0.000 | 100.00% | 0 | 1.001 | 1614.785 |
| *Cue* | 0.36 | 0.386 | 0.353 | 0.14–0.58 | [0.182, 0.59] | [ 0.156, 0.530] | 0.001 | 0.000 | 99.83% | 5.775 | 1 | 13912.092 |
| Phase | 0.1 | 0.100 | 0.098 | −0.02–0.22 | [−0.03, 0.23] | [−0.003, 0.194] | 0.105 | 0.133 | 94.46% | 88.158 | 1 | 12203.321 |
| Cue * Phase | 0.84 | 0.904 | 0.836 | 0.52–1.17 | [0.58, 1.227] | [ 0.559, 1.112] | <0.001 | 0.000 | 99.99% | 0.05 | 1 | 9283.385 |
| **Uninstructed Group (n=20)** | | | | | | | | | | | | |
| (Intercept) | 3.53 | 3.531 | 3.526 | 3.04–4.02 | [3.04, 4.022] | [ 3.098, 3.955] | <0.001 | 0.000 | 100.00% | 0 | 1.001 | 2390.769 |
| *Cue* | 0.23 | 0.194 | 0.228 | 0.01–0.44 | [−0.004, 0.391] | [ 0.058, 0.411] | 0.037 | 0.054 | 98.09% | 33.975 | 1 | 19966.344 |
| Phase | 0.11 | 0.099 | 0.107 | −0.05–0.27 | [−0.05, 0.248] | [−0.025, 0.245] | 0.19 | 0.192 | 90.18% | 81.167 | 1 | 9498.139 |
| *Cue * Phase* | 0.35 | 0.411 | 0.354 | 0.11–0.60 | [0.169, 0.653] | [ 0.158, 0.555] | 0.004 | 0.001 | 99.58% | 8.133 | 1 | 12670.552 |

[*]This table presents results of a linear mixed model predicting subjective pain on medium heat trials as a function of Group (Instructed vs Uninstructed), Cue (Original High vs Original Low), and Phase (Original vs Reversed) across all participants, as well as post-hoc tests in each Group. See **Table 1** for additional information about model specification and presentation.

[†]Estimates based on Bayesian model linear mixed models using the 'brms' function (**Bürkner, 2017**) using the following code: brmPain~Group *Cue*Phase+(1+Cue*Phase|Subject,prior = set_prior("normal(0,2.5)", class="b"), save_all_pars = TRUE, silent = TRUE, refresh = 0, iter = 4000,, warmup = 1000). Posterior estimates and the Region of Partial Equivalence were obtained using the "describe_posterior" function from the package BayesTestR (**Makowski et al., 2019a**) and interpreted as in **Makowski et al., 2019b**. The Region of Partial Equivalence (ROPE) was defined as [−0.176, 0.176] across all participants, [−.170,.170] when restricted to the Instructed Group, and [−.181,.181] when restricted to the Uninstructed Group.

[‡]Estimates based on a linear mixed effects model implemented in the 'lmer' function of lme4 (**Bates et al., 2015**) using the following code: lmer(Pain$_{Medium}$~Group*Cue*Phase+(1+Cue*Phase|Subject)).

[§]Estimates based on a linear mixed effects model implemented in the 'lme' function of nlme (**Pinheiro et al., 2021**) including autoregression using the following code: lme(Pain~Group *Cue*Phase, random = ~1 + Cue*Phase|Subject, correlation = corAR1(), na.action=na.exclude).

## Cue effects on heat-evoked responses in pain-related regions reverse as contingencies change, irrespective of instruction, and prefrontal regions mediate cue effects on subjective pain

Behavioral analyses indicated that predictive cues modulated expectations and subjective pain, and that cue effects on both outcomes updated as contingencies reversed. We next asked which brain regions mediated these effects. We were most interested in current cue effects on brain responses to medium heat, that is the Cue x Phase interaction (see Materials and methods and **Figure 5A**).

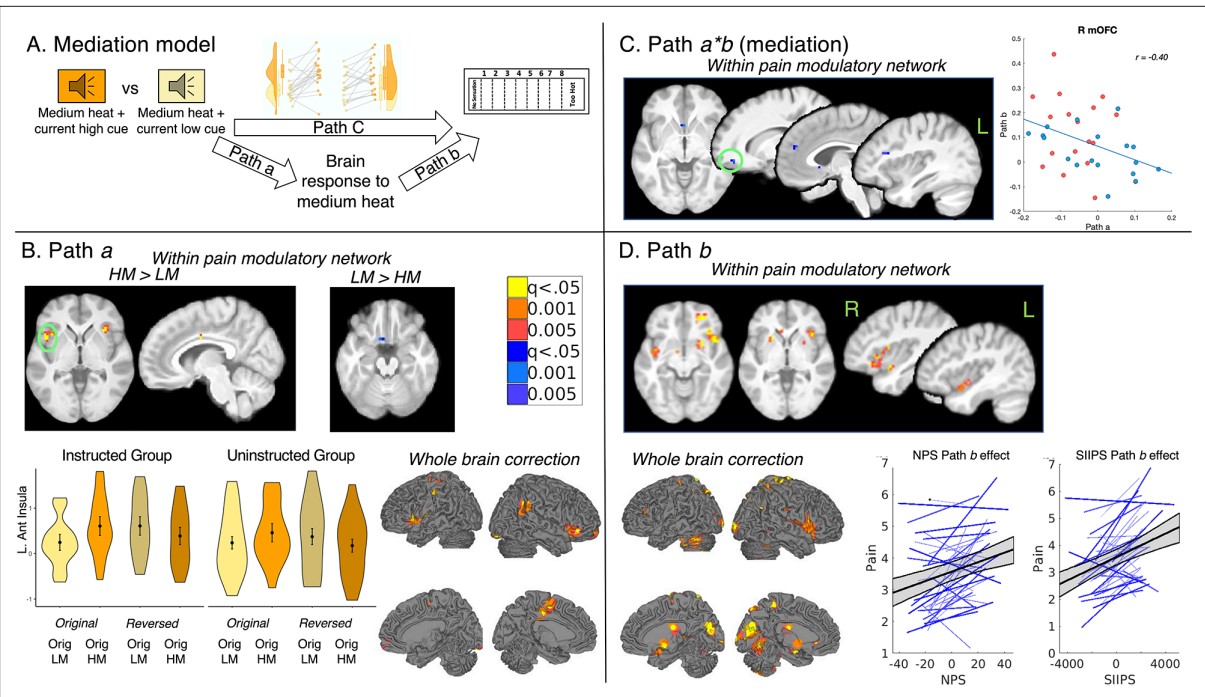

**Figure 5.** Mediation of current cue effects on medium heat pain. We examined brain mediators of current contingency effects on perceived pain on medium heat trials. Results are FDR-corrected within pain modulatory regions and across the whole brain. (**A**) Mediation model. We tested for brain regions that mediate the effects of current cue contingencies on subjective pain, corresponding to the reversals we observed (*Figure 4*). (**B**) Path A: effects of current contingencies. Path a identifies brain regions that show greater activation with the current high pain cue (e.g. Original High Cue during original contingencies, Original Low Cue during reversed contingencies), relative to the current low pain cue. Within pain modulatory regions (see *Figure 5—figure supplement 1*), we observed positive Path a effects (HM >LM) in the bilateral anterior insula, dorsal anterior cingulate, right anterior prefrontal cortex, and left dorsal posterior insula, and negative Path a effects (LM >HM) in the left subgenual ACC. We extracted trial-by-trial estimates from the left anterior insula and visualized average responses as a function of Group, Cue, and Phase (bottom left; *Figure 2*). Both groups showed greater left insula activation when medium heat was preceded by the Current High Cue, and cue effects did not differ by group. Differences were confirmed with extracted average timecourses (see *Figure 5—figure supplement 3*). Whole brain FDR-correction (bottom right) additionally identified positive Path a effects in the M1, S1, and right inferior parietal lobule (see *Figure 5—figure supplement 4*). (**C**) Mediation of current cue effects on pain. We observed significant negative mediation by several pain modulatory regions, including (from left to right) the subgenual ACC, the right VMPFC/OFC, the dorsomedial prefrontal cortex, and the left DLPFC. Extracting responses from each of these regions indicated that individuals who showed larger cue effects (i.e. Path a effects) showed more negative associations between brain activation and subjective pain (i.e. Path b). This is consistent with the fact that mediation can be driven by the covariance between paths, and is consistent with a modulatory suppression effect. Representative correlations are depicted for the right OFC; r=–0.40, P<.001, with Instructed Group participants in red and Uninstructed Group in blue. (**D**) Path b: associations with pain controlling for cue. Path b regions are positively associated with pain, controlling for cue (and temperature, since we tested only medium heat trials). We observed positive Path b effects in the VLPFC, bilateral putamen, bilateral anterior insula, and other regions within the pain modulatory network (top). We also observed significant Path b effects on NPS and SIIPS expression. Spaghetti plots in the lower right illustrate associations between trial-level pattern expression and pain, controlling for cue, for all individuals (blue) and the entire sample (95% CI illustrated in gray). For additional regions identified in whole brain search and uncorrected results, see *Figures 3–5* and *Figure 5—source data 1* (whole-brain corrected results) and *Figure 5—source data 2* (uncorrected results).

The online version of this article includes the following source data and figure supplement(s) for figure 5:

**Source data 1.** Mediation of current cue contingencies: Small-volumes corrected results.

**Source data 2.** Mediation of current cue contingencies: Uncorrected results.

**Figure supplement 1.** A priori regions of interest.

**Figure supplement 2.** Current contingency mediation: Effects of cue and phase within pain modulatory network.

**Figure supplement 3.** Raw timecourses in Path A regions.

**Figure supplement 4.** Current contingency mediation: Whole-brain FDR correction.

**Figure supplement 5.** Raw timecourses in value-related regions of interest.

**Figure supplement 6.** Current contingency mediation: Whole-brain uncorrected.

**Figure supplement 7.** Current contingency mediation including Group as a moderator: Whole-brain uncorrected.

Path *a* identified regions that showed stronger activation in response to medium heat following current high pain cues relative to current low pain cues. Within pain modulatory regions, we observed significant positive Path *a* effects (current high cue >current low cue) in the bilateral anterior insula, left dorsal posterior insula, dACC, and right anterior prefrontal cortex, and negative associations in the left subgenual ACC (sgACC; see *Figure 5B* and *Figure 5—source data 1*). Extracting trial-level responses confirmed that regions with positive Path *a* activation showed greater activation when medium heat was preceded by the initial high pain cue relative to the initial low pain cue during the original contingencies, whereas they showed greater activation when medium heat was paired with the initial low pain cue when contingencies were reversed, and these reversals were observed for both groups (see *Figure 5B* and *Figure 5—figure supplements 2 and 3*). Whole brain FDR correction additionally indicated positive Path *a* effects in left M1, S1, and right inferior parietal lobule (see *Figure 5B*, *Figure 5—figure supplement 4*, and *Figure 5—source data 1*). We observed marginal Path *a* effects on the bilateral striatum (see *Table 2* and *Figure 5—figure supplement 5*); no other ROIs were modulated by current cue contingencies and there were no effects of current cues on the NPS or SIIPS (see *Table 2*).

Path *b* identified voxels that were associated with subjective pain while controlling for cue (see Materials and Methods). We observed positive Path *b* effects within pain-related regions including bilateral anterior insula, pregenual ACC, bilateral putamen, bilateral amygdala, left thalamus, right ventrolateral prefrontal cortex (VLPFC), and right middle insula (see *Figure 5D* and *Figure 5—source data 1*). Whole brain FDR-correction also revealed positive Path *b* effects in the bilateral cerebellum, right S1, bilateral superior parietal lobule, and other regions (see *Figure 5D*, *Figure 5—figure supplement 4*, and *Figure 5—source data 1*). No negative Path *b* effects survived correction within the pain modulatory mask or whole brain search. We observed significant Path *b* effects on responses to medium heat in the bilateral striatum, as well as both signature patterns (see *Figure 5D* and *Table 2*).

Finally, we tested for voxelwise mediation of current cue effects on pain. Within regions previously implicated in studies of pain and placebo, we observed significant negative mediation by the right VMPFC/OFC, left dorsolateral prefrontal cortex (DLPFC), the dorsomedial prefrontal cortex (DMPFC), and the sgACC (see *Figure 5C* and *Figure 5—source data 1*). Whole brain FDR correction additionally identified negative mediation in the left inferior parietal lobule (IPL; see *Figure 5—figure supplement 4*). There were no positive mediators of cue effects in pain in voxel-wise analyses. Negative mediation is consistent with suppression; indeed, extracting responses within mediators indicated that individuals with the strongest positive Path *a* effects (i.e. HM >LM) in these regions showed large negative associations between activation and pain (see *Figure 5C*). This suppression may be consistent with down-regulation in modulatory regions. We did not observe mediation by any value-related ROI or signature pattern (see *Table 2*). See *Figure 5—figure supplement 6* and *Figure 5—source data 2* for whole-brain uncorrected results.

Notably, we did not observe significant moderation by Group in any of the paths at FDR-corrected thresholds or in any of our a priori regions of interest (i.e. correction within pain modulatory regions or whole brain, ROI-wise analyses, or pain signature patterns). This suggests that the dynamic effects of predictive cues and reversals on pain-related brain responses are similar whether individuals learn through instruction or experience, despite stronger influences of cues on subjective pain within the Instructed Group. Uncorrected results, which do point to potential group differences in Path *a* effects in the rostral ACC, left hippocampus, and left thalamus, are presented in *Figure 5—figure supplement 7* and *Figure 5—source data 2*.

## Responses in brainstem, orbitofrontal cortex, and right prefrontal cortex maintain initial contingencies despite reversals, particularly in uninstructed participants

While the main mediation analysis isolated brain regions whose responses to cues on medium heat trials updated upon reversal, some regions may show sustained responses to initial contingencies. We therefore conducted a second mediation analysis to identify regions that responded to original contingencies and did not reverse as contingencies changed. We were most interested in Path *a*, which identified regions that showed stronger activation in response to cues that were originally paired with high pain relative to cues that were originally paired with low pain (see *Figure 6*), while controlling for current contingencies. No regions survived FDR correction within pain modulatory

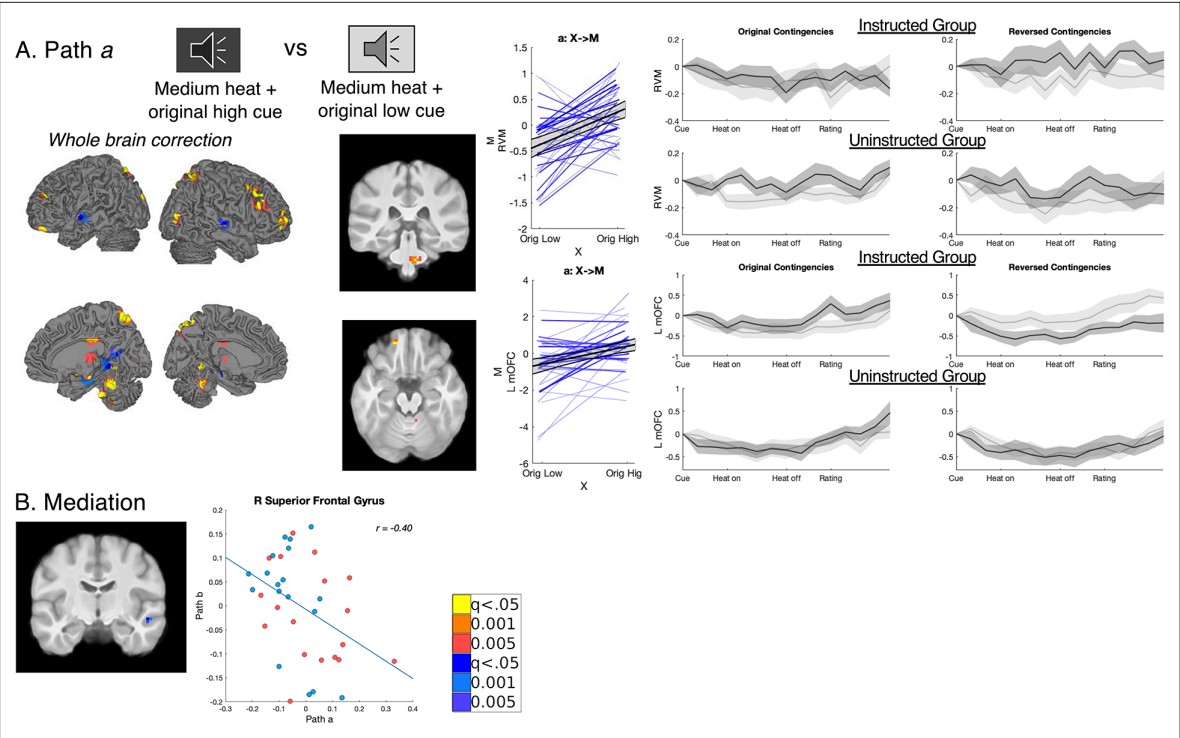

**Figure 6.** Original cue effects on medium heat pain. We conducted a second mediation analysis to isolate effects of original contingencies, controlling for current contingencies. (**A**) Effects of original contingencies. The goal of our second mediation analysis was to specifically identify regions that continued to respond to the original contingencies across the entire task, regardless of reversals. (**B**) Path a: Regions that show greater activation to original high pain contingencies despite reversals. Path *a* identified regions that showed greater activation to the Original High Cue (dark gray) relative to the Original Low Cue (light gray) across the entire task, while controlling for current contingencies. No voxels survived correction within pain modulatory regions. However, whole brain correction revealed that a number of regions including the brainstem's rostroventral medulla (RVM), right DLPFC, left medial OFC (mOFC), and other regions (see *Figure 6—figure supplement 2* and *Figure 6—source data 1*) continued to show higher activation when medium heat was paired with the original high pain cue regardless of Phase. Extracting trial-by-trial responses from the RVM (top) confirmed that this region showed greater heat-evoked activation with the Original High Cue during both original and reversed contingencies and that effects were present in both the Instructed Group and the Uninstructed Group. In the mOFC, however, responses did reverse within the Instructed Group (bottom), suggesting that failure to reverse was driven by Uninstructed Group participants. Similar effects were observed in the VMPFC region of interest (See *Figure 5—figure supplement 5*). See *Figure 6—figure supplements 2 and 3* for means within other Path *a* regions. (**B**) Associations between original contingencies and pain were statistically mediated by a cluster in the right superior frontal gyrus, in which individuals who had larger effects of original cues on brain responses (controlling for current contingencies; i.e. Path a, x-axis) also had stronger negative associations between brain activation and subjective pain (Path b, y-axis). There were no additional mediators of original cue effects on pain based on whole brain correction identified additional effects in the right DLPFC, precuneus, and cerebellum (see *Figure 6—figure supplement 1* and *Figure 6—source data 1*). Whole brain uncorrected results are presented in *Figure 6—figure supplements 4 and 5* and *Figure 6—source data 2*.

The online version of this article includes the following source data and figure supplement(s) for figure 6:

**Source data 1.** Mediation of original cue contingencies: Small-volumes corrected results.

**Source data 2.** Mediation of original cue contingencies: Uncorrected results.

**Figure supplement 1.** Original contingency mediation: Whole-brain FDR correction.

**Figure supplement 2.** Original contingency mediation: Effects of cue and phase.

**Figure supplement 3.** Raw timecourses in Path A regions.

**Figure supplement 4.** Original contingency mediation: Whole-brain uncorrected results.

**Figure supplement 5.** Original contingency mediation, including Group as a moderator: Whole-brain uncorrected results.

regions, and there were no effects of original cues on the NPS or SIIPS (all p's>0.2). Whole brain correction revealed significant positive Path *a* effects (original high cue >original low cue) in a brainstem cluster overlapping with the rostroventral medulla (RVM; consistent with the pontine reticular nucleus based on the Brainstem Navigator; *Singh et al., 2021*), as well as the left medial OFC (area Fo3), right lateral prefrontal cortex, right DLPFC, medial cerebellum, and right occipital cortex (see

*Figure 6B*, *Figure 6—figure supplement 1*, and *Figure 6—source data 1*) there were no negative Path *a* effects. Extracting trial-level responses from Path *a* regions (see *Figure 6B* and *Figure 6— figure supplements 2 and 3*) indicated that effects in most regions were driven primarily by lack of reversal in the Uninstructed Group, although we did not observe significant moderation by Group in any regions at FDR-corrected thresholds (see *Figure 6—figure supplement 5* and *Figure 6—source data 1*), and that only the RVM maintained original contingencies in both groups. ROI-wise analyses within value-related regions indicated the VMPFC was significantly modulated by initial contingencies (see *Table 2*) driven by greater activation in responses to the original low pain cue. Extracting timecourses suggests that these differences were driven by the Uninstructed Group (see *Figure 5— figure supplement 5*) and an adjacent region of VMPFC showed significant moderation by Group in uncorrected voxelwise analyses (see *Figure 6—figure supplement 4* and *Figure 6—source data 2*), although we did not observe significant group differences in ROI-wise analyses when we included Group as a potential moderator (p>0.6), and Path *a* effects remained significant when controlling for Group (*a*=–0.09, p=0.019).

Path *b* effects were similar to those observed when controlling for current cues; see *Figure 6— figure supplement 1*, *Figure 6—source data 1*, and *Table 2*. Whole brain correction revealed significant negative mediation of original cue effects on pain in the right superior temporal gyrus (see *Figure 6B*, *Figure 6—figure supplement 1*, and *Figure 6—source data 1*). Extracting responses from this region indicated that, similar to mediators of current cues on pain, mediation was driven by the covariance between Paths *a* and *b*, such that individuals who showed stronger original cue effects on right superior temporal gyrus responses to heat also showed stronger negative associations between activation and subjective pain (see *Figure 6B* and *Figure 6—figure supplement 2*). Additional regions identified in uncorrected voxelwise analyses are reported in *Figure 6—figure supplement 3* and *Figure 6—source data 2*. Finally, consistent with mediation of current contingencies, Group did not moderate Path *b* or mediation effects even at uncorrected thresholds, indicating associations between brain activation and pain were similar regardless of instruction (see *Figure 6B* and *Figure 6—figure supplement 4*).

## Quantitative models reveal that instructed participants reverse expectations upon instructions and learning is faster in uninstructed participants

We observed no group differences in the effects of cues and reversals on brain responses to noxious stimuli in pain-related regions, suggesting that pain-related responses are mediated similarly whether or not participants are instructed about contingencies. However, we did observe possible group differences in the VMPFC and other regions that maintained original contingencies in the Uninstructed Group, but not the Instructed Group, although group differences were only evident in uncorrected voxelwise analyses. Our mediation models and behavioral analyses that include effects of Phase assume that expectations and responses update completely upon reversal, either through instruction in the Instructed Group or when contingencies reverse in the Uninstructed Group. However, these models may not capture differences if dynamic learning proceeds more gradually (i.e. continuously as a function of pairings between cues and temperatures), and it is possible that groups differ in the dynamics of learning and associations between learning and brain activation, consistent with previous work (*Atlas et al., 2016*).

To formally examine these dynamics, we applied a quantitative model of instructed reversal learning (*Atlas et al., 2016*; *Atlas and Phelps, 2018*) which accounts for how expectations update dynamically as a function of both experience and instruction. The model includes two parameters: $\alpha$, a standard learning rate that captures the extent to which expected value (EV) updates in response to prediction errors, and $\rho$, which guides whether and how EV reverses upon instruction (see Materials and methods). Here, we extended this model to predict subjective pain on medium heat trials. This model accounted for variations in pain reports better than other plausible models, including a standard Rescorla-Wagner model without the $\rho$ parameter and a hybrid model of adaptive learning modified to reverse upon instruction (*Atlas et al., 2019*).

Consistent with our task manipulation, instructed reversal parameters (i.e. $\rho$) varied as a function of Group (fit to individuals: t(38) = 3.013, p=0.005; see *Figure 7A*), such that participants in the Instructed Group showed larger reversals at the time of verbal instruction (fit to individuals: Instructed:

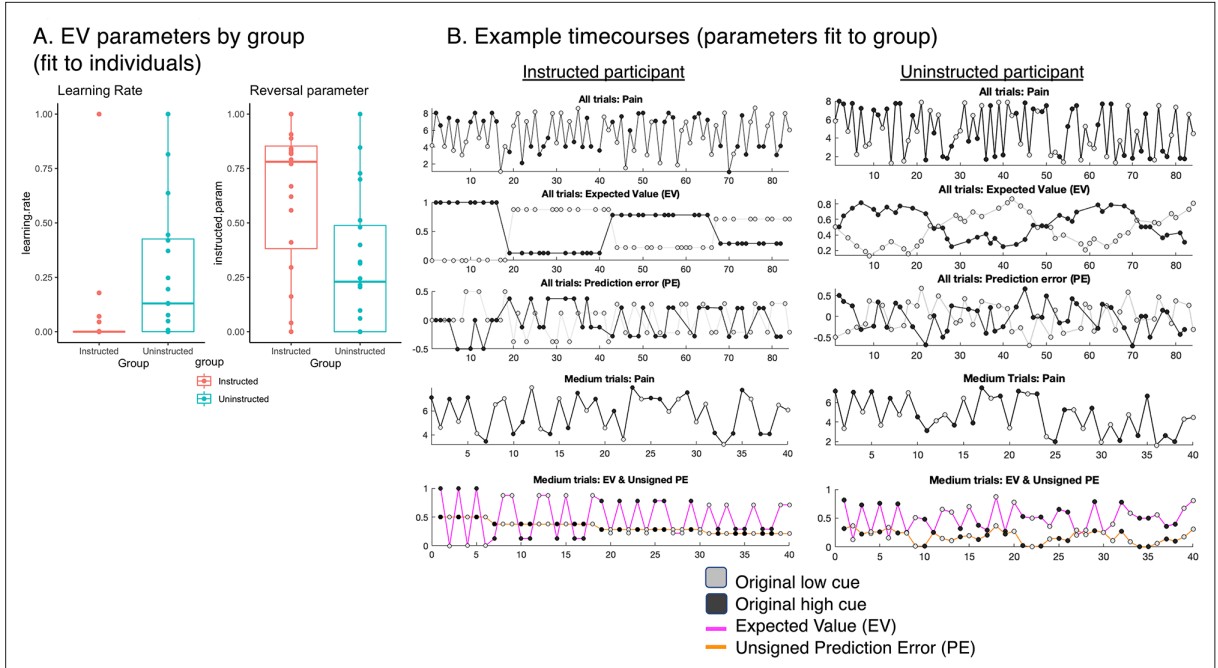

**Figure 7.** Instructed learning model fit to pain on medium heat trials. We fit a computational model of instructed reversal learning (*Atlas et al., 2016*) to pain reports on medium heat trials to isolate the dynamics of expected value and how expected value updates with instruction. (**A**) *Group differences in learning parameters*. Fitting models to individuals revealed group differences in learning rate (α, left), such that participants in the Uninstructed Group (blue) showed stronger updates of expected value in response to prediction errors relative to the Instructed Group (red), whereas the Instructed Group showed stronger reversals at the time when instructions were delivered, based on the instructed reversal parameter ($\rho$, right). (**B**) *Predicted timecourse of expected value based on jack-knife model fits*. We used model parameters from a jack-knife model fitting procedure (see Materials and Methods) to generate predicted timecourse of expected value (EV) for each group. Here we depict model predictions for an example participant in the Instructed Group (left) and the Uninstructed Group (right). As shown in the second row, EV reverses immediately upon instruction in the Instructed Group and reverses more gradually in the Uninstructed Group. We focused on responses fit to medium pain ratings (fourth row) and modeled associations between heat-evoked activation and the timecourse of EV and unsigned PE on medium heat trials (bottom row).

*M*=0.62, SD = 0.35; Uninstructed: *M*=0.31, SD = 0.31). This confirms our task manipulation (instructed reversals should only be seen in the group that was exposed to instructions) and validates the model's application to subjective pain. Consistent with our previous work on instructed threat learning (*Atlas et al., 2016*), learning rates (i.e. α) were close to zero in the Instructed Group (*M*=0.065, SD = 0.22), indicating that there was little additional learning as a function of experience between instructed reversals, as might be expected given that feedback was entirely consistent with instructions. Learning rates were indeed higher in the Uninstructed Group (*M*=0.28, SD = 0.34), and differed significantly between groups (fit to individuals: t(38) = –2.32, p=0.026). Differences in $\rho$ and α were observed when models were fit to individuals, and when they were fit across the group using a jack-knife model fitting procedure. Jack-knife estimates revealed a significant group difference in the learning rate α (t(1,38) = 33.07, p<0.001, CI = [0.69, 0.78]), driven by higher α values in the Uninstructed Group (*M*=0.264, *SE = 0.028*) relative to the Instructed Group (*M*=0, SE = 0), and a significant group difference in the instructed reversal parameter $\rho$ (t(1,38) = –9.41, p<0.001, CI = [–0.32, –0.21]) driven by higher $\rho$ parameters in the Instructed Group (*M*=0.875, SE = 0.003), relative to the Uninstructed Group (*M*=0.139, *SE = 0.022*). Thus expected value updates primarily upon instruction in the Instructed Group with very little additional learning between reversals (consistent with the Cue x Phase interactions we modeled behaviorally and that formed the foundation of our mediation models), whereas individuals in the Uninstructed Group update expected value over time as a function of experience, that is pairings between cues and heat, as depicted in *Figure 7B*.

## Expected value dynamically modulates responses to noxious stimulation, with differences between groups in the rostral anterior cortex

We next searched for neural correlates of dynamic expected value (EV) signals on medium heat trials. We used the learning time-course generated from fits to pain ratings in each group and searched for regions that correlated with EV. *Figure 7B* depicts example EV timecourses using the same parameters that were used to evaluate associations between EV and medium heat-evoked brain responses in each group. We used robust regression to evaluate associations within each group, and those that were consistent while controlling for Group, or differed significantly between Groups.

Whole brain correction revealed significant positive associations with instruction-based EV in the right MPFC in the Instructed Group (see *Figure 8—figure supplement 1*, and *Figure 8—source data 1*) and negative associations with feedback-driven EV in the left rACC in the Uninstructed Group (see *Figure 8—figure supplement 1*, and *Figure 8—source data 1*). Consistent with this, FDR correction within a priori pain modulatory regions revealed positive group differences (Instructed >Uninstructed) in the rACC, as well as the left anterior insula, left dorsal posterior insula, and the bilateral thalamus (see *Figure 8A* and *Figure 8—source data 1*). Extracting responses from these regions revealed that there were positive associations with EV in the Instructed Group, and negative associations in the Uninstructed Group. Whole brain correction additionally revealed positive differences in the DMPFC and left middle cingulate (see *Figure 8A*, *Figure 8—figure supplements 1 and 2*, and *Figure 8—source data 1*). ROI-wise tests within a priori value related regions indicated that groups differed in the left striatum, driven by positive associations in the Instructed Group (see *Table 2* and *Figure 8B*). Robust regression did not identify any significant associations across all participants (i.e. main effect, controlling for group) between EV and responses to medium heat based on corrected voxelwise analyses or in regions of interest or signature patterns. However, uncorrected results indicate negative associations in the left lateral OFC, hippocampus, and other regions that have been implicated in prior studies (see *Figure 8—figure supplement 3* and *Figure 8—source data 2*). Fitting models to heat-evoked autonomic responses revealed similar patterns of activation based on whole-brain and ROI-based correction (see *Figure 8—figure supplement 4* and *Figure 8—source data 3*), and while group differences did not survive correction when we fit models to anticipatory SCR (see *Figure 8—figure supplement 5* and *Figure 8—source data 4*), we observed group differences in associations with EV in overlapping portions of the VMPFC and left putamen at uncorrected thresholds in all three models (see *Figure 8—figure supplement 2* and *Figure 8—source data 2–4*). For complete results of models fit to SCR, see *Figure 8—figure supplements 4–5* and *Figure 8—source data 3 and 4*.

We searched for correlates of instructed and feedback-driven EV signals within Instructed Group participants to test whether brain responses were preferentially related to instructed or feedback-driven learning within participants exposed to both types of information. Controlling for uninstructed EV, instructed EV was positively associated with activation near the left nucleus accumbens based on correction within pain-related regions (see *Figure 8—source data 5*), and ROI-wise analyses revealed significant associations with instructed EV bilaterally in the striatum (see *Table 2*). Whole brain correction additionally identified positive associations in the left anterior insula, left rACC, and right DMPFC (see *Figure 8—figure supplement 6* and *Figure 8—source data 5*). There were no regions that were preferentially associated with uninstructed EV within the Instructed Group in corrected voxelwise search or ROI-wise analyses. Finally, whole brain correction revealed significant differences between instructed and uninstructed EV in the bilateral middle cingulate (see *Figure 8C*, *Figure 8—figure supplement 6* and *Figure 8—source data 5*). Uncorrected voxelwise results are reported in *Figure 8—figure supplement 7* and *Figure 8—source data 5*.

## Associations with unsigned prediction error differ between groups

In addition to analyses of EV, we evaluated associations between brain responses to medium heat and unsigned prediction errors (PEs). No regions showed significant associations within pain related ROIs; however, whole brain correction revealed a positive association with PE, controlling for group, in a wide swath of contiguous activation encompassing the right anterior insula, right striatum, and right amygdala (see *Figure 8D*). Whole brain correction also revealed a significant difference between groups in a large contiguous cluster that included the right SII, right superior temporal gyrus, and right temporo-parietal cortex (see *Figure 8E*) driven by negative associations with PE in the Instructed

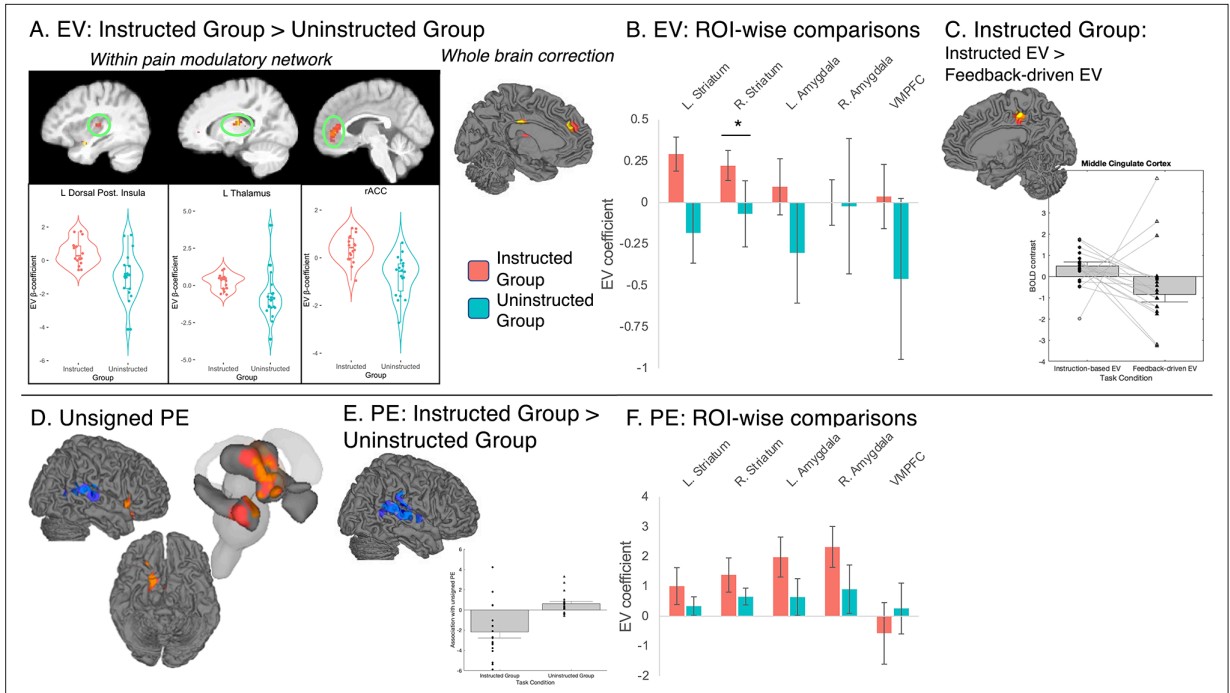

**Figure 8.** Group differences in associations with expected value and prediction error on medium heat trials. We used the timecourse of expected value (EV) based on fitting computational models to pain reports from each group (see **Figure 7**) to isolate the neural correlates of instructed and uninstructed expected value (EV) and prediction error (PE) during pain processing. We examined associations between brain responses to medium heat and the timecourse of EV and unsigned PE (i.e. absolute value of PE) and used robust regression (**Wager et al., 2005**) to compare groups. (**A**) Group differences in expected value within pain modulatory regions. The left dorsal posterior insula, left thalamus, and rostral anterior cingulate cortex (rACC) showed positive associations with EV within the Instructed Group (red) and negative associations within the Uninstructed Group (blue). Whole brain FDR-correction additionally identified group differences in the left middle cingulate cortex (see surface map). (**B**) Associations with EV in value-related ROIs. Extracting contrast values within a priori value-related regions of interest (ROIs) revealed significant associations in the bilateral striatum within the Instructed Group and significant group differences in the left striatum (see **Table 4**). (**C**) Comparing instructed and feedback-driven EV within the Instructed Group. Direct comparisons of the timecourse of EV within Instructed Group participants who were exposed to both types of information revealed significant differences in the middle cingulate cortex, driven by positive associations with instruction-based EV. (**D**) Main effects of unsigned PE. Unsigned PE was associated with activation across groups in the right insula, striatum, and right amygdala. (**E**) Group differences in unsigned PE. There were stronger associations with unsigned PE in the right SII and temporal gyrus, driven by negative associations in the Instructed Group and positive associations in the Uninstructed Group. (**F**) Associations with PE in value-related ROIs. ROI-wise analyses revealed significant associations between unsigned PE and activation in the right striatum and bilateral amygdala. See also **Figure 8—figure supplements 1–9** and **Figure 8—source data 1 and 2**. Error bars denote standard error of the mean (n = 18 per group).

The online version of this article includes the following source data and figure supplement(s) for figure 8:

**Source data 1.** Associations with expected value (EV): Small-volumes corrected results.

**Source data 2.** Associations with expected value (EV): Uncorrected results.

**Source data 3.** Associations with expected value (EV) based on fits to heat-evoked SCR.

**Source data 4.** Associations with expected value (EV) based on fits to cue-evoked SCR.

**Source data 5.** Comparing instructed and feedback-driven expected value (EV) within the Instructed Group.

**Source data 6.** Associations with unsigned prediction error (PE).

**Figure supplement 1.** Expected value: Whole-brain correction.

**Figure supplement 2.** Expected Value on medium heat trials: Moderation by group (Instructed >Uninstructed).

**Figure supplement 3.** Expected value based on fit to pain ratings: Whole-brain uncorrected results.

**Figure supplement 4.** Expected value based on fit to heat-evoked SCR: Whole-brain uncorrected results.

**Figure supplement 5.** Expected value based on fit to anticipatory SCR: Whole-brain uncorrected results.

**Figure supplement 6.** Instructed vs Feedback driven Expected Value within Instructed Group: Whole-brain corrected results.

**Figure supplement 7.** Instructed vs feedback driven expected value within instructed group: whole-brain uncorrected results.

**Figure supplement 8.** Absolute value of prediction error on medium trials: whole-brain corrected results.

**Figure supplement 9.** Absolute value of prediction error on medium trials: whole-brain uncorrected results.

Group and positive associations in the Uninstructed Group (see *Figure 8—figure supplement 8* and *Figure 8—source data 6*). Within value-related ROIs, unsigned PEs were positively associated with responses to heat in the right striatum and the bilateral amygdala (see *Figure 8F* and *Table 2*). Associations with right striatum were observed within each group, whereas associations with amygdala were only observed in the Instructed Group; however, group differences were not significant in any region. Finally, there was no association between PE and NPS or SIIPS expression in either group (see *Table 2*). Voxelwise uncorrected results are reported in *Figure 8—figure supplement 9* and *Figure 8—source data 6*.

## Post-task ratings

We used ANOVAs to evaluate effects of Group and Cue on post-task affect ratings, and to measure effects of Group, Cue, and Phase for retrospective expectancy ratings (i.e. retrospective ratings of expected pain at the beginning and end of the task as a function of Cue). There were no differences in reported affect as a function of Group or Cue (all p's>0.09; see *Figure 4—figure supplement 2*). We observed a significant Group x Phase interaction on retrospective expectancy (F(1,36) = 4.386, p=0.043) and a marginal Group x Cue x Phase interaction (F(1,36) = 3.635, p=0.065). Post-hoc pairwise comparisons indicated that the Instructed Group reported differences in expected pain as a function of Cue at the beginning of the task ($p_{adjusted}$ = 0.041), but not the end of the task (p>0.9), whereas the Uninstructed Group did not report significant differences at any point (all p's>0.1).

## Discussion

We measured whether cue-based expectancy effects on pain and brain responses to noxious heat update dynamically as contingencies change, and whether these relationships vary as a function of whether individuals learn through instruction or experience. All participants demonstrated robust cue-based expectancy effects on pain, consistent with previous work from our group and others (*Colloca et al., 2008a*; *Atlas et al., 2010*; *Wiech et al., 2014*; *Fazeli and Büchel, 2018*; *Jepma et al., 2018*; *Michalska et al., 2018*; *Koban et al., 2019*; *Abend et al., 2021*). Here, we provide new evidence that predictive cues shape pain, autonomic responses, and heat-evoked brain activation even when contingencies change repeatedly, whether reversals are accompanied by instructions or learned through experience. Reinforcement learning models indicated that these effects emerge dynamically, consistent with error-driven learning. We observed dissociations in the associations between expected value and brain responses to heat in several brain regions, including the rostral anterior cingulate cortex (rACC), which was positively associated with expected value in the Instructed Group and negatively associated in the Uninstructed Group. Finally, cue effects on several pain-related regions including the bilateral anterior insula and dorsal anterior cingulate updated dynamically as contingencies changed regardless of group, whereas the RVM responded to original contingencies throughout the task. Dynamic cue effects on pain were mediated by modulatory frontal regions implicated in placebo analgesia, including the VMPFC/OFC and DLPFC. Here we discuss these findings and their implications for future work and our understanding of pain, predictive processing, and the interaction between learning and instructed knowledge.

Our study presents a novel examination of expectancy-based pain modulation during reversal learning. Cue effects on pain reports, skin conductance, and responses to heat in several pain-related regions, including the bilateral anterior insula, dACC, and sgACC, reversed as contingencies changed, whether participants were instructed about contingencies or learned purely through experience. Thus pain, arousal, and brain responses to noxious stimulation are highly sensitive to changing contingencies. However, we did not observe significant cue-based modulation of well-validated neural signature patterns that predict pain (*Wager et al., 2013*; *Woo et al., 2017*), although both patterns predicted pain when controlling for cue. This suggests that cue-based predictions may alter dynamic learning and update decision-making without changing underlying nociception, consistent with previous findings in placebo analgesia (*Zunhammer et al., 2018*). The brain mediators of dynamic cue effects on pain, DLPFC, VMPFC, DMPFC and sgACC, further link our findings with placebo analgesia. Meta-analysis indicates that these regions all play a modulatory role in placebo analgesia, with activation being inversely related to pain (*Atlas and Wager, 2014b*). The negative mediation (i.e. suppression) we observed is consistent with this modulation or down-regulation: Individuals who showed

greater activation of these regions in response to high pain cues showed stronger negative associations between brain activation and subjective pain, controlling for cue. Identifying factors that predict individual differences in brain mediators is an important direction for future research, as is testing the causal contributions of these modulatory regions to placebo analgesia and expectancy-based pain modulation through interventions such as transcranial magnetic stimulation (*Krummenacher et al., 2010*) or in individuals with neurosurgical lesions (*Motzkin et al., 2021*).

While most regions that have been implicated in prior studies of placebo and expectancy-based pain modulation updated dynamically as contingencies changed, we observed sustained responses to initial contingencies in a portion of the brainstem consistent with the rostroventral medulla (RVM). RVM responses to medium heat stimuli were elevated in response to the original high pain cue relative to the original low pain cue and remained elevated throughout the task in both groups. Together with the periaqueductal gray (PAG), the RVM is a key component of the descending pain modulatory system (*Fields, 2004*; *Fields, 2006*), which regulates the release of endogenous opioids that can block ascending pain signals, a process thought to be critical for expectancy-based placebo analgesia (*Levine et al., 1978*; *Ter Riet et al., 1998*; *Amanzio and Benedetti, 1999*). Previous neuroimaging studies have demonstrated that placebo analgesia engages the RVM (*Eippert et al., 2009*; *Crawford et al., 2021*; *Yanes and Akintola, 2022*) and that placebo effects on the RVM are blocked by the opioid antagonist naloxone (*Eippert et al., 2009*). Preclinical work demonstrates that endogenous opioid release engages long lasting pain modulation (*Watkins and Mayer, 1982a*; *Watkins and Mayer, 1982b*). We have hypothesized that placebo analgesia may engage endogenous opioids to lead to long lasting pain modulation, whereas pain-predictive cues may modulate pain dynamically through dopaminergic prediction errors (*Atlas and Wager, 2012*; *Atlas, 2021*). The fact that RVM responds to initial contingencies throughout the task might be consistent with such long-lasting effects, and may explain why instructions reverse placebo analgesia only after brief conditioning (*Schafer et al., 2015*). Interestingly, we did not observe cue-based modulation of the PAG in our analyses; however, an initial processing pipeline that used affine normalization rather than nonlinear warping (*Atlas et al., 2021*) indicated that the PAG also responded to initial contingencies throughout the task, similar to the RVM. Thus both opioidergic regions may be associated with long-lasting effects that do not update as contingencies change. Future studies should combine pain-predictive cues with opioid and dopamine antagonists or employ positron emission tomography to understand the contribution of endogenous opioids and dopamine to different forms of pain modulation.

Cue effects on pain-related brain responses did not differ between groups, i.e. as a function of whether individuals were instructed about contingencies, indicating that responses to contingency reversals and links with subjective pain are similar regardless of whether individuals learn through experience or instruction. Thus once an expectation or prediction is generated, it has similar effects on downstream responses regardless of how contingencies were established. Based on these findings, one might assume that expectancy-based pain modulation and predictive coding operate similarly whether predictions are generated via learning or instruction. However, quantitative learning models revealed differences between groups in how responses updated dynamically from trial to trial, which were in turn associated with differences in brain activation. Individuals who were exposed to instructed reversals updated pain immediately upon instruction without additional learning from intermittent reinforcement, whereas individuals who learned purely from experience had higher learning rates, meaning they updated expectations as a function of pairings between predictive cues and heat outcomes. We observed similar differences when we fit models to heat-evoked skin conductance on medium trials in exploratory analyses. This is consistent with prior work focusing on autonomic arousal during threat learning (*Atlas et al., 2016*) and confirmation bias in reward-related decision making (*Doll et al., 2009*). Isolating the different timecourses of expected value and prediction error then allowed us to identify differences between groups in the brain circuitry associated with these computations.

Several regions, including the rostral anterior cingulate cortex (rACC), showed differential associations with expected value (EV) depending on group. RACC activation was positively associated with EV in the Instructed Group (i.e. greater activation with high pain expectancy), and negatively associated with EV in the Uninstructed Group (i.e. greater activation with low pain expectancy). The rACC also showed preferential associations with instruction-based EV when we compared associations between instruction-based and experience-based EV within the Instructed Group. The rACC has

been implicated in numerous studies of placebo analgesia and expectancy-based pain modulation (*Petrovic et al., 2002*; *Bingel et al., 2006*; *Eippert et al., 2009*; *Geuter et al., 2013*), and is a key component of the opioidergic endogenous pain modulation circuit (*Zubieta et al., 2005*; *Wager et al., 2007*; *Navratilova et al., 2015*), along with the PAG and the RVM. However, in contrast to the RVM, which showed stable responses over time regardless of instruction, the rACC showed different dynamics and different patterns of responses to threat during reversal learning depending on whether individuals were exposed to contingency instructions. This may be consistent with other studies that implicate the rACC as a key region that determines whether or not placebos engage descending modulation through connectivity with the PAG (*Bingel et al., 2006*; *Eippert et al., 2009*); in other words, rACC might act as a hub to flexibly engage downregulation depending on contextual factors. Future studies should examine the relationship between these regions and whether connectivity differs in more dynamic environments or as a function of instruction. We observed similar differences in value coding in the left dorsal posterior insula, a region thought to be specific for pain (*Segerdahl et al., 2015*), and in portions of the bilateral thalamus that connect with the prefrontal cortex and the temporal lobe (*Behrens et al., 2003*; *Johansen-Berg et al., 2005*).

In addition to regions implicated in prior studies of placebo and pain modulation, we evaluated the contribution of systems involved in value-based learning, namely the striatum, OFC/VMPFC, and the amygdala. We previously observed dissociations in these regions during aversive learning (*Atlas et al., 2016*), such that responses in the striatum and OFC/VMPFC tracked EV that updated with instructions, whereas the right amygdala was associated with EV based purely on experiential learning. The present study builds on this work by focusing on heat-evoked responses (i.e. the US in a fear conditioning model), rather than anticipation (i.e. responses to the CS), which allows us to test whether error-based learning shapes responses to noxious stimuli and whether this learning is modulated by instructions. ROI-based analyses indicated that the bilateral striatum was associated with EV that updated based on instruction in the Instructed Group, consistent with our previous findings (*Atlas et al., 2016*). VMPFC/OFC associations with value were also sensitive to instructions, although we observed different patterns across the broad region and did not observe consistent findings in ROI-based or corrected analyses. For instance, uncorrected voxelwise analyses revealed that the association between EV and VMPFC activation differed between groups (positive associations with EV in the Instructed Group and negative associations with EV in the Uninstructed Group), regardless of whether models were fit to pain, heat-evoked SCR, or anticipatory SCR. In contrast, we observed negative associations with EV in the left lateral OFC in both groups. Both portions of the VMPFC/OFC have been implicated in previous studies of placebo analgesia (*Lieberman et al., 2004*; *Wager et al., 2004*; *Petrovic et al., 2010*; *Ellingsen et al., 2013*), and we previously showed that pain-predictive cues modulate responses in both regions, with greater activation when medium heat is paired with low pain cues than high pain cue (*Atlas et al., 2010*). Lateral OFC is thought to evaluate value of independent options, whereas medial OFC and VMPFC have been linked to value-based choice for the purpose of comparison (*Rudebeck and Murray, 2014*). We recently showed that surgical lesions of the bilateral VMPFC enhance instructed cue effects on pain in a stable environment (*Motzkin et al., 2021*). Additional studies that use lesion symptom mapping in heterogeneous frontal samples or measure aversive learning in nonhuman primates can provide further insights on the contribution of different portions of the VMPFC/OFC to pain modulation and instructed learning. Together, these findings build on previous work indicating that the striatum and VMPFC update upon instruction in both appetitive and aversive learning (*Li et al., 2011a*; *Atlas et al., 2016*) and that OFC value signals are sensitive to higher order knowledge across species (*Wilson et al., 2014*; *Lucantonio et al., 2015*; *Schuck et al., 2018*).

Because we focused on responses to the US rather than the CS here, we also were able to extend previous work by examining the role of unsigned prediction errors (PEs), which are linked to associability, which gates attention and learning dynamically. Associability has been linked to the amygdala in previous studies of pain and aversive learning (*Li et al., 2011a*; *Zhang et al., 2016*; *Atlas et al., 2019*), meta-analyses indicate that unsigned PEs are associated with activation in the dorsal and ventral striatum, as well as other regions (*Corlett et al., 2022*), and one study of aversive learning showed responses to both associability and unsigned PE in the amygdala (*Boll et al., 2013*). Consistent with these studies, we observed associations between unsigned PE and activation in the bilateral amygdala and the right striatum in both groups in ROI-based analyses. Whole brain analyses revealed

associations between unsigned PEs and activation in the right hemisphere encompassing the insula, striatum, and amygdala, and that the associations between PE and activation in the right SII and temporal cortex differed across groups (negative associations in the Instructed Group and positive in the Uninstructed Group). Thus, although cue effects on subjective pain were mediated similarly across groups, we still observe unique responses to PE that differed between groups.

We note that the amygdala's association with PE within the Instructed Group indicates that instructions do have some impact on amygdala activation, seemingly in contrast to our previous work on instructed learning during fear conditioning (*Atlas et al., 2016*; *Atlas, 2019*). However, because error-driven learning was nearly absent between instructed reversals in the Instructed Group, the PE regressor in this group captures a constant effect that decreases linearly over time, consistent with prior evidence that amygdala responses habituate during aversive learning (*Büchel et al., 1998*). Thus we did not observe evidence of amygdala sensitivity to instructions per se in the present study. The absence of pure experiential learning signals in the right amygdala may reflect the present study's focus on BOLD responses to noxious heat (i.e. unconditioned stimuli), in contrast to our previous study (*Atlas et al., 2016*) which focused on responses to unreinforced cues (i.e. conditioned stimuli). In addition, our experimental design was optimized for analyses of the heat period rather than the cue (i.e. we used a fixed anticipation period to avoid variations in dread that might accompany jittered anticipatory intervals *Berns et al., 2006*), so we were unable to separately analyze responses to pain-predictive cues to determine whether the amygdala showed similar responses in the present study. We note that raw timecourses suggest that cue-based differences in the amygdala may emerge in response to predictive cues, as well as during the heat period (see *Figure 5—figure supplement 5*). This contrasts with the bilateral striatum and nearly all regions identified in mediation analyses, in which cue-based differences do not emerge until the heat period. Future work should incorporate longer delays between cues and aversive outcomes or utilize intermittent reinforcement (e.g. catch trials without heat presentation) to separately analyze responses to cues and noxious heat in the amgydala, VMPFC, and other key regions.

These results highlight that pain-related outcomes (e.g. subjective ratings, physiological arousal, learning systems, sensory circuits) may show distinct patterns of learning and sensitivity to higher order factors. Studies of expectancy-based pain modulation and predictive coding in other domains should measure parallel behavioral outcomes to capture the complexity of perceptual decision-making, rather than focusing on a single outcome measure (e.g. choices during instrumental learning).

## Future directions and outstanding questions

This work highlights several promising avenues of inquiry that should be addressed in future work, in addition to those highlighted above. Comparisons between appetitive and aversive learning would reveal whether the differences observed here are driven by threat-specific processes or general differences in adaptive learning and flexibility. One limitation of our design is that the two groups received slightly different instructions at the start of the experiment: The Instructed Group was informed about cue-heat contingencies, whereas the Uninstructed Group was instructed to pay attention and try to figure out the relationships between cues and heat outcomes. Although these differences did not impact brain mediators of expectancy or pain, the latter instruction would be more likely to engage inference and model-based learning (*Doll et al., 2012*; *Doll et al., 2015*; *Dayan and Berridge, 2014*). These differences should be resolved in future work by directly comparing instructed and model-based learning. In addition, future studies should include a group that undergoes initial learning in the absence of instructions and then receives instructed reversals to determine whether learned associations can be reversed on the basis of higher order knowledge, consistent with dissociations observed in studies of placebo (*Benedetti et al., 2003*; *Schafer et al., 2015*). Alternatively, future studies can include outcomes that diverge from instructions to examine the interplay between instructed and experiential learning (e.g. *Doll et al., 2009*). In addition, future work should manipulate not only instructions and learning but also the uncertainty of the expectation to understand how instructions, learning, uncertainty, and precision drive perception from a predictive coding framework. Prior work on placebo analgesia indicates that the precision of the expectation is linked to placebo responses in the PAG and RVM (*Grahl et al., 2018*); it is unknown how precision and uncertainty modulated responses in the present study.

We chose to focus on within-subjects effects and did not examine how these differences vary across individuals as a function of factors such as anxiety, which has previously been shown to impact adaptive learning (*Browning et al., 2015*). However, we recently examined how instructed reversals impact pain expectations in youth with clinical anxiety and found that youth with and without anxiety showed similar responses to expectancy and instruction, although youth with anxiety showed greater autonomic arousal during pain anticipation (*Abend et al., 2021*). In exploratory analyses, we measured the association between anticipatory responses to the cues themselves and brain responses to noxious heat, however we did not observe associations between anticipatory arousal and responses within a priori networks. We also found that quantitative models fit to heat-evoked and anticipatory SCR were highly similar to models fit to subjective pain. However, as the number of participants with useable skin conductance or pupil dilation data was extremely limited due to technical malfunctions and variations in arousal, we consider these findings exploratory and do not make strong claims based on these results. Future studies should continue to compare the effects of learning, instructions, and expectations on subjective pain with effects on physiological outcome measures to determine whether autonomic responses to noxious stimuli are shaped by pain decision-making (*Mischkowski et al., 2019*) or whether physiological responses are shaped through independent pathways, e.g. through a dual process model (*Ohman and Soares, 1993*; *Mineka and Ohman, 2002*).

## Conclusion

Together, these findings reveal that instructions and learning lead to both interactive and dissociable processes even within individuals. We view these findings in light of theories on the relationship between conditioning and expectancy (*Rescorla, 1988*; *Kirsch, 1997*; *Kirsch et al., 2004*) and long-standing debates about whether placebo effects depend on conditioning or expectancy. We suggest that considering the brain mechanisms that mediate dynamic expectancy-based pain modulation shines new light on these distinctions. The human brain contains parallel pain modulatory circuits that (i) update as contingencies change (e.g. insula), (ii) continue to respond to initial contingencies regardless of whether they were learned through instruction or experience (e.g. RVM), or (iii) respond to experiential learning differentially as a function of whether or not individuals were exposed to instructions (e.g. rACC). These findings indicate that we gain new insights on clinically relevant outcomes from measuring how instructions and learning interact to shape outcomes, rather than assuming that circuits and processes are sensitive to either expectancy or conditioning. Understanding these processes in clinical populations may shed light directly on the mechanisms of therapeutic interventions, for example the interplay between instructed and exposure-based interventions in cognitive behavioral therapy for chronic pain and affective disorders.

## Materials and methods

### Participants

Forty-nine participants (25 female, $M_{age}$ = 28.04 years, $SD_{age}$ = 7.04) were recruited and consented to participate in an fMRI study designed to measure 'how pain and emotions are processed in the human brain and influenced by psychological factors'. Participants provided informed consent in accordance with the Declaration of Helsinki, and the protocol was approved by the NIH's Combined Neuroscience Institutional Review Board (Protocol 15-AT-0132, PI: Atlas). Participants were eligible to participate if they were between 18 and 50, fluent in English, healthy (i.e. had no medical conditions that affect pain or somatosensation, no psychiatric, neurological, autonomic, or cardiovascular disorders, no chronic systemic diseases, and no medication that can affect pain perception), right-handed, and had received a medical exam at NIH within the previous year. All participants underwent urine toxicology testing to ensure they had not used recreational drugs that alter pain. Participants were drawn from a pool of subjects who had completed an initial screening visit that tested whether participants reliably reported increased pain with increased temperatures ($r^2$ >0.4) and exhibited pain tolerance at or below 50 °C (the maximum temperature we applied during the study). Nine participants who provided consent did not complete the experiment due to ineligibility based on calibration (n=4) technical failures (n=1), compliance with procedures (n=2), or anatomical abnormalities identified in a clinical scan (n=2) and were not included in the current analyses. The final sample included 40 participants (22 female; $M_{age}$ = 27.00 years, $SD_{age}$ = 6.21). As detailed in our clinical protocol (15-AT-0132; https://clinicaltrials.gov/

identifier NCT02446262), sample size was based on power analyses from our previous studies and on a behavioral pilot experiment conducted prior to the fMRI study. We computed the effect size of cue-based differences in reported pain in a sample of 12 participants (6 per group) and determined that we need a minimum of 5–9 participants to achieve 80–95% power to detect cue-based differences in pain, including reversals (data available upon request). We therefore included 20 participants per group in the fMRI experiment. Two participants in the Uninstructed Group and two participants in the Instructed Group were excluded from fMRI analyses due to excessive head motion or technical issues during the scan (see Procedure), leaving a final sample of 36 participants for fMRI analyses (18 Uninstructed Group, 18 Instructed Group). All 40 participants were included in behavioral analyses.

## Materials and procedure

### Stimuli and apparatus

We delivered thermal stimulation to the left (non-dominant) volar forearm using a 16x16 ATS contact heat thermode controlled with a Pathway pain and sensory evaluation system (Medoc Ltd, Ramat Yisha, Israel). Each heat stimulus lasted 8 s and consisted of three phases: a 1.5 s on-ramp phase in which the temperature of the thermode rose from 32 °C to the target temperature level, a 5 s plateau phase in which target temperature was maintained, and a 1.5 s off-ramp phase in which the temperature returned to 32 °C. Thermode placement was adjusted between each block of trials (i.e. every 12 trials) to avoid sensitization, habituation, and skin damage. Temperatures ranged from 36°C to 50°C, in increments of 0.5 °C, and were selected based on a thermal pain calibration conducted immediately prior to the experiment. Thermode temperature was maintained at 32 °C between trials.

Experiment Builder (SR-Research, Ontario, Canada) was used to deliver visual and auditory stimuli, to trigger noxious stimulation on the Pathways computer, and to synchronize task timing with physiological recording. Physiological data, including electrodermal activity (EDA), respiration, electrocardiography, and peripheral pulse, were recorded from the left hand using Biopac recording equipment and accompanying AcqKnowledge software (Goleta, CA) at a sample rate of 500 Hz. Participants recorded pain ratings using a trackball with their right hand while EDA was recorded from the left hand and heat was applied to the left arm. EDA was collected using Biopac's EDA100C-MRI module (Biopac Systems, Inc, Goleta, CA). For each participant, two pre-gelled EDA electrodes (EL509; Biopac Systems, Inc, Goleta, CA) were prepared and applied to the hypothenar muscles of the left hand. Pupillometry and gaze position was recorded with Eyelink 1000-Plus (SR-Research, Ontario, Canada).

Participants also completed questionnaires prior to the experiment, including the State-Trait Anxiety Inventory (STAI Form X *Gaudry et al., 1975*), the Positive and Negative Affect Scale (*Watson et al., 1988*), and Behavioral Inhibition/Activation Scale (*Carver and White, 1994*). For the present manuscript, we focused on pain reports and brain responses evoked by painful heat.

### Procedure

Participants underwent an adaptive staircase pain calibration prior to the experimental task (see *Figure 1A*). The adaptive staircase calibration procedure has been described in depth in previous work (*Atlas et al., 2010*; *Atlas et al., 2012*; *Mischkowski et al., 2019*; *Dildine et al., 2020*; *Amir et al., 2021*). Participants experience temperatures across eight skin sites on the left volar forearm and provide ratings on a 10-point scale, where 1 denotes warmth, 2 denotes pain threshold, 5 denotes moderate pain, 8 denotes maximum tolerable pain, and 10 denotes the most pain imaginable. We use iterative regression to isolate each participant's threshold, tolerance, and the strength of the correlation between temperature and pain (i.e. $r^2$). All participants had previously completed the task outside of the MRI facility to establish initial eligibility, and completed the task again on the day of the fMRI scan (*Amir et al., 2021*). Participants were eligible to continue if they reported reliable increases in pain as a function of temperature ($r^2 > 0.4$) and reported maximum pain tolerance at 50 degrees or less. Four participants were deemed ineligible based on calibration on the day of the study. The calibration procedure also allowed us to identify four skin sites per individual that responded most similarly across temperatures (i.e. lowest average residuals based on overall temperature-pain regression) and to individually calibrate temperatures associated with ratings of low pain (2 on 10-point scale), medium pain (5 on 10-point scale), and high pain (maximum tolerable pain; 8 on 10-point scale). These temperatures and skin sites were used during the main experiment, as described below.

Following the pain calibration, each eligible participant was positioned in the fMRI scanner. They were then randomized (n=20 per group) to either the Instructed Group (12 female; $M_{age}$ = 27.05 years, $SD_{age}$ = 6.64) or the Uninstructed Group (10 female; $M_{age}$ = 26.95 years, $SD_{age}$ = 5.91) and given instructions about the experiment (see *Figure 1A*). Both groups were told "In this task you will hear two sounds, followed by heat from the thermode. The sound will last a few seconds and will be followed by a short variable delay period, and then heat from the thermode. You do not need to respond when you hear the cue." Instructed Group participants were then told, "It is there just to let you know what level of heat will be next." They then heard both cues and were informed which cue would be followed by high heat and which would be followed by low heat stimulation. Uninstructed Group participants were instead told "Your job is to pay attention to the cues and try to figure out the relationship between the sounds that you hear and the heat that you feel." They then heard both cues but were simply told that one was the first cue and the other was the second cue. We used two auditory cues (a cymbal and an accordion), which were counterbalanced across participants. Following instructions and between each block of the task, participants provided expectancy ratings in response to each cue and an experimenter moved the thermode to the next inner arm location.

During the experiment, each trial began with a 2 s auditory cue, followed by a 6 s anticipation interval, and then heat from the thermode (see *Figure 1B*). After an 8–10 s temporal jitter, participants provided pain ratings using an 8-point visual analogue scale. They were instructed to use the same anchors as the adaptive staircase calibration (i.e. 1=non-painful warmth, 2=pain threshold, 8=maximum tolerable pain) and we included boxes at 0 and above 8 to denote 'no sensation' and 'too hot', respectively (see *Figure 1B*). Participants viewed the scale for 3 s then had 5 s to record their ratings. There was a 4–6 s temporal jitter before the next cue was presented. The two jitters always combined to 14 s within a single trial, for a total trial duration of 48 s. There were 7 blocks of trials with 12 trials per block (i.e. 84 trials total). We used two trial orders (counterbalanced across participants within Group) that each included (1) a brief conditioning phase, (2) a test of cue-based expectancy effects, and (3) three contingency reversals (*Figure 1C and D*). During the conditioning phase, Original Low cues were followed by stimulation calibrated to elicit ratings of low pain and Original High cues were followed by stimulation calibrated to elicit high pain (see *Figure 1C and D*). The conditioning phase included 3 Original High cue +high heat pairings and 2–3 Original Low cue +low heat pairings (i.e. 5–6 trials; see *Figure 1D*). Following conditioning, each cue was paired intermittently with stimulation calibrated to elicit ratings of medium pain. This provides a test of cue-based expectancy effects, consistent with our previous work (*Atlas et al., 2010*; *Johnston et al., 2012*; *Michalska et al., 2018*; *Abend et al., 2021*). Intermittent reinforcement continued at a 50% reinforcement rate until contingencies reversed.

Halfway through runs 2, 4, and 6, the screen displayed instructions to the Instructed Group indicating that contingencies had reversed ("The relationship between the cues and heat will now reverse"; see *Figure 1C*). Immediately following instructions, participants experienced at least one medium heat trial paired with each cue, which provides an immediate test of instructed reversals in the Instructed Group (*Atlas et al., 2016*; *Atlas and Phelps, 2018*; *Abend et al., 2021*). Following the medium heat trials, the new contingencies were reinforced (i.e. the high pain temperature was paired with the previous low pain cue and the low pain temperature was paired with the previous high pain cue), leading to an experience-based reversal. Medium trials were then delivered, and learning continued with the same reinforcement rates until the next reversal (see *Figure 1D*). Uninstructed Group participants experienced the same trials, but a fixation cross was displayed instead of instructions.

We used two pseudorandom trial orders with three contingency reversals, which were counterbalanced across participants. Each trial order ensured that no condition was repeated three times in a row. When we visualized responses as a function of trial order, we noticed that one trial order presented the same cue-heat condition as the first trial on 6 out of the 7 blocks (i.e. medium heat paired with the original high pain cue). Because the first trial of each block is applied to a new skin site, the novel stimulus was rated as much higher than all other trials. We therefore omitted the first trial from all analyses in the main manuscript, because this novelty response contaminated the otherwise strong reversal behavior. Furthermore, due to a programming error, Instructed Group participants in one of the two trial orders received two incompatible trials following the third instructed reversal (i.e. they received one low stimulus with the previous low cue and one high stimulus with the previous high cue). Because this experience contradicted instructions and we are interested in measuring the effects

of veridical instructions on reversal learning, we only analyzed the trials prior to these instructions (i.e. two reversals instead of three) in these participants (n=10).

Following the fMRI scan, participants rated affect associated with each cue and provided retrospective expectancy ratings to report how much pain they expected in response to each cue at the beginning and the end of the task.

## BOLD FMRI data acquisition and preprocessing

BOLD fMRI data were collected on a 3T Siemens Skyra scanner at the NIH's MRI Research Facility / Functional Magnetic Resonance Imaging Facility. After positioning the participant in the scanner bore, we collected a localizer followed by a T1-MPRAGE collected in the sagittal plane (256 slices). We collected 7 runs of multi-echo data with a 2.5 s TR and 3 mm isotropic voxels (191 volumes collected anterior to posterior; flip angle = 90°; acquisition matrix = 70 x 0 x 0x64; $1^{st}$ echo = 11ms; $2^{nd}$ echo = 22.72ms; $3^{rd}$ echo 34.44ms).

Multi-echo data were preprocessed and combined using the 'afni_proc.py' program in Analysis of Functional Images (AFNI; *Cox, 1996*). We used '@SSwarper' to nonlinearly align the anatomical scans to the MNI152_T1 template (MNI152_2009_template_SSW.nii.gz in AFNI). These nonlinear transformations were passed to afni_proc.py for general preprocessing and quality control. We removed the first 4 TRs of functional data to reach magnetization steady state, leaving a total of 187 TRs of fMRI data per run during subsequent processing and analysis steps. We performed slice time correction using AFNI's 3dTshift program then performed motion correction by aligning echo 2 volumes to a low-motion minimum outlier base image (MIN_OUTLIER, estimated by '3dToutcount') and warped to align with the anatomical volume and with the template. All volumetric transformations, both linear and non-linear, were combined into a single transformation to avoid repeated resampling of the EPI volumes. Motion correction estimates were computed based on echo 2 volumes using 6 parameters (3 translations, 3 rotations) and we applied the same corrections to each echo. We made sure there was valid data for each echo and each TR prior to combining multi-echo data. We combined across the three echoes using AFNI's @compute_OC_weights function to generate a weighted combination of the three echoes. These 'optimally combined' data were used for subsequent analyses. Preliminary analyses indicated that combining across the echoes with optimal combination led to better heat-related activation in pain-related regions than analyses of a single echo or using other approaches to combine multi-echo data (e.g. TE-dependent analysis *Kundu et al., 2012*; *Lombardo et al., 2016*). In future analyses, we may formally compare optimal combination with other approaches for echo combination and denoising. Following optimal combination, data were normalized to percent signal change and smoothed using a 4 mm full-width half max smoothing kernel. Data were then analyzed using single trial estimates in MATLAB (The Mathworks, Inc, Natick, MA), as described below. Four participants were excluded from fMRI analyses due to technical issues with the fMRI scanner during data collection (n=2) or excessive head motion (motion >2 mm; n=2), leaving a final sample of 36 participants for fMRI analyses (18 in Instructed Group).

## Psychophysiological data processing

Skin conductance data was preprocessed in AcqKnowledge (Biopac Systems, Inc, Goleta, CA). During preprocessing, data were smoothed (1000-sample Gaussian smoothing kernel) and filtered (25 Hz FIR low-pass filter) in AcqKnowledge, then imported into MATLAB and downsampled to 250 Hz. Data were then analyzed using Ledalab's continuous decomposition analysis (CDA), which accounts for phasic and tonic signals and can capture multiple responses during the 8 s heat period (*Benedek and Kaernbach, 2010*). We analyzed both (a) cue-evoked anticipatory responses that occurred within 4 s following cue presentation; and (b) responses that occurred between heat onset and 4 s after heat offset. We used the average phasic driver ('SCR') in trial-wise analyses. Participants were included in analyses if they had >4 trials with measurable SCR. Thirty-six participants were included in analyses across temperatures (18 in Instructed Group); 30 participants were included in analyses of responses to medium heat (13 in Instructed Group); and 29 participants were included in analyses of anticipatory responses (15 in Instructed Group).

Pupillometry data were processed in MATLAB using both publicly available software and custom code. Data were imported to MATLAB and blinks were interpolated using the 'GazeVisToolbox' (available at https://github.com/djangraw/GazeVisToolbox, copy archived at

swh:1:rev:afbb6cb89e6fc494e0a578d44461af085fa5d46b; *Jangraw et al., 2014*). Consistent with our previous work (*Mischkowski et al., 2019*), we interpolated from 100ms prior to each blink to 100ms following each blink to avoid extreme values surrounding each blink. Data were aligned to event markers and we visualized individual trials to exclude trials that were contaminated by artifacts. Subjects with fewer than seven useable trials (n=20; 9 in Instructed Group) were excluded from analyses of pupillary data. Useable subjects had a mean of 42.9 useable trials (*M*=54.4% of trials). 1000 Hz data was downsampled to 10 Hz and then we computed baseline-corrected mean pupil dilation and area-under the curve as measures of pupillary response during the heat period.

## Statistical analysis of expectations, pain, autonomic responses, and heat-evoked neural signature pattern expression

We used the statistical software R (*R Development Core Team, 1996*) to analyze effects of our experimental manipulations on expectancy ratings, pain reports, physiological arousal, and heat-evoked brain signature pattern expression (see below, "Brain-based classifier analyses"). We measured effects across the entire task, as well as before the first reversal. We used two-way mixed ANOVAs implemented through the R function 'anova_test' from the package "rstatix" (*Kassambara, 2021*) to analyze the effects of Group and Cue on expectancy ratings prior to the task and after conditioning, as well as post-task ratings. All other analyses were conducted using multilevel linear mixed effects models in R.

In each analysis, we modeled within-subjects effects of Cue (original high pain cue / CS+ versus original low pain cue / CS-), Phase (original versus reversed contingencies), and Cue x Phase interactions (i.e. current high pain cue / CS+ versus current low pain cue / CS-) at the first level, and Group was modeled at the second (i.e. between-subjects) level. Consistent with our previous work on aversive reversal learning (*Atlas et al., 2016*; *Atlas and Phelps, 2018*), reversals (i.e. Phase effects) were coded relative to instructed reversals in the Instructed Group, and relative to experienced reversals in the Uninstructed Group (i.e. the first time the previous high pain cue was paired with low heat, or vice versa; see *Figure 1C*). Analyses across temperatures also included a first-level factor for Heat Intensity, and analyses during acquisition omitted the effect of Phase. We also included an effect of Time (linear effect of trial) in model comparisons to evaluate whether cue effects varied over time, although Bayesian model comparison indicated that including Time did not improve any models.

We evaluated linear mixed models using both Bayesian and frequentist statistics packages to ensure results were robust to analysis approaches. Bayesian linear mixed models were implemented using the *brms* package (*Bürkner, 2017*) in R (*R Development Core Team, 1996*). All coefficients were evaluated with a Gaussian prior centered on 0 with a 2.5 SD (i.e. a mildly informative conservative prior). For each outcome measure, we evaluated all potential models ranging from intercepts only to a maximal model (*Barr et al., 2013*) with random slopes for every factor and all potential interactions, then used 'bayesfactor_models' from the *bayestestR* package (*Makowski et al., 2019a*) to perform model comparison and to draw inferences about acceptance and rejection of null effects. In most cases, Bayesian model comparisons supported maximal models (*Barr et al., 2013*), that is including all fixed factors (except Time), all interactions, and random intercepts and slopes for each subject. We used the 'describe_posterior' function (*Makowski et al., 2019a*) to evaluate probability of direction (similar to a frequentist p-value with values >95% indicating the effect is likely to exist), and the percentage of the posterior distribution within the region of partial equivalence (ROPE), which can delineate significance. Published guidelines suggest that when if <1% of the distribution is within the ROPE, one can reject the null hypothesis, whereas if >99% is in the ROPE, one can accept the null hypothesis (*Makowski et al., 2019b*).

We then evaluated the corresponding model using 'lmer' from the *lme4* package (*Bates et al., 2015*) to provide frequentist statistics and using 'lme' from the *nlme* package (*Pinheiro et al., 2021*) to incorporate autoregression. If frequentist models corresponding to the best Bayes model did not converge, we removed the covariance between random effects (i.e. specified that slopes and intercepts are uncorrelated using ||), and we used optimizers to achieve convergence. Final models are specified in *Tables 1–3*. We acknowledge findings from all approaches in our Results, using guidelines for Bayesian modeling from *Makowski et al., 2019b*; *Makowski et al., 2019a*. Results across approaches were largely consistent (see *Tables 1–3*).

## Computational modeling of pain reversal learning and modulation by instructions

We applied a computational model of instructed reversal learning (**Atlas et al., 2016**; **Atlas and Phelps, 2018**) to predict pain reports on medium heat trials. The model is a standard reinforcement learning model (**Rescorla and Wagner, 1972**) that describes how expected value (EV) updates in response to prediction errors in the environment, depending on learning rate. We include an additional parameter, $\rho$, which guides whether cues' expected values (*EV*) are exchanged when instructions are delivered. If $\rho = 0$, each cue's EV remains stable (i.e. retains the same *EV* prior to instructions) whereas if $\rho = 1$, the EVs of the two respective cues are exchanged. Mathematically for two cues (a and b), this is implemented for trial (t) as:

$$EV_{t+1}(x_b) = \rho * V_t(x_a) + (1 - \rho) * V_t(x_b)$$

For complete details, see **Atlas et al., 2016**. The present model fitting procedure differed in that we fit models to pain ratings on medium heat trials, rather than SCR on unreinforced trials, although we fit models to SCR in exploratory analyses, as described below. In the present study, we assume that pairings between predictive stimuli and high and low intensity heat engages stimulus-based learning, similar to CS-US pairings in classical conditioning experiments. Fitting to medium heat trials isolates the timecourse of expected value, since the stimulus temperature is constant, and therefore the only factor likely to guide cue-based variation in pain is presumably the cue's dynamic expected value based on learning and/or instructions.

Consistent with our previous work (**Atlas et al., 2016**; **Atlas and Phelps, 2018**), the initial EV was set as 0.5 for Uninstructed Group participants, while we used asymmetric initial expected values for Instructed Participants who were informed about contingencies (1 for high pain cue; 0 for low pain cue). Models were fit at the level of the individual, with two free parameters ($\alpha$ and $\rho$), and we used 20 iterations with a random starting value for each parameter for each iteration. We minimized the deviance between expected value and pain on medium heat trials. EV was bounded between 0 and 1; however, we used MATLAB's glmfit.m function to relate EV with pain on medium heat trials using the normal link function, which includes an intercept and therefore accounts for differences in scale between EV and pain reports. Consistent with other analyses, the first trial of every block was not included during model fitting due to novelty responses.

To evaluate goodness-of-fit, we compared the instructed Rescorla-Wagner model described above with three other variations of plausible learning models: (a) an instructed Rescorla-Wagner model with initial value, $\alpha$, and $\rho$ as free parameters; (b) a traditional Rescorla-Wagner model without an instructed reversal parameter (initial value and $\alpha$ as free parameters); and (c) an instructed model of associability (i.e. a Hybrid model; **Li et al., 2011b**) with $\rho$, $\eta$, and κ as free parameters (**Atlas et al., 2019**). Models were fit to each individual's pain ratings on medium heat trials, since the stimulus temperature was constant and therefore the only factor likely to guide cue-based variation in pain would presumably be dynamic expected value based on learned and/or instructed value. Models were analyzed at the level of the individual, which provides estimates that can be tested using standard statistics, and through an iterative jack-knife procedure, which is less sensitive to noise based on individual estimates (**Wu, 1986**; **Miller et al., 1998**; **Atlas and Phelps, 2018**). See **Atlas and Phelps, 2018** for complete details on jack-knife model fitting procedures. Group analyses and model comparisons were conducted using individual estimates. Goodness-of-fit was evaluated based on Akaike's Information Criterion (**Akaike, 1974**), which evaluates model fit with a penalty for additional parameters and is appropriate for nested models, and models were compared based on Bayesian Model Selection, as implemented in SPM's SPM_bms.m (**Stephan et al., 2009**).

The instructed Rescorla-Wagner model with pre-determined initial values and free parameters for $\alpha$ and $\rho$ (i.e. the same model used in our previous work; **Atlas et al., 2016**; **Atlas and Phelps, 2018**) fit the data better than any of the alternatives, as indicated by Bayesian Model Selection, which gave this model a posterior probability of 87.95% relative to the other alternatives (Rescorla-Wagner with initial free parameters: 3.61%; Rescorla-Wagner without instruction parameter: 5.1%; Hybrid model: 3.35%). We therefore focus on this model in the main manuscript and used this model to generate regressors for use in fMRI analyses. We used t-tests on subject-level fits to assess whether each parameter ($\alpha$ and $\rho$) varied as a function of group. The mean parameters based on jack-knife estimates were used to generate regressors for fMRI analyses (see 'Neural correlates of expected value', below).

We used the same approach to fit computational models to anticipatory SCR and SCR evoked during medium heat simulation.

## FMRI analyses
### Single-trial analyses

Following preprocessing in AFNI, we used single trial analyses to estimate heat-evoked responses on a trial-by-trial basis and avoid assumptions about the fixed shape of the hemodynamic response. We used flexible basis functions optimized to capture heat-evoked BOLD responses, consistent with previous work on heat-evoked fMRI (*Atlas et al., 2010*; *Atlas et al., 2012*; *Atlas et al., 2014a*; *Wager et al., 2013*; *Woo et al., 2017*). We applied principal components analysis-based spike detection (scn_session_spike_id.m, available at https://canlab.github.io/; *Wager, 2022*) to identify potential spikes and noise in the data which were modeled as nuisance covariates, along with the movement parameters from AFNI's preprocessing pipeline. We used the function single_trial_analysis.m (https://canlab.github.io/) to generate trial-by-trial estimates of height, width, delay, and area-under-the-curve (AUC) for each heat period. Trials in which subjects failed to respond were omitted from analyses ($M_{all}$ = 1.69, $SD_{all}$ = 2.38; $M_{Instructed}$ = 1.11, $SD_{Instructed}$ = 1.08, $range_{Instructed}$ = 0–3; $M_{Uninstructed}$ = 2.28, $SD_{Uninstructed}$ = 3.12, $range_{Uninstructed}$ = 0–11). We focused on AUC estimates in subsequent analyses, consistent with our previous work. We computed variance inflation factors (VIFs) using the single_trial_weights_vifthresh.m function to identify bad trials, that is those who coincided with spikes or motion and were therefore not reliable estimates. We excluded any trials with VIFs >2 from subsequent analyses ($M$=3.39, $SD$ = 2.38), and smoothed trial estimates with a 4 mm gaussian kernel, consistent with previous work (*Atlas et al., 2010*; *Atlas et al., 2012*; *Atlas et al., 2014c*). Trial estimates were passed into voxel-wise second level analyses across trials and across participants using the general linear model (fit_gls_brain.m; https://canlab.github.io/) and robust regression (robfit.m; https://canlab.github.io/) to examine neural correlates of associative learning (see below, '*Neural correlates of expected value*'). Trial-level estimates were also employed in multilevel mediation analyses (see below, '*Multilevel mediation analyses*').

### Multilevel mediation analyses

We used multilevel mediation to examine whether brain activity mediated the effect of predictive cues on subjective pain on medium heat trials. Mediation was implemented by the MATLAB function *mediation.m* (https://canlab.github.io/). Cue was included as the input variable (i.e. X; coded as 1 for High cue, –1 for Low cue), pain was included as the output variable (i.e. Y), and we searched for potential mediators. Voxelwise mediation, or mediation effect parametric mapping (*Wager et al., 2009*; *Atlas et al., 2010*) yields interpretable maps that are similar to simultaneous partial regressions, although implemented using mixed effects. The mediation effect (a*b) identifies regions whose activity contributes to variance in the effect of the independent variable on the dependent variable (Path *c*). For individual *i*, trial *j*:

$$Y_{ij} = d_{0j} + c_j X_{ij} + e_{0ij}. \tag{1}$$

Path *a* denotes the effect of the input variable on the potential mediator (M; brain activation in a given voxel), thereby representing cue effects on brain responses to medium heat:

$$M_{ij} = d_{1j} + a_j X_{ij} + e_{1ij}. \tag{2}$$

Path *b* measures the association between the mediator and outcome, controlling for the input variable. Here, this represents brain regions that predict pain, controlling for cue type:

$$Y_{ij} = d_{2j} + c'_j X_{ij} + b_i M_{ij} + e_{2ij}. \tag{3}$$

In multilevel mediation, the difference between the total effect (Path *c*: the effect of cues on subjective pain) and the direct effect (Path c`: the effect of cues on subjective pain when controlling for the

mediator, see *equation 3*) is equivalent to the sum of the product of Path *a* and Path *b* coefficients and their covariance (*Shrout and Bolger, 2002*; *Kenny et al., 2003*). Our mixed models included intercepts ($d_{0-2}$) and error terms ($e_{0-2}$) for each individual. Group level estimates are computed by treating person-level intercepts and slopes as random effects. For complete details, see *Atlas et al., 2010*.

We performed two voxel-wise multilevel mediation analyses: (1) a search for brain mediators of current cue effects on pain, i.e. the Cue x Phase interaction (see *Figure 5*), and (2) a search for mediators of original cue effects on pain, controlling for current contingencies (see *Figure 6*). For both models, Path *a* evaluates the effect of predictive cue on brain response to medium heat, Path *b* captures the association between brain response and pain, controlling for cue, and the mediation effect (Path a*b) tests whether brain responses contribute to variance between cues and subjective pain on medium heat trials. We evaluated mediation overall irrespective of group, and with Group as a potential moderator of all paths.

We focus in particular on effects of current contingencies (i.e. Cue x Phase interactions) on pain to capture the behavioral reversals we observed (*Figure 5*). In this main analysis, we coded the input variable (X) as the contrast between current high cues (i.e. original high cues during original contingencies and original low cues during reversed contingencies both coded as 1) and current low cues (i.e. original low cues during original contingencies and original high cues during reversed contingencies both coded as –1). Our second mediation searched for mediators of original cue effects (i.e. original high cue coded as 1, original low cue coded as –1) and controlled for current contingencies by including the X variable from the current cue contingencies mediation (i.e. the Cue x Phase interaction) as a covariate in all paths. These effects are thus most likely to be driven by responses during the reversed runs. In both mediations, Path *b* evaluates the link between brain activation and pain, while controlling for the cue (see *equation 2* above).

We ran two types of mediation analyses on medium heat trials: (a) voxel-wise mediation analyses, which search for brain regions that mediate the effects of predictive cues on pain; (b) statistical tests of whether brain responses within ROIs formally mediated cue effects on subjective pain. We evaluated mediation analyses irrespective of Group, and with Group as moderator. We used bootstrapping to estimate the significance of the mediation effect (*Shrout and Bolger, 2002*; *Kenny et al., 2003*) in analyses irrespective of Group, and used ordinary least squares to estimate moderated mediation when Group was included in the model. We omitted the first trial of each run from analyses to be consistent with behavioral results.

## Neural correlates of expected value and prediction error

Whereas our mediation analyses tested effects of Phase, i.e. immediate changes in response to instruction or contingency reversal, we used quantitative models to test whether expected value dynamically shapes responses to noxious stimulation. We used parameters from the best-fitting models for each group, based on jack-knife estimation, to generate the timecourse of expected value (EV) for each subject based on their sequence of trials. We chose to use the mean of the group-level estimates to avoid noise that might come from individual-level model fits. We examined the neural correlates of expected value on medium heat trials only, which avoids confounds due to temperature. We focused on how expected value influenced responses to medium intensity heat, rather than responses to cues themselves, as we were most interested in how pain-related responses are influenced by learned expectations, and we did not optimize the anticipatory period to jointly estimate cue-evoked responses and responses to heat. Our main manuscript focuses on the timecourse of EV based on fits to pain reports; we include associations based on fits to heat-evoked and anticipatory SCR in Supplementary Materials.

Noxious stimulation might also be accompanied by prediction errors; for example, if an individual expects high pain and receives medium heat, this should generate an appetitive PE (i.e. better than expected) if the deviation is noticed. However, expected value and prediction error are inversely correlated in the standard RL model we used. We therefore measured associations with unsigned PE (i.e. the absolute value of PE), as signed PE is the inverse of expected value on our medium heat trials. We focused on responses to medium heat trials only as PE would be correlated with temperature

if we analyzed responses across all trials (i.e. low temperatures are always expected or better than expected, whereas high temperatures are always expected or worse than expected).

We report three group-level analyses: (1) Analysis across all participants testing for differences by group; (2) Analyses within each group to isolate effects of instructed learning (Instructed Group) or feedback-driven learning (Uninstructed Group); (3) Comparisons of instructed and feedback-driven learning within the Instructed Group. Individual results were computed using the MATLAB function fit_gls_brain (https://canlab.github.io/) and group results were computed using robust regression (*Wager et al., 2005*) using the function robfit.m (https://canlab.github.io/).

## Pain modulatory regions of interest

We tested for cue-based modulation of brain regions that have been previously implicated in studies of expectancy-based pain modulation by applying an a priori mask generated from our previous meta-analysis of fMRI studies of placebo analgesia and expectancy-based modulation (*Atlas and Wager, 2014b*). We included regions that showed either expectancy-related increases or decreases in activation within the mask. *Figure 5—figure supplement 1A* depicts the mask, which includes regions that show increased activation with expected pain relief (i.e. activation inversely related to subjective pain) such as the DLPFC, rACC, PAG, and VMPFC, and regions that show reduced activation with expected pain relief, including the insula, thalamus, cingulate, and secondary somatosensory cortex. We report results FDR-corrected within this mask to evaluate responses within pain modulatory regions.

We also tested for effects on two recently developed brain-based classifiers that have been shown to be sensitive and specific to acute pain, the Neurologic Pain Signature (NPS; *Wager et al., 2013*) and the Stimulus Intensity Independent Pain Signature (SIIPS; *Woo et al., 2017*). We used the unthresholded NPS pattern for all analyses, and the function apply_mask.m (https://canlab.github.io/) to compute dot products. We computed the dot-product of each signature with beta coefficients and contrast maps for analyses of computational models, and trial-level images to use the brain response as an outcome in our multilevel mediation analyses.

## Value-processing regions of interest

In addition to pain modulatory networks, we were also interested in testing effects of predictive cues on brain regions involved in value-based learning. To this end, we examined responses within 5 a priori regions of interest (see *Figure 5—figure supplement 1B*): the bilateral striatum, bilateral amygdala, and the ventromedial prefrontal cortex (VMPFC). We used the same ROI masks that were applied in our prior work on instructed reversal learning (*Atlas et al., 2016*). While the amygdala and striatum masks were defined based on Atlases in MNI space (amygdala ROI available at https://canlab.github.io/; striatum ROI based on combining putamen and caudate masks from the Automated Anatomical Labeling atlas for SPM8 [http://www.gin.cnrs.fr/AAL; *Tzourio-Mazoyer et al., 2002*]), the VMPFC ROI was functionally defined in our previous work (*Atlas et al., 2016*) by analyzing deactivation in response to shock. We used the same ROI mask here since analyses as a function of heat intensity elicited significant decreases in this region. We averaged trial-level AUC estimates across each ROI to conduct mediation analyses and averaged across beta coefficients and contrast maps to analyze ROI-wise associations with expected value. Results are reported in *Table 4*.

## Whole brain exploratory analyses

In addition to the analyses in a priori networks and regions of interest involved in pain, placebo, and value-based processing, we also conducted exploratory voxel-wise whole brain analyses. We report whole brain results at FDR-corrected q<0.05 in the main manuscript, and present exploratory uncorrected results at p<0.001 in Source Data accompanying each figure for completeness and for use in future meta-analyses. In cases when FDR-corrected p-values exceeded p=0.001, we use uncorrected results for inference. Anatomical labels were identified using the SPM Anatomy Toolbox (*Eickhoff et al., 2005*), which includes the Thalamic Connectivity Toolbox (*Behrens et al., 2003*; *Johansen-Berg*

*et al., 2005*), and the Brainstem Navigator (http://www.nitrc.org/projects/brainstemnavig/; *Singh et al., 2021*).

## Acknowledgements

This research was funded by the Intramural Research Program of the National Center for Complementary and Integrative Health (Project ZIAAT-000030; PI Lauren Atlas) as well as NIMH Intramural Research Program Project ZIAMH-002782 (PI Daniel Pine). We thank Linda Ellison-Dejewski and Adebisi Ayodele for assistance with participant screening and clinical support, and thank Paul Taylor for assistance with fMRI preprocessing pipelines. The authors declare no conflicts of interest.

## Additional information

### Funding

| Funder | Grant reference number | Author |
|---|---|---|
| National Center for Complementary and Integrative Health | ZIAAT-000030 | Lauren Y Atlas |
| National Institute of Mental Health | ZIAMH-002782 | Daniel S Pine |

The funders had no role in study design, data collection and interpretation, or the decision to submit the work for publication.

### Author contributions

Lauren Y Atlas, Conceptualization, Resources, Data curation, Software, Formal analysis, Supervision, Funding acquisition, Investigation, Visualization, Methodology, Writing - original draft, Project administration, Writing – review and editing; Troy C Dildine, Esther E Palacios-Barrios, Conceptualization, Data curation, Formal analysis, Investigation, Methodology, Project administration, Writing – review and editing; Qingbao Yu, Data curation, Software, Formal analysis, Methodology, Writing – review and editing; Richard C Reynolds, Resources, Data curation, Software, Methodology, Writing – review and editing; Lauren A Banker, Data curation, Formal analysis, Investigation, Methodology, Project administration, Writing – review and editing; Shara S Grant, Data curation, Formal analysis, Writing – review and editing; Daniel S Pine, Resources, Supervision, Writing – review and editing

### Author ORCIDs

Lauren Y Atlas 🔾 http://orcid.org/0000-0001-5693-4169

### Ethics

Clinical trial registration NCT02446262.
Consent was obtained as described in Materials and methods. Data were collected under NIH protocol 15-AT-0132, identifier NCT02446262 at clinicaltrials.gov.

### Decision letter and Author response

Decision letter https://doi.org/10.7554/eLife.73353.sa1
Author response https://doi.org/10.7554/eLife.73353.sa2

## Additional files

### Supplementary files

- Supplementary file 1. Heat intensity effects on pain in fMRI participants (n = 36).
- Supplementary file 2. Bayesian multilevel model evaluating effects of Group, Cue, Phase, and Trial on medium heat pain using Beta family.
- Transparent reporting form

## Data availability

Pain, SCR, and signature data are available on OSF; neuroimaging data has been uploaded to neurovault.

The following datasets were generated:

| Author(s) | Year | Dataset title | Dataset URL | Database and Identifier |
|-----------|------|---------------|-------------|-------------------------|
| Atlas LY | 2022 | Instructions and experiential learning have similar impacts on pain and pain-related brain responses but produce dissociations in value-based reversal learning | https://neurovault.org/collections/XSHXWBEV/ | NeuroVault, XSHXWBEV |
| Atlas LY | 2022 | How instructions and learning shape pain and expectancy-based pain modulation: Pain, SCR, and brain signatures | https://osf.io/adc8e/ | Open Science Framework, adc8e |

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
