## [Editor Report]

This fundamental paper advances our understanding of the commonalities and differences in the neural basis of directly experienced and instructed aversive learning in humans. The study uses compelling experimental design and analyses along with functional magnetic resonance imaging (fMRI) to inform neuro-computational models of how explicitly informed vs experientially acquired information influences learning about cues predicting painful stimuli. This work will be of broad interest to neuroscientists interested in pain and aversive learning and memory.

---

## [Decision Letter]

**Decision letter after peer review:**

Thank you for submitting your article "Instructions and experiential learning have similar impacts on pain and pain-related brain responses but produce dissociations in value-based reversal learning" for consideration by *eLife*. Your article has been reviewed by 3 peer reviewers, one of whom is a member of our Board of Reviewing Editors, and the evaluation has been overseen by Christian Büchel as the Senior Editor. The following individual involved in review of your submission has agreed to reveal their identity: Christoph W Korn (Reviewer #3).

The reviewers have discussed their reviews with one another and feel that new analyses, clarification of methods and possibly new data are required to make a definitive assessment of the manuscript. The Reviewing Editor has drafted this to help you prepare a revised submission.

Essential revisions:

1) It appears that a limited number of subjects were used in the study and it is unclear from the description of the methods whether this is sufficient to draw strong conclusions. The authors need to address this point.

2) It is not clear how to interpret the findings that some brain regions don't follow reversals but maintain initial contingencies. The authors should explore in more detail a) what stimuli are actually driving these changes (e.g. CS, US, etc), b) possibly incorporate autonomic indices into the analyses to help clarify this and c) provide a better explanation of the methods and analyses used so that the reader can better understand them and interpret the findings.

3) The authors trained subjects with high and low pain intensities and then tested with a medium intensity stimulus to control for objective differences. The study would be improved by including an analysis which incorporates the different shock intensities particularly for fitting the Reinforcement Learning models.

*Reviewer #1 (Recommendations for the authors):*

1) Although it appears that path b effects refer to brain activity associations with pain ratings, this should be more explicitly defined in the paper (possibly where it is introduced on line 668). For example, how was cue controlled for, how were reversals incorporated?

2) The authors show that there are differences in the brain regions mediating instructed vs. directly experienced learning (Figure 6). However, the regions responsible for this difference differ from those discovered in the senior author's previous work (Atlas et al. *eLife* 2016). The authors should discuss the reasons for these differences.

3) It is not clear whether their imaging approach can temporally differentiate cue from heat pain. The authors should discuss the details of what could be represented in the bold response (including analysis periods during cue and shock, temporal dynamics of the bold response, etc.). If the bold response to cues cannot be distinguished from heat stimulation this could explain differences between cortical and subcortical structures in how they reverse (or do not reverse) their representations when contingencies change. Furthermore, because the Atlas et al. 2016 paper used a design in which some cues were not followed by aversive stimulation, the differences they see in the representations here could be due to that.

4) The paper could be strengthened if the authors incorporated the autonomic measures which they said they don't report these here because they will use them in another paper. This may allow them to determine whether the subcortical regions which don't reverse are more closely related to autonomic indices of learning (rather than self-report), as they suggest in the introduction, and whether the differences between self-report and autonomic measures explain differences in the brain regions engaged during instructed vs. experiential learning between this paper and the group's past work.

*Reviewer #2 (Recommendations for the authors):*

1. The sample size of N = 20 in each of the two groups seems rather low and raises the question whether the study was sufficiently powered. The small sample size is of particular concern because due to a programming error, data of only two instead of three reversals could be used in half of the instructed group (N = 10). It is mentioned in the methods section that the sample size is based on power analyses from previous studies but these studies are not further specified or referenced. The authors also state that the sample size is "deemed sufficient based on a behavioural pilot experiment" and refer to their clinical protocol registered at clinicaltrials.gov. Unfortunately, I was unable to find information about the pilot study, sample size and sample size calculation in the preregistration that would justify the low N. The site only seems to mention a sample size of N = 400 for a set of five studies but no breakdown for single studies. Please add this important information to the main manuscript.

2. In contrast to the very detailed and complex methods and Results sections, the discussion of the findings is very cautious and largely descriptive. I appreciate that the authors want to stay close to the data in their discussion, but the most relevant and surprising findings could be explored in more detail to add depth. As a minimum, the authors could offer potential explanations for their findings which could be tested in future studies. Please find below four examples of surprising findings that would benefit from deeper exploration/discussion:

– The PAG kept responding to initial contingencies following reversals. How does this relate to the role of this structure in fear learning and descending pain modulation?

– As evident from Figure 2C, cue effects show a sharp increase from trial 2 to trial 3 whereas trial 1 and 2 are almost identical. This pattern can be seen in both groups but seems to be more prominent in the instructed group. How do the authors interpret the difference between reversals and the difference between groups?

– The fact that pain ratings increased over time only in the instructed group is surprising. How does this relate to other findings of this group?

– Effects of predictive cues and reversals on pain-related brain responses are not significantly different between group, although pain ratings indicate a stronger effect in the instructed group (page 30). Please discuss this dissociation in more detail.

3. As mentioned in the introduction, the authors reported sustained amygdala engagement in response to original cues in one of their previous papers, indicating that this structure might maintain a representation of original contingencies. In the present study the amygdala does not seems to show this pattern. How do the authors explain this difference in findings? How are reversals in the current paradigm different from those previously studied?

4. Although peripheral-physiological data including pupil dilation, gaze position and EDA were recorded, they are not included in the manuscript because "they may be analyzed and reported separately in future work". This decision seems unfortunate as this data could potentially aid the more detailed interpretation of the findings (see point 2).

5. The instructions that were given to the two groups differ in one key aspect that is not discussed in the manuscript. While the instructed group was only informed about the meaning of the cues ("It is there just to let you know what level of heat will be next"), the uninstructed group was given a specific task ("Your job is to pay attention to the cues and try to figure out the relationship between the sounds that you hear and the heat that you feel."). To what extent could this task difference have influenced the results? Related to this issue, it is also worth noting that explicit information about reversals in the instructed group provided orientation to those who had lost track of contingencies because at least at the time point of the reversal they could be sure about the correct contingency. In contrast, the uninstructed group was not given an external reference throughout the whole experiment.

6. If I understand the methods section correctly, the authors used a 100% reinforcement rate during the conditioning phase. In the subsequent test phase, each cue was followed by medium pain and reinforcement was lowered to 50%. Were participants informed about the change in reinforcement rate or was there a break between conditioning and test phase? Please clarify.

7. The figures are very small and difficult to read. Please particularly check the violin plots in Figure 4 and 5 where parts of the figure legend seem to be missing.

*Reviewer #3 (Recommendations for the authors):*

Below, I list a few suggestions for improvement.

1. Focus on stimuli of medium intensity: The authors nicely control for objective pain intensity by focusing on analyzing pain stimuli of medium intensity. However, I would find it very interesting if the reported pattern of results – especially the fit of the RW model variants – holds if stimuli of all intensities are included in the analyses.

a. Specifically, in the uninstructed group, the exact temporal succession of stimuli of high and low intensity should inform stimulus-by-stimulus learning. Maybe I just did not grasp correctly, which trials were included for fitting.

b. As a sanity check, it would be nice to see the ratings and also the fMRI signals for all received intensities. I understand that the medium intensity stimuli control for objective differences – but still it would be nice to see the reflection of these objective differences in behavior and fMRI signals.

c. By including all stimuli, the authors could also look into prediction errors (PEs) and into the expectations during the cue phase. The authors note that the study design was not optimized for such analyses and that PEs and expectations are negatively correlated. But I am wondering if this profoundly precludes such analyses – even though these would be exploratory.

2. Physiological data: The authors note that they might look into those data in the future. A slightly more nuanced explanation of the reasons for refraining from analyzing the physiological data might be helpful (especially given that the authors have experience with such analyses).

3. Brain responses that are unaffected by reversals: In my view, this is an interesting an unexpected finding (see e.g., Figure 5). Can it be excluded that these signals do not "just" relate to the differences between the cues in a rather unspecific way? Can it be that the first reversal induces a second "context" and that there are brain signals (maybe in the same regions) that "remain stable" in this new "switched context?" Relatedly, the authors might want to conjecture whether such "stable" responses would become unstable with longer learning periods in the switched context. I am rather unfamiliar with the role of these regions for pain perception.

4. Learning rates of the instructed group: I find the wording regarding the learning rate a bit confusing. It sounds a bit as if the authors did not expect the learning rates to be low (or even zero). But given the explicit instructions and the nice RW model variant, this is exactly how they should look. The instructions say everything and no learning is needed. Testing instructions that are incorrect (as might be the case for some types of placebo/nocebo instructions) would be really interesting here; i.e., participants get information but still have to learn from experience (in spite of the received information). Maybe the authors have that idea in their mind when reporting their findings about the learning rates of the instructed group.

---

## [Author Response]

Essential revisions:1) It appears that a limited number of subjects were used in the study and it is unclear from the description of the methods whether this is sufficient to draw strong conclusions. The authors need to address this point.

We now include further information on our power analyses and choice of sample size. Please see Response 2.1 below.

2) It is not clear how to interpret the findings that some brain regions don't follow reversals but maintain initial contingencies. The authors should explore in more detail a) what stimuli are actually driving these changes (e.g. CS, US, etc), b) possibly incorporate autonomic indices into the analyses to help clarify this and c) provide a better explanation of the methods and analyses used so that the reader can better understand them and interpret the findings.

We will briefly address each of these editorial suggestions, which are covered in greater detail in individual responses below.

a) To determine which stimuli are driving responses, we provide additional data to distinguish responses to the CS (visual cue) from responses to the US (i.e. heat delivery) in Figure 5 —figure supplements 3 and 5 and Figure 6 —figure supplement 3, although we acknowledge some constraints due to task timing and address these in our revised Discussion. Based on the timecourse of extracted data within regions identified as showing differential cue effects (i.e. Path A in either mediation analysis), differential responses emerge at the time of thermal stimulation, rather than in response to the cue. The only regions that seem to be sensitive to cue during the anticipatory interval are the amygdala regions of interest (ROIs). In our revised discussion, we now address the role of the amygdala and encourage future studies to specifically evaluate effects of instructions on amygdala responses to both the CS and US. Thus while it is the integration of CS and US that generates value signals and updates based on prediction errors, we believe the most important differences based on expectancy are present at the time of heat stimulation / US outcome. We address this further below in response to points 1.2 and 1.3.

We have also examined regions that were identified as maintaining original contingencies in more detail by inspecting timecourses and trial-wise responses in these regions (see Figure 6 —figure supplements 2 and 3). Of regions identified in these analyses, only the rostroventral medulla (RVM) shows clear sensitivity to initial contingencies on both original and reversed runs in both groups across participants. We also observe heterogeneity within the VMPFC (i.e. some portions that reverse, and others that do not), which we address in the discussion. For additional discussion on findings in the RVM, please see response 2.2.

b) We have also incorporated autonomic indices (skin conductance [SCR], pupil dilation) as suggested by all three reviewers. We focus primarily on SCR, as we only had useable pupillary data from 20 participants. Quantitative models fit to SCR were largely consistent with models fit to pain ratings, and analyses of anticipatory SCR were largely inconclusive. Specific results are discussed below in response 1.4.

c) We have provided more detail on the methods and analyses throughout the manuscript, as discussed in individual responses below. We hope this clarifies the details of our analyses and aids interpretation.

3) The authors trained subjects with high and low pain intensities and then tested with a medium intensity stimulus to control for objective differences. The study would be improved by including an analysis which incorporates the different shock intensities particularly for fitting the Reinforcement Learning models.

We have clarified in our revision that all heat intensities are included in the reinforcement learning model but that the model’s goodness-of-fit is based on the medium trials since these trials would show the strongest modulation by expected value, as the stimulus intensity is fixed. these are modified strictly by expected value. In addition, we evaluated a model in which goodness-of-fit was evaluated across all temperatures. Findings were largely consistent with models fit to medium heat trials alone. Please see Response 3.1 for complete information.

Reviewer #1 (Recommendations for the authors):1) Although it appears that path b effects refer to brain activity associations with pain ratings, this should be more explicitly defined in the paper (possibly where it is introduced on line 668). For example, how was cue controlled for, how were reversals incorporated?

The Reviewer is correct that Path *b* effects refer to brain activity associations with pain ratings when controlling for cue, which is different from a standard brain-behavior analysis. This is done in the context of mediation, which simultaneously evaluates partial regression for all paths. We did not initially include this information because it has been discussed in previous papers using this approach, but we have provided additional information about Path *b* on page 56 as suggested by the reviewer.

Reversals are incorporated as a function of how we modeled cue, i.e. in the first mediation model we assume cue values reverse entirely upon reversal whereas in the second mediation model we model Cue as the X variable and include reversals as a covariate. In both cases, Path *b* evaluates positive associations with subjective pain controlling for cue. We note that we also control for temperature as we evaluate associations only within medium heat trials.

To clarify this point, we provided additional information in the Methods section regarding multilevel mediation and how each path is computed. We made the following changes (additions in red):

– P. 18: “Path *b* identified voxels that were associated with subjective pain while controlling for cue (see Methods).”

– P. 55: “Mediation was implemented by the Matlab function *mediation.m* (https://canlab.github.io/). Cue was included as the input variable (i.e. X; coded as 1 for High cue, -1 for Low cue), pain was included as the output variable (i.e. Y), and we searched for potential mediators. Voxelwise mediation, or mediation effect parametric mapping (Wager et al., 2009; Atlas et al., 2010) yields interpretable maps that are similar to simultaneous partial regressions, although implemented using mixed effects. The mediation effect (a*b) identifies regions whose activity contributes to variance in the effect of the independent variable on the dependent variable (Path *c*; for individual *i*, trial *j*: (1) Y_ij_ = d_0j_ + **c**_j_X_ij_ + e_0ij_). Path *a* denotes the effect of the input variable on the potential mediator (M; brain activation in a given voxel), thereby representing cue effects on brain responses to medium heat (i.e. (2) M_ij_ = d_1j_ + **a**_j_X_ij_ + e_1ij_). Path *b* measures the association between the mediator and outcome, controlling for the input variable. Here, this represents brain regions that predict pain, controlling for cue type (i.e. (3) Y_ij_ = d_2j_ + c_j_' X_ij_ + **b**_i_ M_ij_ + e_2ij_). In multilevel mediation, the difference between the total effect (Path *c*: the effect of cues on subjective pain) and the direct effect (Path c`: the effect of cues on subjective pain when controlling for the mediator; see equation 3) is equivalent to the sum of the product of Path *a* and Path *b* coefficients and their covariance (Shrout and Bolger, 2002; Kenny et al., 2003). Our mixed models included intercepts (d_0-2_) and error terms (e_0-2_) for each individual. Group level estimates are computed by treating person-level intercepts and slopes as random effects. For complete details, see Atlas et al., 2010.”

– P. 56: “We focus in particular on effects of current contingencies (i.e. Cue x Phase interactions) on pain to capture the behavioral reversals we observed (Figure 4). In this main analysis, we coded the input variable (X) as the contrast between current high cues (i.e. original high cues during original contingencies and original low cues during reversed contingencies both coded as 1) and current low cues (i.e. original low cues during original contingencies and original high cues during reversed contingencies both coded as -1). Our second mediation searched for mediators of original cue effects (i.e. original high cue coded as 1, original low cue coded as -1) and controlled for current contingencies by including the X variable from the current cue contingencies mediation (i.e. the Cue x Phase interaction) as a covariate in all paths. These effects are thus most likely to be driven by responses during the reversed runs. In both mediations, Path *b* evaluates the link between brain activation and pain, while controlling for the cue (see equation 2 above).”

2) The authors show that there are differences in the brain regions mediating instructed vs. directly experienced learning (Figure 6). However, the regions responsible for this difference differ from those discovered in the senior author's previous work (Atlas et al. eLife 2016). The authors should discuss the reasons for these differences.

Thank you for encouraging us to compare our current findings with our previous work in the domain of fear conditioning. We note that there are both similarities and differences with our previous findings. As noted in point 1.3 below, the biggest difference is that our previous work studied responses to the CS on unreinforced trials, whereas our current study tests responses to the painful outcome itself. Thus our 2016 study isolates mechanisms of aversive learning, but this study tests the consequences of learning on perception and pain-related responses. We think both are important for the fields of learning, predictive coding, pain, and perception. We have been more explicit about this point on pages 34 and 37.

Despite this methodological and theoretical difference, we observe both similarities and differences between the present study and our prior work. In both studies, instructions altered value-based processing in the striatum and OFC/VMPFC, as well as the DLPFC, rACC, ACC, and insula, key nodes implicated in pain modulation. We have added additional discussion to highlight the similarities, as this provides further evidence that the striatum and VMPFC/OFC reverse based on instruction, consistent with our previous work and several studies of appetitive learning.

While our initial submission was agnostic as to the role of the amygdala in instructed or uninstructed learning, the additional analysis of unsigned prediction error suggested by Reviewer 3 (see point 3.1.c) revealed new findings indicating that there were associations between unsigned PE and amygdala responses to heat in both groups, including the Instructed Group, in which the timecourse of learning updated with instructions. This is distinct from our previous work, in which we only examined EV (since EV was correlated with PE) and extends findings from other studies that indicate unsigned PEs in the amygdala in the absence of instruction (Boll et al., 2012). We now discuss these differences in depth on pp. 36-37.

We have expanded our discussion on the similarities and differences between these studies and implications for our understanding of the amygdala’s response to instructions in the revised discussion:

– Pp. 34-37, added: “In addition to regions implicated in prior studies of placebo and pain modulation, we evaluated the contribution of systems involved in value-based learning, namely the striatum, OFC/VMPFC, and the amygdala. We previously observed dissociations in these regions during aversive learning (Atlas et al., 2016), such that responses in the striatum and OFC/VMPFC tracked EV that updated with instructions, whereas the right amygdala was associated with EV based purely on experiential learning. The present study builds on this work by focusing heat-evoked responses (i.e. the US in a fear conditioning model), rather than anticipation (i.e. responses to the CS), which allows us to test whether error-based learning shapes responses to noxious stimuli and whether this learning is modulated by instructions. ROI-based analyses indicated that the bilateral striatum was associated with EV that updated based on instruction in the Instructed Group, consistent with our previous findings (Atlas et al., 2013). VMPFC/OFC associations with value were also sensitive to instructions, although we observed different patterns across the broad region and did not observe consistent findings in ROI-based or corrected analyses. For instance, uncorrected voxelwise analyses revealed that the association between EV and VMPFC activation differed between groups (positive associations with EV in the Instructed Group and negative associations with EV in the Uninstructed Group), regardless of whether models were fit to pain, heat-evoked SCR, or anticipatory SCR. In contrast, we observed negative associations with EV in the left lateral OFC in both groups. Both portions of the VMPFC/OFC have been implicated in previous studies of placebo analgesia (Lieberman et al., 2004; Wager, 2004; Petrovic et al., 2010; Ellingsen et al., 2013), and we previously showed that pain-predictive cues modulate responses in both regions, with greater activation when medium heat is paired with low pain cues than high pain cue (Atlas et al., 2010). Lateral OFC is thought to evaluate value of independent options, whereas medial OFC and VMPFC have been linked to value-based choice for the purpose of comparison (Rudebeck and Murray, 2014). We recently showed that surgical lesions of the bilateral VMPFC enhance instructed cue effects on pain in a stable environment (Motzkin et al., 2021) additional studies that use lesion symptom mapping in heterogeneous frontal samples or measure aversive learning in nonhuman primates can provide further insights on the contribution of different portions of the VMPFC/OFC to pain modulation and instructed learning. Together, these findings build on previous work indicating that the striatum and VMPFC update upon instruction in both appetitive and aversive learning (Li et al., 2011a; Atlas et al., 2016) and that OFC value signals are sensitive to higher order knowledge across species (Wilson et al., 2014; Lucantonio et al., 2015; Schuck et al., 2018).

Because we focused on responses to the US rather than the CS here, we also were able to extend previous work by examining the role of unsigned prediction errors (PEs), which are linked to associability, which gates attention and learning dynamically. Associability has been linked to the amygdala in previous studies of pain and aversive learning (Li et al., 2011; Zhang et al., 2016; Atlas et al., 2019), meta-analyses indicate that unsigned PEs are associated with activation in the dorsal and ventral striatum, as well as other regions (Corlett et al., 2022), and one study of aversive learning showed responses to both associability and unsigned PE in the amygdala (Boll et al., 2012). Consistent with these studies, we observed associations between unsigned PE and activation in the bilateral amygdala and the right striatum in both groups in ROI-based analyses. Whole brain analyses revealed associations between unsigned PEs and activation in the right hemisphere encompassing the insula, striatum, and amygdala, and that the associations between PE and activation in the right SII and temporal cortex differed across groups (negative associations in the Instructed Group and positive in the Uninstructed Group). Thus, although cue effects on subjective pain were mediated similarly across groups, we still observe unique responses to PE that differed between groups.

We note that the amygdala’s association with PE within the Instructed Group indicates that instructions do have some impact on amygdala activation, seemingly in contrast to our previous work on instructed learning during fear conditioning (Atlas et al., 2016; Atlas 2019). However, because error-driven learning was nearly absent between instructed reversals in the Instructed Group, the PE regressor in this group captures a constant effect that decreases linearly over time, consistent with prior evidence that amygdala responses habituate during aversive learning (Büchel et al., 1998). Thus we did not observe evidence of amygdala sensitivity to instructions per se in the present study. The absence of pure experiential learning signals in the right amygdala may reflect the present study’s focus on BOLD responses to noxious heat (i.e. unconditioned stimuli), in contrast to our previous study (Atlas et al., 2016) which focused on responses to unreinforced cues (i.e. conditioned stimuli).” [See additional edits provided in response to point 1.3, below]

3) It is not clear whether their imaging approach can temporally differentiate cue from heat pain. The authors should discuss the details of what could be represented in the bold response (including analysis periods during cue and shock, temporal dynamics of the bold response, etc.). If the bold response to cues cannot be distinguished from heat stimulation this could explain differences between cortical and subcortical structures in how they reverse (or do not reverse) their representations when contingencies change. Furthermore, because the Atlas et al. 2016 paper used a design in which some cues were not followed by aversive stimulation, the differences they see in the representations here could be due to that.

We agree with the Reviewer and have acknowledged the difference in study design (i.e. focus on US vs focus on CS; lack of intermittent reinforcement) as a likely reason for why we see differences between the previous work and the present study, as we discuss above in point 1.2. We use a slow event related analysis to facilitate single trial analysis and attempt to separate responses to heat from responses to cues but cannot rule out the possibility that anticipatory processes might contribute to differences observed during heat, since we use a fixed anticipatory interval. We chose not to jitter the duration of anticipation because previous studies indicate that longer anticipation periods are associated with increased dread, which can in turn modulate responses to aversive outcomes (Berns et al., 2006).

To explore whether anticipatory responses contributed to our findings, we extracted raw timecourses for regions identified as showing cue-based differences in responses to medium heat (i.e. Path a in either mediation analysis) as well as a priori regions of interest. Most regions that were identified as showing cue-based modulation during medium heat showed differences based on cue only after heat onset, and usually late in the heat period (see Figure 5 —figure supplement 3 and Figure 6 —figure supplement 3). Within value-related ROIs (Figure 5 —figure supplement 5), cue-based differences in the striatum and VMPFC emerge at the time of heat, rather than cue, whereas the amygdala shows some sensitivity to cue prior to heat onset. We therefore believe that our findings are unlikely to be driven by responses to cues per se.

We agree with the Reviewer that the focus on heat rather than cue, as well as the intermittent reinforcement used in our prior paper, can account for some of our findings, as we discuss in revisions provided in response 1.2 above. We made the following additional changes to address this point in the revision:

– *Supplemental Figures:* Added raw timecourses relative to cue onset for regions that showed differential responses to determine specificity to heat vs anticipation.

– *P.* 37, *added*: “In addition, our experimental design was optimized for analyses of the heat period rather than the cue (i.e. we used a fixed anticipation period to avoid variations in dread that might accompany jittered anticipatory intervals (Berns et al., 2006)), so we were unable to separately analyze responses to pain-predictive cues to determine whether the amygdala showed similar responses in the present study. We note that raw timecourses suggest that cue-based differences in the amygdala may emerge in response to predictive cues, as well as during the heat period (see Figure 5 —figure supplement 5). This contrasts with the bilateral striatum and nearly all regions identified in mediation analyses, in which cue-based differences do not emerge until the heat period. Future work should incorporate longer delays between cues and aversive outcomes or utilize intermittent reinforcement (e.g. catch trials without heat presentation) to separately analyze responses to cues and noxious heat in the amgydala, VMPFC, and other key regions.”

4) The paper could be strengthened if the authors incorporated the autonomic measures which they said they don't report these here because they will use them in another paper. This may allow them to determine whether the subcortical regions which don't reverse are more closely related to autonomic indices of learning (rather than self-report), as they suggest in the introduction, and whether the differences between self-report and autonomic measures explain differences in the brain regions engaged during instructed vs. experiential learning between this paper and the group's past work.

We note that our initial decision to omit analyses of concurrent autonomic data was informed by initial editorial comments that the paper risked being too complicated. Nonetheless, we have analyzed autonomic data that was collected during the fMRI experiment based on the Reviewers’ and Editors’ suggestions.

*Pupil dilation:* Following quality control assessment, we had useable pupillometry data from 20 participants (9 in Instructed Group) and an average of 54.4% of trials per participant. We attribute the limited useable data to technical issues, as our scanner’s head coil obscured the eye for many participants. Nonetheless, for participants with useable data, we computed baseline-corrected mean pupil dilation (PD) and area under the curve for each trial, and analyzed effects of Temperature, Cue, Phase, and Group on pupillary response during the heat period. Both outcomes were significantly influenced by Temperature across all trials, but there were no influences of any experimental factors, and effects of Temperature were insufficient to reject the null hypothesis based on Bayesian models (see Figure 2 – Source data 1). We report these findings in the revised manuscript on page 10, as delineated below, but did not analyze PD further due to power limitations.

*Skin conductance responses:* To analyze skin conductance data, we used Ledalab’s continuous decomposition analysis (CDA), which accounts for phasic and tonic signals and can capture multiple responses during the 8-second heat period (Benedek and Kaernbach, 2010). We focused on responses that occurred between heat onset and 4s after offset to capture heat-evoked responses, and responses between cue-onset and 4s later as anticipatory responses. We used the average phasic driver (“SCR”) in all trial-wise analyses. Analyses were limited to participants who had greater than 4 SCR responses for each analysis and thus sample sizes varied across analyses.

Thirty-six participants were included in analyses across temperatures (18 in Instructed Group). We observed significant effects of Temperature and all experimental factors on heat-evoked SCR (see Figure 2 and Figure 2 – Source data 1). We therefore analyzed effects of Cue and Phase on responses to medium heat. Thirty participants were included in analyses (13 in Instructed Group). Heat-evoked SCRs were influenced by predictive cues on medium trials and reversed as contingencies changed, but the magnitude of these differences did not differ by Group (see Figure 4 and Figure 4 – Source data 1), in contrast to subjective pain ratings, which showed larger reversals in the Instructed Group. These effects were statistically significant, but of undecided significance in Bayesian models. We report these results in the revised manuscript (see below for specific changes). Because only 13 Instructed Group participants had variations in heat-evoked SCR on medium heat trials, we take these results with caution and did not analyze associations between trial-by-trial SCR and brain responses.

We also examined anticipatory SCR by analyzing cue-evoked responses, i.e. responses following cue presentation, before heat onset. 29 participants were included in the analysis (15 Instructed Group, 14 Uninstructed Group), which included all trials since responses should be unrelated to subsequent temperature. We analyzed cue-evoked SCR as a function of Cue, Group, and Phase. Like heat-evoked SCR, the magnitude of cue-evoked SCR did not differ by group (see Figure 4 – Source data 1). However, in contrast to heat-evoked SCRs, we did not observe any impact of contingency reversals: Anticipatory responses were shaped by original contingencies consistently across time, as evidenced by a main effect of Cue without a significant interaction with Phase (all p’s > 0.8, see Figure 4 – Source data 1). However, there was not sufficient evidence to reject the null hypothesis based on Bayesian models. Although we therefore avoid making inference based on these findings, we acknowledge that this mirrors results of our secondary mediation analysis, in which cues shaped responses in the RVM but did not vary as contingencies changed.

We conducted an exploratory analysis to test for associations between trial-by-trial anticipatory SCR and subsequent brain response to medium heat. Twenty-one subjects had variation in cue-induced SCR on medium trials and were included in this analysis (11 in Instructed Group); we therefore did not explore potential group differences due to the low sample sizes. We did not observe any association between anticipatory SCR and subsequent heat-evoked activation in value-related ROIs, pain signature patterns, or within regions involved in pain and placebo. Whole brain correction did reveal one cluster of activation in left SII, and we observed additional regions in uncorrected voxelwise results, including positive associations in the right ventral striatum (but no other subcortical regions); uncorrected results are presented within Author response table 1. Due to the limited sample size and failure to observe any associations in a priori regions of interest or corrected analyses, as well as the fact that Bayesian analyses supported the null hypothesis of no effect of experimental factors on anticipatory arousal, we have decided not to include this analysis in the revision.

**Author response table 1. sa2table1:** 

Contrast	Region	x	y	z	voxels	Volume (mm^3^)
Positive association with anticipatory SCR	R Caudate Nucleus (Nucleus Accumbens)	10	10	-8	6	162
Negative association with anticipatory SCR	R Superior Temporal Gyrus (Area TE 3)	62	-10	-4	104	2808
	L operculum	-56	14	-2	32	864
	R Middle Occipital Gyrus (Area hOc4la)	46	-74	4	83	2241
	L Superior Temporal Gyrus (Area TE 3)	-62	-32	16	200	5400
	L Insula Lobe (Area Ig2)	-38	-20	14	33	891
	R Superior Temporal Gyrus	50	-40	14	25	675
	L Postcentral Gyrus (Area 3b)	-44	-16	46	44	1188
	L dACC	-8	-4	44	19	513
	R Postcentral Gyrus (Area 4p)	34	-28	46	15	405
	L Posterior-Medial Frontal	-2	-8	68	38	1026

*Fitting models to SCR:* While the sparse SCR data limited our ability to directly measure associations between SCR and brain activation, our quantitative models may be less impacted by the missing data as our iterative jackknife approach leverages responses across all participants. We therefore fit quantitative models to (a) heat-evoked SCR on medium heat trials and (b) cue-evoked SCR on all trials. We included all participants in the analysis and used a jack-knife approach with the same model reported in the main manuscript.

Results from fits to heat-evoked SCR were largely consistent with our findings when we fit models to pain reports: Expected value updated upon instruction in the Instructed Group (*M_rho_* = 0.79, *SD* = 0.024) without any additional learning as a function of stimulus outcome pairings (*M_alpha_* = 0; *SD* = 0), and learning rates were higher in Uninstructed Participants (Uninstructed Group: *M_alpha_* = 0.40, *SD* = 0.09; t(36) = -98.43, p <.001), whereas the instructed reversal parameter was higher in the Instructed Group (Uninstructed Group: *M_rho_* = 0.09, *SD* = 0.04; t(36) = 64.48, p <.001). Next, we measured the association between brain responses to medium intensity heat and the dynamic timecourse of expected value (EV) based on the model. We observed group differences in associations with EV in the DMPFC, overlapping with the region identified when we fit models to pain reports (see Figure 8 —figure supplement 2). Furthermore, although there were no other regions in which group differences or main effects survived multiple comparisons correction, uncorrected results were nearly identical to the regions that showed group differences and main effects when we fit quantitative models to pain ratings. Thus fitting quantitative models to heat-evoked SCR did not reveal any meaningful findings beyond what we discovered based on models fit to pain reports. We now briefly report these results in the manuscript’s supplementary materials.

We also fit the same models to cue-evoked SCR, under the assumption that anticipatory arousal indexes EV on a given trial. There were no group differences in the instructed reversal parameter (Instructed Group: *M_rho_* = 0.23, *SD* = 0.13; Uninstructed Group: *M_rho_* = 0.28, *SD* = 0.15; p > 0.2) but a significant difference in learning rates again driven by faster learning in Uninstructed Group participants (Instructed Group: *M_alpha_* = 0.33, *SD* = 0.44; Uninstructed Group: *M_alpha_* = 0.79, *SD* = 0.42; t(36) = -3.28, p = .002). We searched for regions that displayed correlations with EV based on fits to cue-evoked SCR. Although no regions survived multiple comparisons correction whether we searched for consistent associations across groups or group differences in associations with EV, we again observed largely overlapping findings in uncorrected analyses, as shown in Figure 8 —figure supplement 2. All three models revealed group differences in the VMPFC and the left putamen, regardless of whether EV was fit to pain, heat-evoked SCR, or anticipatory SCR. We report these findings in supplementary materials and include them in the discussion, as they mirror our previous findings in which learning-related responses in the striatum and OFC updated with instruction (Atlas et al., 2016) and builds on similar findings in reward learning (Li et al., 2011, Doll et al., 2009).

In summary, our results using heat-evoked SCR as an outcome in quantitative models are largely consistent with dynamic models fit to pain, while analyses of cue-evoked SCR suggest preliminary support that anticipatory arousal is sensitive to initial contingencies throughout the task, but are not sufficient to reject the null hypothesis. Interestingly, our investigation of SCR did not provide additional insights into the amygdala or other subcortical regions. However, the additional analysis of unsigned prediction error does provide further insight on the role of the amygdala (see Response 3.1.c, below).

We have reported the full results of these analyses here for Reviewers for full transparency in response to all three Reviewers’ suggestions and editorial comments and have included results of models fit to SCR in Supplementary Materials. We take these findings as highly exploratory, however, and want to be quite cautious with interpreting these results, as (a) the number of participants that showed measurable SCR in either period is limited, particularly in the Instructed Group; (b) Bayesian analyses indicate that these effects have limited practical significance; (c) Ledalab’s continuous decomposition analysis is only one of many different approaches to measuring SCR, and we are now formally comparing different approaches across datasets to identify the most sensitive measure to changes in noxious stimulation and associations with pain, which will be reported in a separate paper.

We made the following changes:

– Updated Figures 2 and 4 to include effects on heat-evoked SCR

– Updated Figure 8 —figure supplements 1, 2, 4, and 5 and associated Source Data to report associations between brain activation and computational models fit to SCR

– P. 8, additions in red: “Heat intensity effects on pain, autonomic responses, and brain responses to noxious heat are similar across groups.”

– P. 10, added: “Next, we analyzed heat-evoked autonomic responses during the experiment. SCR and pupil dilation were both significantly influenced by Heat Intensity and exhibited Heat Intensity x Cue X Phase interactions (see Figure 2 – Source Data 1). Both factors had practically significant effects on SCR (< 1% in ROPE), whereas Bayesian analyses of pupillary outcomes indicated that evidence was not sufficient to reject the null hypothesis (100% in ROPE). Because there was no meaningful effect of temperature on pupil dilation and the number of subjects with useable pupil data was substantially less than those with useable skin conductance, we focused on SCR in subsequent analyses of cue effects on physiological arousal. There was no main effect of Group on pupil dilation or SCR, nor any interactions between Group and Heat Intensity for either outcome, suggesting that temperature effects on physiological arousal were similar regardless of whether individuals were instructed about contingencies (see Figure 2B and Figure 2 – Source Data 1). For complete results, see Figure 2 – Source Data 1.”

– P. 14, additions in red: “Cue-based expectations and cue effects on pain and SCR update as contingencies reverse.”

– Pp. 15-16, added: “We also tested whether cues and reversals impacted physiological responses to medium heat, as measured by heat-evoked SCR. Heat-evoked SCRs were influenced by predictive cues on medium trials and reversed as contingencies changed, but the magnitude of these differences did not differ by Group (see Figure 4D and Figure 4 – Source data 1). While effects were statistically significant based on frequentist models, they were not sufficient to reject the null hypothesis of no difference based on Bayesian models (see Figure 4 – Source data 1). Importantly, only 13 Instructed Group participants had variations in heat-evoked SCR on medium heat trials and were included in analyses; we therefore take these results with caution and did not analyze associations between trial-by-trial SCR and brain responses. We also evaluated cue effects on anticipatory SCR, i.e. responses to the cue in the interval prior to heat stimulation, in exploratory analyses. In contrast to other outcomes, anticipatory arousal was associated with a main effect of Cue and a significant Group x Phase interaction (see Figure 4 – Source data 1), but we did not observe any interactions between Cue and Phase, suggesting that anticipatory responses did not vary as contingencies change. However, analyses were limited to 29 participants and Bayesian analyses indicated that the data support the null hypothesis of no effect, and thus we do not make inference based on anticipatory arousal.”

– Pp. 26-27, added: “Fitting models to heat-evoked autonomic responses revealed similar patterns of activation based on whole-brain and ROI-based correction (see Figure 8 —figure supplement 4 and Figure 8 – Source Data 3), and while group differences did not survive correction when we fit models to anticipatory SCR (see Figure 8 —figure supplement 5 and Figure 8 – Source Data 4), we observed group differences in associations with EV in overlapping portions of the VMPFC and left putamen at uncorrected thresholds in all three models (see Figure 8 —figure supplement 2 and Figure 8 – Source Data 2-4). For complete results of models fit to SCR, see Figure 8 —figure supplements 4-5 and Figure 8 – Source Data 3-4.”

– P. 35, added: “uncorrected voxelwise analyses revealed that the association between EV and VMPFC activation differed between groups (positive associations with EV in the Instructed Group and negative associations with EV in the Uninstructed Group), regardless of whether models were fit to pain, heat-evoked SCR, or anticipatory SCR.”

– P. 39, added: “In exploratory analyses, we measured the association between anticipatory responses to the cues themselves and brain responses to noxious heat, however we did not observe associations between anticipatory arousal and responses within a priori networks. We also found that quantitative models fit to heat-evoked and anticipatory SCR were highly similar to models fit to subjective pain. However, as the number of participants with useable skin conductance or pupil dilation data was extremely limited due to technical malfunctions and variations in arousal, we consider these findings exploratory and do not make strong claims based on these results. Future studies should continue to compare the effects of learning, instructions, and expectations on subjective pain with effects on physiological outcome measures to determine whether autonomic responses to noxious stimuli are shaped by pain decision-making (Mischkowski et al., 2019) or whether physiological responses are shaped through independent pathways, e.g. through a dual process model (Ohman and Soares, 1993; Mineka and Ohman, 2002).”

– P. 42, added: “EDA was collected using Biopac’s EDA100C-MRI module (Biopac Systems, Inc, Goleta, CA). For each participant, two pre-gelled EDA electrodes (EL509; Biopac Systems, Inc, Goleta, CA) were prepared and applied to the hypothenar muscles of the left hand.”

– Pp. 48-49, added: “Psychophysiological data processing. Skin conductance data was preprocessed in AcqKnowledge (Biopac Systems, Inc, Goleta, CA). During preprocessing, data were smoothed (1000-sample Gaussian smoothing kernel) and filtered (25-Hz FIR low-pass filter) in AcqKnowledge, then imported into MATLAB and downsampled to 250 Hz. Data were then analyzed using Ledalab’s continuous decomposition analysis (CDA), which accounts for phasic and tonic signals and can capture multiple responses during the 8-second heat period (Benedek and Kaernbach, 2010). We analyzed both (a) cue-evoked anticipatory responses that occurred within 4s following cue presentation; and (b) responses that occurred between heat onset and 4s after heat offset. We used the average phasic driver (“SCR”) in trial-wise analyses. Participants were included in analyses if they had >4 trials with measurable SCR. Thirty-six participants were included in analyses across temperatures (18 in Instructed Group); 30 participants were included in analyses of responses to medium heat (13 in Instructed Group); and 29 participants were included in analyses of anticipatory responses (15 in Instructed Group).

Pupillometry data were processed in MATLAB using both publicly available software and custom code. Data were imported to MATLAB and blinks were interpolated using the “GazeVisToolbox” (available at https://github.com/djangraw/GazeVisToolbox; (Jangraw et al., 2014)). Consistent with our previous work (Mischkowski et al., 2019), we interpolated from 100ms prior to each blink to 100ms following each blink to avoid extreme values surrounding each blink. Data were aligned to event markers and we visualized individual trials to exclude trials that were contaminated by artifacts. Subjects with fewer than seven useable trials (n = 20; 9 in Instructed Group) were excluded from analyses of pupillary data. Useable subjects had a mean of 42.9 useable trials (M = 54.4% of trials). 1000-Hz data was downsampled to 10-Hz and then we computed baseline-corrected mean pupil dilation and area-under the curve as measures of pupillary response during the heat period.”

– P. 49, additions in red: “Statistical analysis of expectations, pain, autonomic responses, and heat-evoked neural signature pattern expression. We used the statistical software R (R Core Team, 1996) to analyze effects of our experimental manipulations on expectancy ratings, pain reports, physiological arousal,”

– P. 53, added: “We used the same approach to fit computational models to anticipatory SCR and SCR evoked during medium heat simulation.”

– P. 57, added: “Our main manuscript focuses on the timecourse of EV based on fits to pain reports; we include associations based on fits to heat-evoked and anticipatory SCR in Supplementary Materials.”

Reviewer #2 (Recommendations for the authors):1. The sample size of N = 20 in each of the two groups seems rather low and raises the question whether the study was sufficiently powered. The small sample size is of particular concern because due to a programming error, data of only two instead of three reversals could be used in half of the instructed group (N = 10). It is mentioned in the methods section that the sample size is based on power analyses from previous studies but these studies are not further specified or referenced. The authors also state that the sample size is "deemed sufficient based on a behavioural pilot experiment" and refer to their clinical protocol registered at clinicaltrials.gov. Unfortunately, I was unable to find information about the pilot study, sample size and sample size calculation in the preregistration that would justify the low N. The site only seems to mention a sample size of N = 400 for a set of five studies but no breakdown for single studies. Please add this important information to the main manuscript.

We are pleased that the Reviewer took the time to review our protocol and apologize that this information was not readily available. Indeed, we state in our IRB-approved clinical protocol that all studies use a sample size of 20 per group, unless power analyses based on pilot behavioral participants indicate that the behavioral effect requires more than 20 participants to observe the effect. In Author response table 2 present the data from the pilot subjects (n = 6 per group), which suggested we need a sample of 5-9 per group to achieve adequate power to observe the behavioral difference in cue effects on pain. We therefore moved forward with 20 participants per group. We have added this information to the main manuscript:

– P. 41: “As detailed in our clinical protocol (15-AT-0132; clinicaltrials.gov identifier NCT02446262), sample size was based on power analyses from our previous studies and on a behavioral pilot experiment conducted prior to the fMRI study. We computed the effect size of cue-based differences in reported pain in a sample of 12 participants (6 per group) and determined that we need a minimum of 5-9 participants to achieve 80%-95% power to detect cue-based differences in pain, including reversals (data available upon request). We therefore included 20 participants per group in the fMRI experiment.”

**Author response table 2. sa2table2:** 

	N	LM mean (SD)	HM mean (SD)	HM-LM mean (SD)	HM & LM correlation	Standardized effect size (cohen’s d_z_)	Repeated measures effect size (d_repeated measures, pooled_)	Required sample size
All participants	12	3.05(1.17)	3.62 (1.23)	0.57 (0.36)	0.91	1.57	1.60	6 for 80% power;8 for 95% power
Instructed Group	6	3.21 (0.79)	3.87 (0.66)	0.66 (0.45)	0.82	1.47	1.52	6 for 80% power;9 for 95% power
Uninstructed Group	6	2.88 (1.53)	3.37 (1.65)	0.48 (0.26)	0.99	1.83	2.07	5 for 80% power;7 for 95% power

2. In contrast to the very detailed and complex methods and Results sections, the discussion of the findings is very cautious and largely descriptive. I appreciate that the authors want to stay close to the data in their discussion, but the most relevant and surprising findings could be explored in more detail to add depth. As a minimum, the authors could offer potential explanations for their findings which could be tested in future studies. Please find below four examples of surprising findings that would benefit from deeper exploration/discussion:– The PAG kept responding to initial contingencies following reversals. How does this relate to the role of this structure in fear learning and descending pain modulation?– As evident from Figure 2C, cue effects show a sharp increase from trial 2 to trial 3 whereas trial 1 and 2 are almost identical. This pattern can be seen in both groups but seems to be more prominent in the instructed group. How do the authors interpret the difference between reversals and the difference between groups?– The fact that pain ratings increased over time only in the instructed group is surprising. How does this relate to other findings of this group?– Effects of predictive cues and reversals on pain-related brain responses are not significantly different between group, although pain ratings indicate a stronger effect in the instructed group (page 30). Please discuss this dissociation in more detail.

Thank you for encouraging us to address these theoretical issues in greater detail in the discussion and for reflecting on all these interesting points. Because the second and third points focus only on the pre-reversal phase, and the fourth point would be highly speculative, we thought it best to focus on the maintenance of initial contingencies in the expanded discussion, which also has a number of additional points we needed to incorporate given the new analyses suggested by the reviewers.

Because we included an additional participant in fMRI analyses and used nonlinear normalization to provide better between-subject correspondence (see comparison in Author response image 1), we no longer observe the lack of reversal in PAG; instead, we observe that a portion of the brainstem near the RVM maintains initial contingencies over time. The RVM works together with the PAG to regulate endogenous opioid release, which is implicating in descending pain modulation. We now discuss this important point in the Discussion.

**Author response image 1. sa2fig1:** Mean structural (T1) image comparison – Images depict the average of all participants’ structural scans following normalization, with the initial pipeline on the left and the pipeline used for the revision, which includes nonlinear normalization, on the right. Nonlinear warping is associated with more consistent normalization, as can be seen in better gray-white matter separation in the group mean image.

We made the following changes:

– Pp. 31-32, added “While most regions that have been implicated in prior studies of placebo and expectancy-based pain modulation updated dynamically as contingencies changed, we observed sustained responses to initial contingencies in a portion of the brainstem consistent with the rostroventral medulla (RVM). RVM responses to medium heat stimuli were elevated in response to the original high pain cue relative to the original low pain cue and remained elevated throughout the task in both groups. Together with the periaqueductal gray (PAG), the RVM is a key component of the descending pain modulatory system (Fields, 2004, 2006), which regulates the release of endogenous opioids that can block ascending pain signals, a process thought to be critical for expectancy-based placebo analgesia (Levine et al., 1978; ter Riet et al., 1998; Amanzio and Benedetti, 1999). Previous neuroimaging studies have demonstrated that placebo analgesia engages the RVM (Eippert et al., 2009; Crawford et al., 2021; Yanes and Akintola, 2022) and that placebo effects on the RVM are blocked by the opioid antagonist naloxone (Eippert et al., 2009). Preclinical work demonstrates that endogenous opioid release engages long lasting pain modulation (Watkins and Mayer, 1982a, 1982b). We have hypothesized that placebo analgesia may engage endogenous opioids to lead to long lasting pain modulation, whereas pain-predictive cues may modulate pain dynamically through dopaminergic prediction errors (Atlas and Wager, 2012; Atlas, 2021). The fact that RVM responds to initial contingencies throughout the task might be consistent with such long-lasting effects, and may explain why instructions reverse placebo analgesia only after brief conditioning (Schafer et al., 2015). Interestingly, we did not observe cue-based modulation of the PAG in our analyses; however, an initial processing pipeline that used affine normalization rather than nonlinear warping (Atlas et al., 2021) indicated that the PAG also responded to initial contingencies throughout the task, similar to the RVM. Thus both opioidergic regions may be associated with long-lasting effects that do not update as contingencies change. Future studies should combine pain-predictive cues with opioid and dopamine antagonists or employ positron emission tomography to understand the contribution of endogenous opioids and dopamine to different forms of pain modulation.”

3. As mentioned in the introduction, the authors reported sustained amygdala engagement in response to original cues in one of their previous papers, indicating that this structure might maintain a representation of original contingencies. In the present study the amygdala does not seems to show this pattern. How do the authors explain this difference in findings? How are reversals in the current paradigm different from those previously studied?

This comment builds on point 1.2 above. As noted above, we believe the strongest difference has to do with our focus on heat-evoked responses, rather than anticipatory responses. In addition, we also now find evidence of unsigned prediction errors in the amygdala, including in the Instructed Group, suggesting that at least in this task, the amygdala is not entirely impervious to instruction. We discuss the open questions about laterality in the amygdala as well as its role in associability and prediction error. Please see changes provided in response to point 1.2 above for full discussion on the differences between the previous paper and the current study with regard to the amygdala.

4. Although peripheral-physiological data including pupil dilation, gaze position and EDA were recorded, they are not included in the manuscript because "they may be analyzed and reported separately in future work". This decision seems unfortunate as this data could potentially aid the more detailed interpretation of the findings (see point 2).

We thank the reviewer for suggesting we include the autonomic data here. We did not include it in our initial submission because we felt the paper was already rather complex with the different analyses, and this was confirmed in *eLife*’s initial editorial decision. Nonetheless, we have included analyses of skin conductance data and pupil dilation in the revised manuscript. Please see response 1.4 above for details.

5. The instructions that were given to the two groups differ in one key aspect that is not discussed in the manuscript. While the instructed group was only informed about the meaning of the cues ("It is there just to let you know what level of heat will be next"), the uninstructed group was given a specific task ("Your job is to pay attention to the cues and try to figure out the relationship between the sounds that you hear and the heat that you feel."). To what extent could this task difference have influenced the results? Related to this issue, it is also worth noting that explicit information about reversals in the instructed group provided orientation to those who had lost track of contingencies because at least at the time point of the reversal they could be sure about the correct contingency. In contrast, the uninstructed group was not given an external reference throughout the whole experiment.

We now address the differences in instruction and the type of attention each is likely to elicit in our revised discussion. We believe that the task instructions given to the Uninstructed Group might engage model-based learning, as they are searching for a rule that explains the relationship between the sounds they hear and the heat they feel. However, the same task can be learned strictly through model-free learning, so we cannot determine empirically whether this is the case. We also believe that studies should directly compare instructed and model-based learning, as both involve maintenance of higher order contingencies and studies point to similar circuit mechanisms (e.g. corticostriatal interactions). Finally, we believe that the impact of these differences is likely to be relatively minor in the present experiment, as we did not find any evidence of group differences in cue effects on pain, or associations between pain and brain responses to noxious heat. We have included this information in the revised discussion.

In addition, the reviewer makes a good point about the instructions leading to increases in attention at the time of reversal. However, we must clarify that the language simply stated “The relationship between the cues and the heat will now reverse” so if someone had not learned the contingencies correctly (which is admittedly less likely here given the pairings with high and low stimuli which are easily discriminable) the instructions would not tell them the correct contingency. We have clarified this in the manuscript.

We made the following changes:

– P. 45, additions in red: “Halfway through runs 2, 4, and 6, the screen displayed instructions to the Instructed Group indicating that contingencies had reversed (“The relationship between the cues and heat will now reverse”; see Figure 1C).”

– P. 38, added: “One limitation of our design is that the two groups received slightly different instructions at the start of the experiment: The Instructed Group was informed about cue-heat contingencies, whereas the Uninstructed Group was instructed to pay attention and try to figure out the relationships between cues and heat outcomes. Although these differences did not impact brain mediators of expectancy or pain, the latter instruction would be more likely to engage inference and model-based learning (Doll et al., 2012, 2015; Dayan and Berridge, 2014). These differences should be resolved in future work by directly comparing instructed and model-based learning.”

6. If I understand the methods section correctly, the authors used a 100% reinforcement rate during the conditioning phase. In the subsequent test phase, each cue was followed by medium pain and reinforcement was lowered to 50%. Were participants informed about the change in reinforcement rate or was there a break between conditioning and test phase? Please clarify.

The Reviewer is correct, but we should also clarify that there were only five conditioning trials before the medium temperatures were introduced (2-3 per temperature), so that we could measure expectancy effects on pain prior to reversal. Thus subjects were not informed about the change in reinforcement rate and there was no break between conditioning and test. We have clarified this in Figure 1 and in the Methods section:

– Figure 1: “Participants first underwent a brief conditioning phase of 5-6 trials in which Original Low Cues (gray) were followed by heat calibrated to elicit low pain (level 2) and Original High Cues (black) were followed by heat calibrated to elicit high pain (level 8). Conditioning was immediately followed by intermittent test trials, in which we delivered medium heat following each cue to test the effects of predictive cues on perceived pain. Following the initial test phase, participants in the Instructed Group were informed about reversals and we delivered medium stimuli to test the effects of instructions.”

– P. 45: “We used two trial orders (counterbalanced across participants within Group) that each included (1) a brief conditioning phase, (2) a test of cue-based expectancy effects, and (3) three contingency reversals (Figures 1C and 1D). During the conditioning phase, Original Low cues were followed by stimulation calibrated to elicit ratings of low pain and Original High cues were followed by stimulation calibrated to elicit high pain (see Figure 1C and 1D). The conditioning phase included 3 Original High cue + high heat pairings and 2-3 Original Low cue + low heat pairings (i.e. 5-6 trials; see Figure 1D). Following conditioning, each cue was paired intermittently with stimulation calibrated to elicit ratings of medium pain.”

7. The figures are very small and difficult to read. Please particularly check the violin plots in Figure 4 and 5 where parts of the figure legend seem to be missing.

We have edited the figures to ensure that they are easier to read and no information is omitted.

Reviewer #3 (Recommendations for the authors):Below, I list a few suggestions for improvement.1. Focus on stimuli of medium intensity: The authors nicely control for objective pain intensity by focusing on analyzing pain stimuli of medium intensity. However, I would find it very interesting if the reported pattern of results – especially the fit of the RW model variants – holds if stimuli of all intensities are included in the analyses.

This is an interesting question. We believe that fitting to medium intensity trials isolates the pure effect of expected value, as the medium trials are otherwise identical, which the Reviewer acknowledges. But it is true that the learning process depends on pairings with the low and high stimuli. We note that these are indeed included in the model, it’s just that goodness of fit is only evaluated on the medium heat trials. So in other words, they are included in the analysis, they are just not included in the evaluation of goodness of fit (see 3.1.a).

We fit the same RW models and evaluated goodness of fit using all trials rather than just medium trials. We again found that there was a complete reversal upon instruction in the Instructed Group, based on the instructed reversal parameter ρ (*M* = 0.93, *SD* = 0.5), consistent with our findings when we fit to medium heat trials. In addition, ρ values were significantly higher in the Instructed Group relative to the Uninstructed Group based on fits to individuals (t(1,38) = 7.40, *CI* = [0.20, 0.36], p <.0001). We also observed that learning rates were near-zero in the Instructed Group (*M* = 0.01 when fitting to all trials vs *M =* 0.06 when fitting to medium trials). However, within the Uninstructed Group the learning rate decreased from *M* = 0.28 to *M = 0*.04, and the between-groups difference in learning rate was no longer statistically significant (p = 0.19). We believe this is consistent with our position that pain on medium heat trials is the best index of value-based learning and learned expectations, whereas ratings on high and low intensity trials are influenced by stimulus intensity and not learning. We therefore have decided to retain our focus on models fit to medium heat trials in the revised manuscript but believe that this indicates that our results regarding how instructions guide learning are similar when we fit to all intensities as parameters were highly consistent within the group that was exposed to instructions.

a. Specifically, in the uninstructed group, the exact temporal succession of stimuli of high and low intensity should inform stimulus-by-stimulus learning. Maybe I just did not grasp correctly, which trials were included for fitting.

The reviewer is correct; the low and high intensity trials are what leads to stimulus-based learning. It is only pain reports on the medium trials that are tested as the outcome of such learning. We have clarified this in the revised Methods section:

– Pp. 51-52 additions in red: “In the present study we assume that pairings between predictive stimuli and high and low intensity heat engages stimulus-based learning, similar to CS-US pairings in classical conditioning experiments. Fitting to medium heat trials isolates the timecourse of expected value, since the stimulus temperature is constant, and therefore the only factor likely to guide cue-based variation in pain is presumably the cue’s dynamic expected value based on learning and/or instructions.”

b. As a sanity check, it would be nice to see the ratings and also the fMRI signals for all received intensities. I understand that the medium intensity stimuli control for objective differences – but still it would be nice to see the reflection of these objective differences in behavior and fMRI signals.

In fact, we did include a figure of pain ratings as a function of heat intensity in our original submission (Figure 2 —figure supplement 1), to accompany our statistical analysis of pain ratings as a function of temperature. We now include an additional analysis of brain responses as a function of stimulus intensity in response to the Reviewer’s suggestion. We also added a new figure (Figure 2), which includes the plot that was previously presented as a supplement along with images of brain responses as a function of heat intensity, pattern expression by the NPS and SIIPS, and effects of heat intensity on SCR. Brain responses show reliable increases as a function of heat intensity, and there was no difference between groups in the effect of heat intensity on signature pattern expression or responses within pain-related regions. Interestingly, ROI-wise analyses suggest that the Uninstructed Group showed stronger negative associations between temperature and VMPFC activation than the Instructed Group. However, we did not observe group differences in this region in uncorrected voxelwise analyses so we do not make strong inferences based on this comparison. We made the following changes:

– Added Figure 2, Figure 2 —figure supplement 1, accompanying Source Data, and updated Table 2 to include heat intensity effects on regions of interest.

– P. 9, deleted “figure supplement 1”: “We next examined pain as a function of heat intensity (i.e. temperature level: low, medium, or high) during the fMRI experiment (see Figure 2). ”

– P. 10, added: “We also evaluated brain responses to noxious stimulation as a function of heat intensity. We note that FDR-corrected thresholds exceeded 0.001 for all voxelwise analyses apart from moderation by group; we therefore interpret main effects of heat intensity at p <.001. We observed robust intensity-related changes within pain modulatory regions, including bilateral insula, striatum, dorsal anterior cingulate, thalamus, and other regions that did not differ between groups (see Figure 2E-G and Figure 2 – Source Data 2-3). Consistent with this, we observed robust expression of both the NPS and SIIPS as a function of temperature-related changes in both groups (all p’s < 0.001, see Table 2) and signature pattern expression did not differ by group (all p’s > 0.2; see Figure 2C and D and Table 2). Thus variations in heat intensity were positively associated with increases in pain-related activation in pain-related regions regardless of whether individuals were instructed about contingencies. Whole brain FDR-correction did reveal significant group differences in the left hippocampus and right primary somatosensory cortex driven by stronger intensity effects in the Instructed Group (see Figure 2H and Figure 2 – Source Data 2-3). Within value-related ROIs, we observed positive effects of heat intensity on the bilateral striatum that did not differ by Group, whereas the VMPFC showed significant Group differences, driven by negative associations between temperature and VMPFC activation in the Uninstructed Group, but not the Instructed Group (see Table 2). There were no associations between heat intensity and amygdala activation.”

c. By including all stimuli, the authors could also look into prediction errors (PEs) and into the expectations during the cue phase. The authors note that the study design was not optimized for such analyses and that PEs and expectations are negatively correlated. But I am wondering if this profoundly precludes such analyses – even though these would be exploratory.

As mentioned above, unfortunately our task and TR make it impossible for us to look at brain responses to cues independent of pain (see response 1.3). In addition, because outcomes are delivered on every trial, if we look at signed prediction errors (PEs) in across all trials (i.e. including all stimulus intensities), then PE will be correlated with temperature: High intensity stimuli will only be expected or worse than expected, whereas low intensity stimuli will always be expected or better than expected. If we control for stimulus intensity using a parametric modulator, most of the variance will be driven by medium trials, which are equally likely to be better or worse than expected; this is the inverse of the expected value.

With these caveats, we did analyze prediction error in two ways: (1) We analyzed associations between brain responses to all heat intensities and both signed and unsigned PEs, as suggested by the Reviewer; (2) We analyzed associations between brain responses to medium heat and unsigned PE (i.e. the absolute value of PE) which is only partially correlated with EV. We report the second analysis in the revised manuscript, and provide results of the first analyses here for transparency.

*PE on all trials:* When we included all trials in the analysis and examined signed PEs, we did indeed observe robust activation in the bilateral insula and thalamus, and no differences between groups, similar to what we observed when we analyzed responses to heat as a function of temperature (see Author response image 2). We therefore do not feel comfortable making inferences based on these results, for the reasons described above (i.e. signed PEs are correlated with stimulus intensity).

No regions survived multiple comparisons correction when we examined unsigned PE across temperatures, whether we searched whole brain or restricted to value-related ROIs. We believe that the best test of unsigned PE is on trials restricted to medium heat trials, as described in the following section and the revised manuscript.*Unsigned PE on medium heat trials:* We did observe interesting findings when we examined unsigned prediction errors at the time of heat onset. We focused on value-related regions based on voxelwise and ROI-based analyses. The bilateral amygdala and right striatum were significantly associated with unsigned PE and effects did not differ significantly across groups. Voxelwise analyses revealed positive associations with PE, controlling for group, in a wide swath of contiguous activation encompassing the right anterior insula, right striatum, and right amygdala. Groups also differed in response to unsigned prediction error: We observed group differences in the large contiguous cluster that included the right SII, right superior temporal gyrus, and right temporo-parietal cortex, driven by stronger associations in the Uninstructed Group.

We believe these findings provide important additional information about the effects of instruction and error-driven learning during pain modulation, and have included them in the revision. These results provide further evidence that striatal prediction errors update with instruction, building on previous findings in bot8/29/22 12:20:00 PMh appetitive and aversive learning (Doll et al., 2009, 2011; Atlas et al., 2016). Although a large body of work focuses on reward prediction errors in the striatum, many studies indicate that the striatum is also sensitive to aversive PEs (Delgado et al., 2008; Garrison et al., 2013; Robinson et al., 2013; Corlett et al., 2022), and meta-analysis indicates that unsigned PEs are also reliably activated by unsigned PEs (Corlett et al., 2022). These results also raise new questions about the role of the amygdala in learning and the impact of instructions, which we hope will be addressed in future research.

Please note that while the PE signal varied strongly from trial to trial in the Uninstructed Group, who had higher learning rates, PE was associated with a gradual reduction in the Instructed Group (see revised Figure 7), as expected value only updated upon instruction in this group. Although the overall difference in unsigned PE magnitude from the beginning of the study to the end was small based on model fits, when used as a parametric modulator, this isolates effects that vary linearly across the duration of the study. This is consistent with associability, which is high early in learning and decreases as an environment becomes stable, and is also consistent with findings that amygdala responses habituate during learning (Büchel et al., 1998). We acknowledge these links explicitly in our discussion, as we recognize that the lack of stochasticity in the Instructed Group’s unsigned PE pattern could be surprising to some readers and deserves attention.

We have incorporated the results from the analysis of unsigned PEs on medium trials in our revision. We made the following changes:

– Updated Figure 7 to visualize responses across all trials and restricted to medium trials and to incorporate timecourse of PE

– Added results of PE analyses to Figure 8 and associated Supplementary Figures and Source Data

– Pp. 28-29, added: “Associations with unsigned prediction error differ between groups. In addition to analyses of EV, we evaluated associations between brain responses to medium heat and unsigned prediction errors (PEs). No regions showed significant associations within pain related ROIs; however, whole brain correction revealed a positive association with PE, controlling for group, in a wide swath of contiguous activation encompassing the right anterior insula, right striatum, and right amygdala (see Figure 8D). Whole brain correction also revealed a significant difference between groups in a large contiguous cluster that included the right SII, right superior temporal gyrus, and right temporo-parietal cortex (see Figure 8E) driven by negative associations with PE in the Instructed Group and positive associations in the Uninstructed Group (see Figure 8 —figure supplement 8 and Figure 8 – Source Data 6). Within value-related ROIs, unsigned PEs were positively associated with responses to heat in the right striatum and the bilateral amygdala (see Figure 8F and Table 2). Associations with right striatum were observed within each group, whereas associations with amygdala were only observed in the Instructed Group; however, group differences were not significant in any region. Finally, there was no association between PE and NPS or SIIPS expression in either group (see Table 2). Voxelwise uncorrected results are reported in Figure 8 —figure supplement 9 and Figure 8 – Source Data 6.”

– P. 36, added: “Because we focused on responses to the US rather than the CS here, we also were able to extend previous work by examining the role of unsigned prediction errors (PEs), which are linked to associability, which gates attention and learning dynamically. Associability has been linked to the amygdala in previous studies of pain and aversive learning (Li et al., 2011; Zhang et al., 2016; Atlas et al., 2019), meta-analyses indicate that unsigned PEs are associated with activation in the dorsal and ventral striatum, as well as other regions (Corlett et al., 2022), and one study of aversive learning showed responses to both associability and unsigned PE in the amygdala (Boll et al., 2012). Consistent with these studies, we observed associations between unsigned PE and activation in the bilateral amygdala and the right striatum in both groups in ROI-based analyses. Whole brain analyses revealed associations between unsigned PEs and activation in the right hemisphere encompassing the insula, striatum, and amygdala, and that the associations between PE and activation in the right SII and temporal cortex differed across groups (negative associations in the Instructed Group and positive in the Uninstructed Group). Thus, although cue effects on subjective pain were mediated similarly across groups, we still observe unique responses to PE that differed between groups.”

– Pp. 36-37, added: “We note that the amygdala’s association with PE within the Instructed Group indicates that instructions do have some impact on amygdala activation, seemingly in contrast to our previous work on instructed learning during fear conditioning (Atlas et al., 2016; Atlas 2019). However, because error-driven learning was nearly absent between instructed reversals in the Instructed Group, the PE regressor in this group captures a constant effect that decreases linearly over time, consistent with prior evidence that amygdala responses habituate during aversive learning (Büchel et al., 1998). Thus we did not observe evidence of amygdala sensitivity to instructions per se in the present study.”

– P. 57: “Neural correlates of expected value and prediction error”

– Pp. 57-58, additions in red: “Noxious stimulation might also be accompanied by prediction errors; for example, if an individual expects high pain and receives medium heat, this should generate an appetitive PE (i.e. better than expected) if the deviation is noticed. However, expected value and prediction error are inversely correlated in the standard RL model we used. We therefore measured associations with unsigned PE (i.e. the absolute value of PE), as signed PE is the inverse of expected value on our medium heat trials. We focused on responses to medium heat trials only as PE would be correlated with temperature if we analyzed responses across all trials (i.e. low temperatures are always expected or better than expected, whereas high temperatures are always expected or worse than expected).”

2. Physiological data: The authors note that they might look into those data in the future. A slightly more nuanced explanation of the reasons for refraining from analyzing the physiological data might be helpful (especially given that the authors have experience with such analyses).

Thank you for suggesting we revisit our decision to exclude the autonomic data. As mentioned above we felt the paper was already complicated and that this might provide confusion, and the number of subjects and trials who displayed autonomic responses was limited. However in response to suggestions from all three reviewers, we now incorporate SCR and pupil dilation data, along with appropriate caveats. Please see Response 1.4 above for details.

3. Brain responses that are unaffected by reversals: In my view, this is an interesting an unexpected finding (see e.g., Figure 5). Can it be excluded that these signals do not "just" relate to the differences between the cues in a rather unspecific way? Can it be that the first reversal induces a second "context" and that there are brain signals (maybe in the same regions) that "remain stable" in this new "switched context?" Relatedly, the authors might want to conjecture whether such "stable" responses would become unstable with longer learning periods in the switched context. I am rather unfamiliar with the role of these regions for pain perception.

We agree that this finding is interesting and unexpected: While we expected that some regions would respond to temperature independent of expectancy (e.g. those that track nociception rather than pain), it is more surprising to find that there are regions that are modulated by expectancy, but do not update as contingencies change, despite the fact that pain does reverse. We have expanded our discussion on the role of the RVM in pain perception in general (see Response 2.2 above), and included raw timecourses for all regions that show this pattern (Figure 6 —figure supplement 3). In both groups, all regions that responded to original contingencies show a specific increase in response to high temperatures during the heat period, and a similar (albeit weaker) increase at the same time when medium heat is preceded by the high pain cue, although based on the average timecourses this effect is strongest in the Instructed Group and during initial contingencies. So we believe that this means this is not an unspecific difference between cues, which we think is in agreement with the Reviewer’s point, if we are interpreting the first question correctly.

The Reviewer has also asked us to consider the role of learning and what would happen if we had longer learning periods. The timecourses are also instructive toward this point. Indeed, the influence of original cues is most pronounced for both groups during original contingencies, which could mean that rather than these regions being impervious to reversals, they may have a slower learning rate than other regions. One putative mechanism for this would be the engagement of endogenous opioids, which we have hypothesized lead to longer-lasting effects than dopaminergic prediction errors. Although our initial findings pointed to the PAG as a key region that did not reverse, we no longer observe consistent findings in the PAG; however, we instead see maintenance of initial learning in a region consistent with the rostroventral medulla (RVM). The RVM is a hub for the engagement of descending pain modulation through the µ-opioid receptor system, and preclinical work suggests that µ-opioid receptor-mediated pain inhibition can be systemic and long lasting. Thus if pain-predictive cues modulate pain by engaging descending inhibition, it is possible that the brain systems that mediate these effects are slower to update than dopaminergic circuits or prefrontal systems involved in higher order decision making. We acknowledge that this is highly speculative, but have included this logic, as well as the possibility that different circuits have different learning rates and that this appears in our task as failure to reverse (since our task has relatively rapid reversals), in our revised discussion, provided above in response 2.2. We thank the reviewer for suggesting we consider these points.

4. Learning rates of the instructed group: I find the wording regarding the learning rate a bit confusing. It sounds a bit as if the authors did not expect the learning rates to be low (or even zero). But given the explicit instructions and the nice RW model variant, this is exactly how they should look. The instructions say everything and no learning is needed. Testing instructions that are incorrect (as might be the case for some types of placebo/nocebo instructions) would be really interesting here; i.e., participants get information but still have to learn from experience (in spite of the received information). Maybe the authors have that idea in their mind when reporting their findings about the learning rates of the instructed group.

The Reviewer is correct; because participants receive valid instructions, it is not surprising that we do not observe substantial additional learning from reinforcement. We agree that approaches such as those used by Doll et al., 2009 in which experiential learning is inconsistent with instructions would engage dynamic learning and necessitate a higher learning rate. We modified the following sections accordingly:

– P. 24, added: “, as might be expected given that feedback was entirely consistent with instructions”

– P. 38, added: “Alternatively, future studies can include outcomes that diverge from instructions to examine the interplay between instructed and experiential learning (e.g., Doll et al., 2009).”